# Deep CRISPR mutagenesis characterizes the functional diversity of *TP53* mutations

Julianne S. Funk [1], Maria Klimovich[1], Daniel Drangenstein[1], Ole Pielhoop[1], Pascal Hunold [1], Anna Borowek[1], Maxim Noeparast [1], Evangelos Pavlakis[1], Michelle Neumann[1], Dimitrios-Ilias Balourdas [2,3], Katharina Kochhan[1], Nastasja Merle[1], Imke Bullwinkel[1], Michael Wanzel[1], Sabrina Elmshäuser[1], Julia Teply-Szymanski[4], Andrea Nist[5], Tara Procida[6], Marek Bartkuhn [6,7], Katharina Humpert[1,8], Marco Mernberger[1], Rajkumar Savai [6,9,10,11], Thierry Soussi [12,13], Andreas C. Joerger [2,3] & Thorsten Stiewe [1,5,6,9] ✉

The mutational landscape of *TP53*, a tumor suppressor mutated in about half of all cancers, includes over 2,000 known missense mutations. To fully leverage *TP53* mutation status for personalized medicine, a thorough understanding of the functional diversity of these mutations is essential. We conducted a deep mutational scan using saturation genome editing with CRISPR-mediated homology-directed repair to engineer 9,225 *TP53* variants in cancer cells. This high-resolution approach, covering 94.5% of all cancer-associated *TP53* missense mutations, precisely mapped the impact of individual mutations on tumor cell fitness, surpassing previous deep mutational scan studies in distinguishing benign from pathogenic variants. Our results revealed even subtle loss-of-function phenotypes and identified promising mutants for pharmacological reactivation. Moreover, we uncovered the roles of splicing alterations and nonsense-mediated messenger RNA decay in mutation-driven *TP53* dysfunction. These findings underscore the power of saturation genome editing in advancing clinical *TP53* variant interpretation for genetic counseling and personalized cancer therapy.

p53, a master regulatory transcription factor, suppresses the proliferative fitness of cancer cells through mechanisms such as cell-cycle arrest, senescence and apoptosis[1]. Mutations in the *TP53* gene are observed in about half of all cancers and, as germline mutations, cause Li–Fraumeni syndrome[2,3]. Despite their prognostic significance[4], integrating *TP53* mutations into clinical decision-making is limited by the complexity of their mutational landscape. Most *TP53* mutations are missense, with over 2,000 identified, predominantly clustering in the DNA-binding domain (DBD)[2]. While the ten most common (and also most studied) 'hotspot' mutants account for ~30% of cases, the remaining ~70% are poorly characterized, making it difficult to predict their pathogenicity and clinical impact[5].

First and foremost, *TP53* mutations result in a loss of p53's tumor suppressor function (loss of function, LOF), which is sufficient to initiate tumorigenesis in humans and mice[6,7]. In some cases, secondary alterations such as aneuploidy can lead to accumulation of missense mutant

[1]Institute of Molecular Oncology, Philipps-University, Marburg, Germany. [2]Institute of Pharmaceutical Chemistry, Goethe University, Frankfurt am Main, Germany. [3]Buchmann Institute for Molecular Life Sciences and Structural Genomics Consortium (SGC), Frankfurt am Main, Germany. [4]Institute of Pathology, Philipps-University, Marburg University Hospital, Marburg, Germany. [5]Genomics Core Facility, Philipps-University, Marburg, Germany. [6]Institute for Lung Health (ILH), Justus Liebig University, Giessen, Germany. [7]Biomedical Informatics and Systems Medicine, Justus-Liebig-University, Giessen, Germany. [8]Bioinformatics Core Facility, Philipps-University, Marburg, Germany. [9]Universities of Giessen and Marburg Lung Center (UGMLC), German Center for Lung Research (DZL), Giessen, Germany. [10]Cardio-Pulmonary Institute (CPI), Giessen, Germany. [11]Lung Microenvironmental Niche in Cancerogenesis, Max Planck Institute for Heart and Lung Research, Bad Nauheim, Germany. [12]Centre de Recherche Saint-Antoine UMRS_938, INSERM, Sorbonne Université, Paris, France. [13]Department of Immunology, Genetics and Pathology, Science for Life Laboratory, Uppsala University, Clinical Genetics, Uppsala University Hospital, Uppsala, Sweden. ✉e-mail: stiewe@uni-marburg.de

proteins that gain neomorphic (gain of function, GOF) properties, promoting tumor growth[5,8–13]. Understanding the functional impact of distinct mutants is clinically crucial for personalized treatment and genetic counseling, but the rarity of many individual mutations makes this challenging. High-throughput screens in isogenic models, such as multiplexed assays of variant effects[14–16], are therefore valuable tools for annotating the *TP53* mutational landscape.

A notable early study screened a complementary DNA library of 2,314 missense variants in a yeast system, revealing widespread LOF but also heterogeneity, with many nonhotspot mutants retaining partial activity[17]. However, yeast lacks the full p53 regulatory network, prompting further screens in human cells[18–20]. While cDNA-based screens in human cells offered important insight, they faced limitations, including nonphysiological expression, absence of post-transcriptional control and lack of (alternative) splicing. These studies also did not assess the impact of p53 mutations on responses to cancer treatments such as radiation, chemotherapy or targeted therapies[18–20].

CRISPR-based methods, which introduce *TP53* variants directly into the endogenous gene locus, provide a more physiological and comprehensive insight into their functions. Recent proof-of-principle studies using CRISPR base or prime editing show promise but still face challenges in achieving full coverage of the mutational landscape[21–24]. In this study, we utilized CRISPR–Cas9-mediated gene editing through precise homology-directed repair (HDR), known as saturation genome editing (SGE)[15,25,26], which has previously been instrumental in defining the functional impact of mutations in genes such as *BRCA1*, *BRCA2*, *CARD11*, *DDX3X*, *BAP1* and *VHL*[25,27–33]. Leveraging this powerful technology, we introduced a panel of 9,225 variants, comprising approximately 94.5% of all *TP53* cancer mutations, into cancer cells with a wild-type (WT) *TP53* gene locus. Unlike cDNA overexpression screens, CRISPR-based editing preserves physiological gene regulation, including endogenous promoters, enhancers, alternative splicing and microRNA binding sites.

We evaluated the effects of these variants on proliferative fitness following p53 pathway activation with Mdm2 inhibitors, finding similar results across other p53 stimuli, including radiation, chemotherapy and starvation. These fitness effects correlated with mutation frequency in patients, evolutionary conservation and structure–function relationships. CRISPR editing also enabled the accurate annotation of partial LOF (pLOF) and splice mutations, demonstrating widespread elimination of frameshift or nonsense transcripts via nonsense-mediated decay (NMD). Furthermore, we identified synonymous and missense mutants, previously considered functionally normal, that altered messenger RNA splicing and resulted in complete LOF. For instance, the recurrent L137Q mutation caused an in-frame deletion, which is targetable by splice-switching oligonucleotides (SSOs), providing proof-of-principle for a p53 reactivation strategy.

## Results

### Isogenic model for *TP53* mutagenesis by CRISPR-HDR

To assess the functional impact of *TP53* variants in a controlled isogenic environment, we used HCT116 colorectal carcinoma cells, which are *TP53* WT with a prototypical p53 response[34–37]. Refining established SGE techniques[15,25,26], we inactivated one of the two *TP53* alleles to ensure unambiguous genotype–phenotype correlations (Fig. 1a, Extended Data Fig. 1a–c and Supplementary Note 1). To avoid confounding effects from p53's DNA-damage response during CRISPR–Cas9 gene editing[38–40], we reversibly silenced expression from the remaining *TP53* copy using a LoxP-flanked transcriptional stop cassette (LoxP-Stop-LoxP, LSL) containing selection markers. For mutagenesis via HDR, the resulting HCT116 LSL/Δ cell line was transfected with a CRISPR–Cas9 nuclease and a donor vector providing the desired mutation for templated repair.

We validated the editing performance by introducing a panel of *TP53* variants, including some common cancer mutations with

known LOF or pLOF, a nonsense mutation and the WT for reference. We observed successful donor integration in 75.9% of clones and specific mutations in 56.4% (Fig. 1b). After Cre excision of the LSL cassette, we found comparable p53 protein expression levels in WT and missense mutants, further enhanced by Mdm2 inhibition with Nutlin-3a (N3a) (Fig. 1c and Extended Data Fig. 1d,e). As expected, N3a induced p21/CDKN1A expression and characteristic p53 signatures in WT and pLOF mutant cells, but not in LOF missense or nonsense mutants (Fig. 1c–e and Supplementary Fig. 1). Real-time live-cell imaging confirmed growth inhibition in WT cells, which was diminished by pLOF mutations, and fully abrogated by LOF mutants (Fig. 1f,g and Extended Data Fig. 1f,g). Single-cell RNA sequencing (RNA-seq) further confirmed LOF effects and revealed minimal clonal variability (Extended Data Fig. 2 and Supplementary Note 2).

Notably, we did not observe GOF effects in any of the missense variants under these conditions. The GOF of missense variants, particularly R175H, is best documented for promoting metastasis[41,42], and depends on secondary alterations that stabilize the mutant p53 protein as it is inherently unstable in nontransformed cells[9,13,43,44]. We did not observe constitutive stabilization in our engineered HCT116 cells, and mutant p53 levels remained similar to WT levels in parental HCT116 and other nontransformed cell types (Fig. 1c and Extended Data Fig. 1h). N3a-induced stabilization was significantly lower than that seen in tumor cells with natural *TP53* mutations (Extended Data Fig. 1i,j) and insufficient to drive cell migration (Extended Data Fig. 3a–c). However, serial in vivo passaging revealed progressively increasing mutant p53 protein levels (Extended Data Fig. 3d,e), coinciding with increased R175H-dependent migration, invasion and liver metastasis in a subcutaneous xenograft model (Extended Data Fig. 3f–r and Supplementary Note 3).

In conclusion, deleterious *TP53* mutations in HCT116 cells immediately caused LOF, increasing proliferation and survival under p53-activating conditions (Fig. 1 and Extended Data Figs. 1 and 2). In contrast, potential GOF effects, as shown for R175H, manifested only after long-term in vivo passaging, promoting migration, invasion and metastasis without impacting proliferative fitness (Extended Data Fig. 3). Therefore, measuring proliferative fitness shortly after mutagenesis, particularly under p53 activation with N3a, effectively captures LOF effects, while minimizing the influence of GOF effects.

### R175 mutational scan shows functional diversity in variants

Leveraging the editability of HCT116 LSL/Δ cells, we conducted a mutational scan of codon R175, the most frequently mutated p53 codon in cancer. We generated a library of 27 distinct variants, including missense substitutions, deletions/insertions, and nonsense and silent/synonymous mutations. We co-transfected HCT116 LSL/Δ cells with a *TP53*-targeting CRISPR–Cas9 nuclease and the R175 variant library, maintaining an average coverage of at least 1,000 independently edited cells per variant (Fig. 2a). Targeted amplicon sequencing validated the editing, confirming that variant distributions in the donor plasmid matched those in the edited cell libraries across biological replicates, even after Cre-induced recombination to activate *TP53* variant expression (Fig. 2b–d and Supplementary Table 1). In the absence of treatment, the variant distribution in the Cre-recombined cell libraries remained stable for 8 weeks, with only minor depletion of synonymous variants (Fig. 2e).

Upon N3a treatment, we observed a time- and dose-dependent shift in variant distribution (Fig. 2e and Supplementary Fig. 2a). The pattern remained consistent across a range of different Mdm2 and Mdmx inhibitors (Fig. 2f and Supplementary Fig. 2b). Synonymous variants became depleted, while frameshift and nonsense variants—grouped as 'null' mutations—were enriched, as expected for LOF mutations. Missense variants showed varied responses, allowing us to classify them into three categories: LOF variants such as R175H, pLOF variants

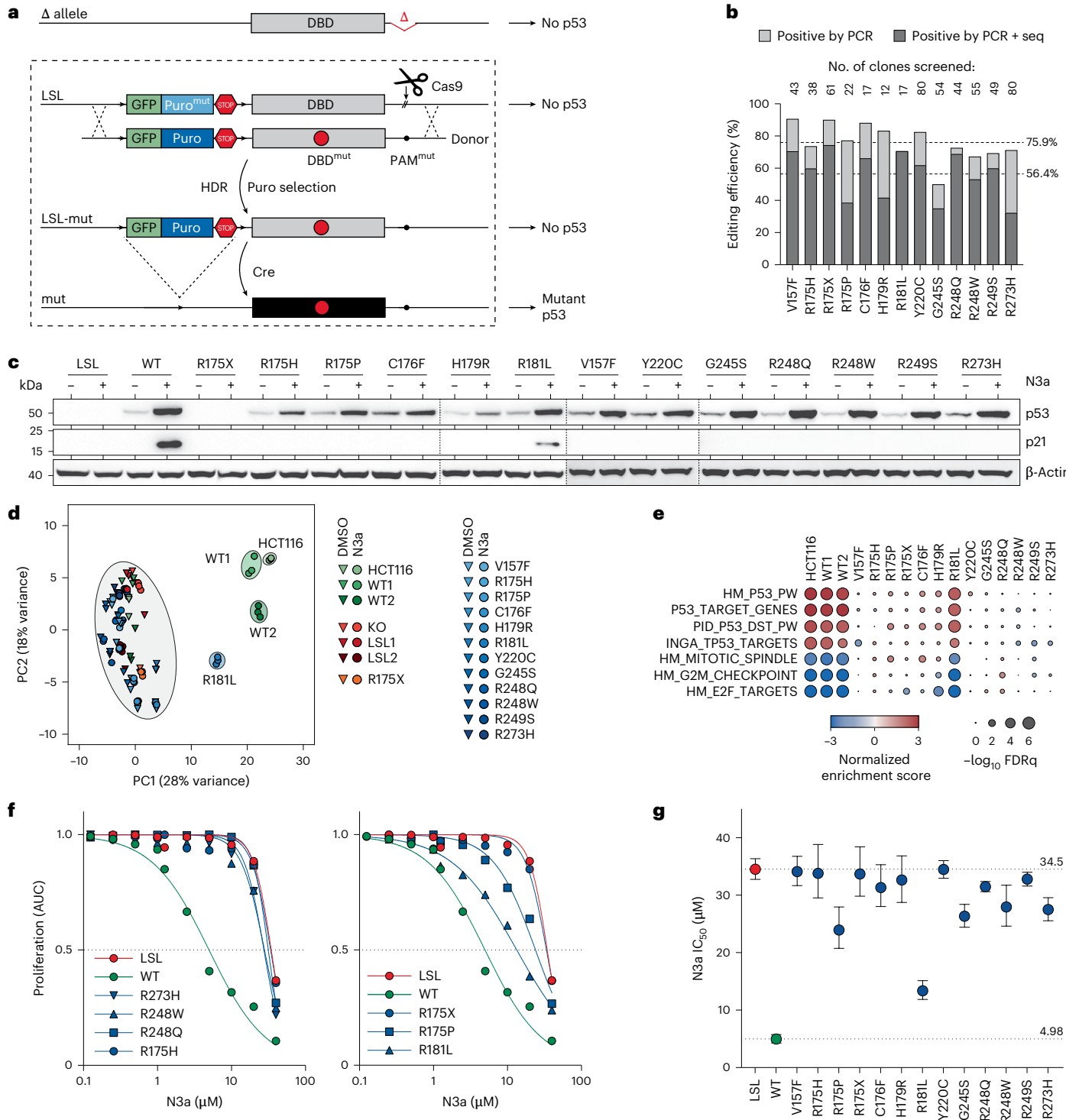

**Fig. 1 | Panel of single *TP53* mutations in HCT116 cell lines. a**, Scheme for CRISPR–Cas9-mediated *TP53* mutagenesis via homology-directed repair (HDR) in HCT116 LSL/Δ cell line. **b**, Editing efficiency as percentage of single-cell clones that contain a targeted integration of the donor and the desired mutation analyzed by PCR and sequencing, respectively. Shown are results for single mutations and the mean across the panel. **c**, Western blot demonstrating mutant p53 and p21 protein expression in HCT116 clones after Cre-mediated excision of the LSL cassette in absence and presence of 10 μM N3a. **d**, Principal component analysis based on RNA-seq data of indicated cell clones ±N3a. **e**, Gene set enrichment analysis for p53-related gene expression signatures comparing indicated N3a- and DMSO-treated cell clones. **f,g**, Proliferation of *TP53*-mutant cell clones in presence of increasing concentrations of N3a analyzed by real-time live-cell imaging. **f**, Area under the proliferation curve relative to untreated. **g**, 50% inhibitory concentration (IC$_{50}$, with 95% CI) for N3a with p53-null (LSL, red) and WT (green) as reference. 95% CI, 95% confidence interval; AUC, area under the curve; FDRq, false discovery rate q-value; LSL, LoxP-Stop-LoxP; Puro, puromycin.

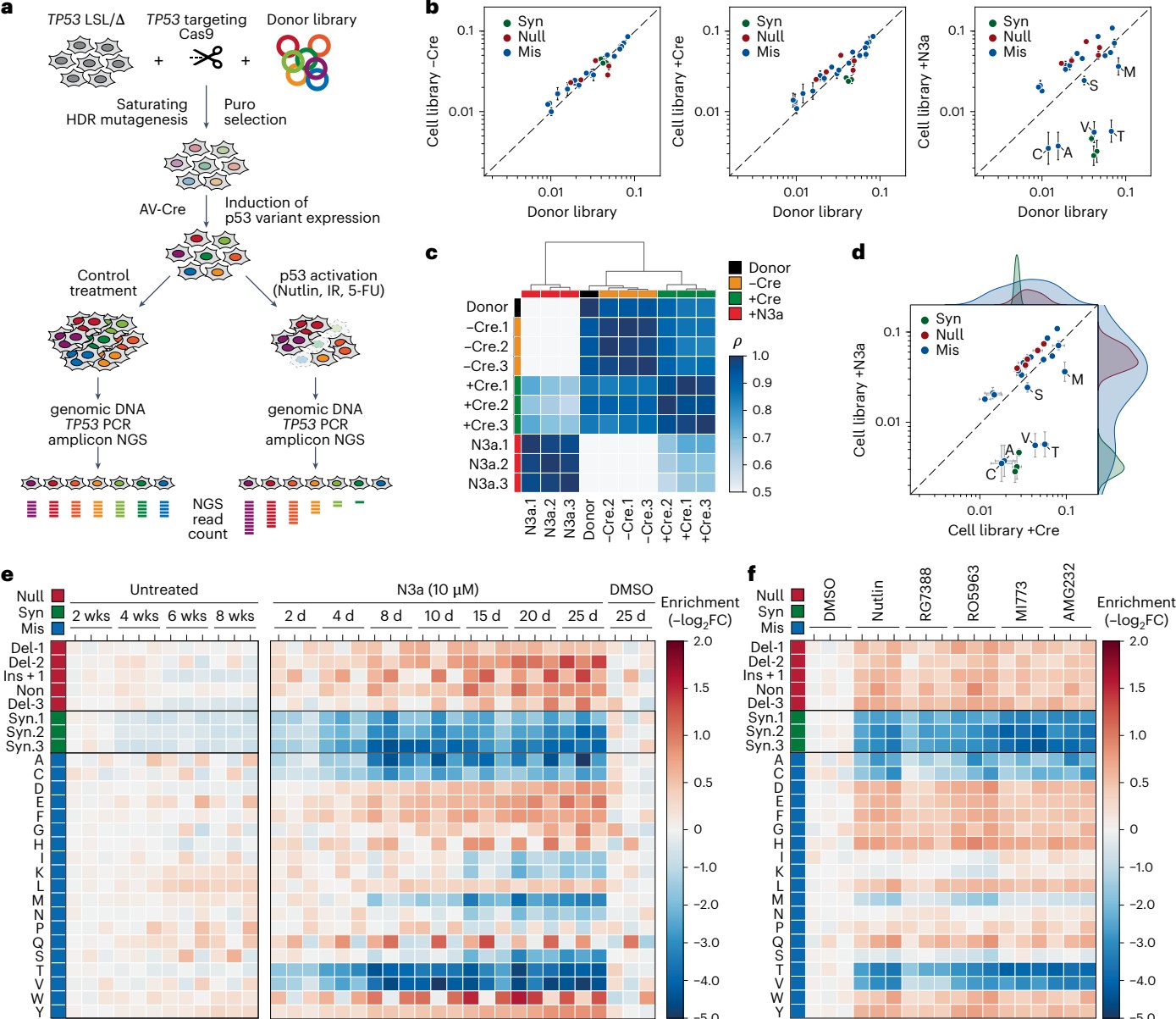

**Fig. 2 | Saturating mutagenesis scan of *TP53* codon R175. a**, Scheme for CRISPR–Cas9-mediated saturating mutagenesis scan via HDR in HCT116 LSL/Δ cell line and analysis of p53-mediated stress responses by NGS. **b**, Quality control plots illustrating correlation of variant abundance between donor (plasmid) library and variant cell libraries before and after Cre recombination (−Cre and +Cre) and following 8 d of N3a treatment (+N3a). Shown is the mean ± s.d. abundance (*n* = 3 biological replicates) for synonymous (syn, green), null (red) and missense (mis, blue) variants. Missense variants that are depleted by N3a are individually labeled. Dashed line, line of identity. **c**, Heatmap showing pair-wise correlation coefficients (*ρ*, Spearman). Dendrogram shows hierarchical clustering of samples using average linkage and Euclidean distance. **d**, Quality control plots illustrating correlation of variant abundance between variant cell libraries after Cre recombination (+Cre) and following 8 d of N3a treatment (+N3a). Shown

is the mean ± s.d. abundance (*n* = 3 biological replicates) for synonymous (syn, green), null (red) and missense (mis, blue) variants. Missense variants that are depleted by N3a are individually labeled. Kernel density estimation plots illustrate separation of variant classes following N3a treatment. Dashed line, line of identity. **e**, Heatmaps showing the temporal changes of variant abundance in the absence or presence of N3a (*n* = 3 biological replicates per condition). Enrichment or depletion is shown as the −log₂ fold change relative to control conditions: the mean of the 2-week untreated samples (left panel) and the 25-d DMSO-treated samples (right panel). **f**, Response to Mdm2/Mdmx inhibitors. Heatmap of variant enrichment/depletion after 8 d of treatment relative to the mean of the DMSO-treated control replicates (*n* = 3 biological replicates per condition). del, deletion; ins, insertion; IR, ionizing radiation; 5-FU, 5-fluorouracil; FC, fold change; non, nonsense; wks, weeks.

and WT-like variants that behaved similarly to synonymous mutations (Supplementary Fig. 2c). We repeated the scan in H460 lung adenocarcinoma cells and obtained highly correlated results (Extended Data Fig. 4 and Supplementary Note 4), suggesting that the fitness impact of these mutations is conserved across cell types.

Importantly, none of the R175 missense variants significantly enhanced cellular fitness beyond the effect of nonsense mutations, indicating again, at least by this measure, no discernible GOF

phenotype. All recurrent R175 variants found in cancer fell into the LOF and pLOF categories, with the most frequent ones uniformly classified as LOF. This demonstrates the mutational scan's power to correctly identify cancer-associated variants.

We further evaluated the response of R175 variants to different p53-activating stimuli, including DNA damage and nutrient deprivation. We treated the R175 cell library with varying doses of radiation, 5-fluorouracil, starvation in Hank's balanced salt solution (HBSS) and

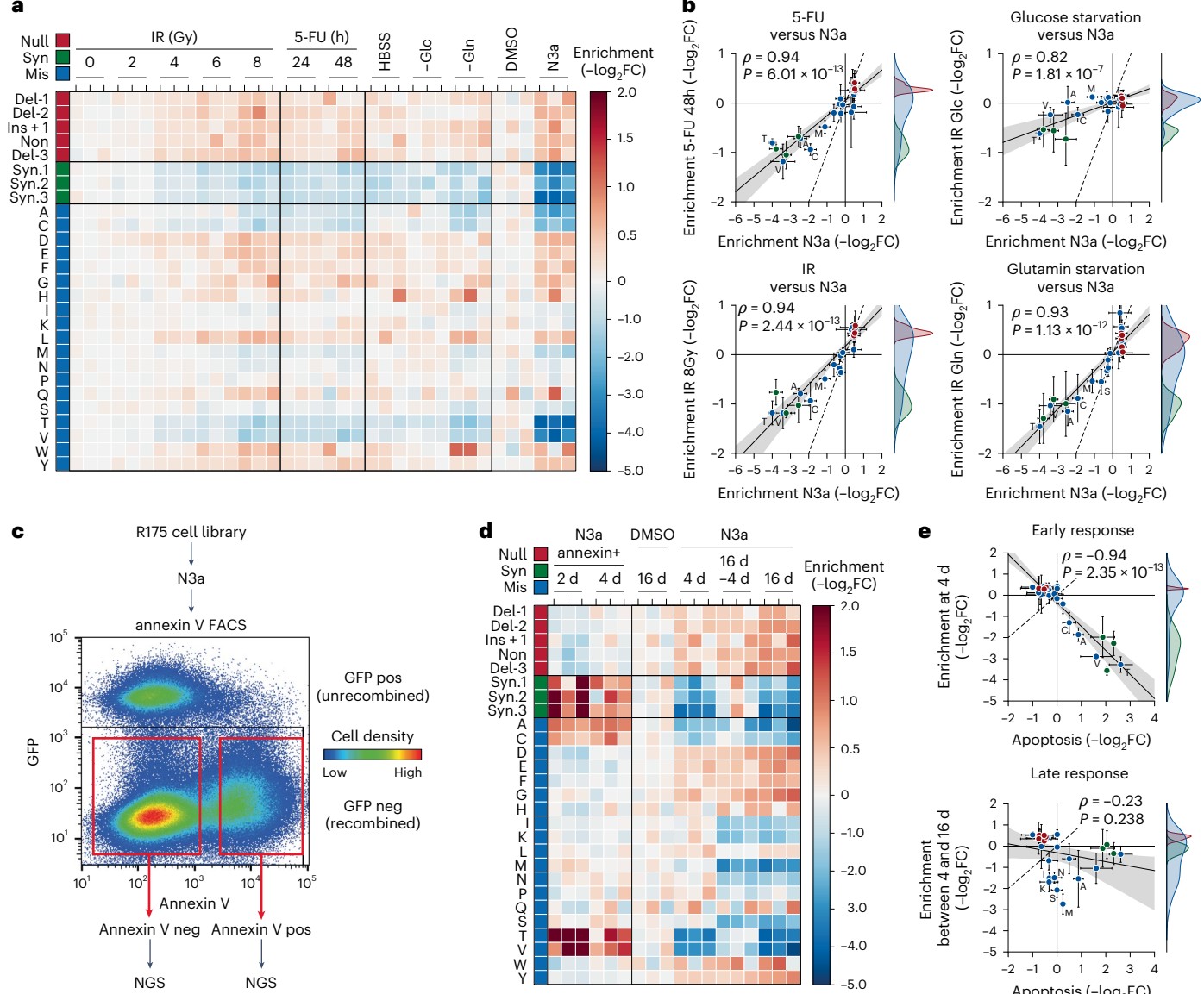

**Fig. 3 | Differential impact of R175 variants on stress responses and effector mechanisms. a,b,** Comparison of different stress factors. **a,** Heatmap showing changes in variant abundance in response to DNA damage (IR, ionizing radiation; 5-FU, 5-fluorouracil) or nutrient starvation (−Glc, glucose starvation; −Gln, glutamine starvation) compared with control treatment with DMSO and N3a. Shown is the enrichment as the −log$_2$ fold abundance change relative to the mean of the controls: unirradiated cells for IR samples, untreated cells for 5-FU samples and unstarved cells in regular growth medium for starvation samples ($n$ = 3 biological replicates per condition). **b,** Scatter plots illustrating the correlation between enrichment under DNA damage or nutrient deprivation and specific p53 activation with N3a. Shown is the mean ± s.d. enrichment ($n$ = 3 biological replicates). Dashed line, line of identity. **c–e,** Proapoptotic activity of R175 variants. **c,** Experimental scheme and a representative FACS scatter plot demonstrating the sorting strategy based on annexin V staining. GFP-negative (neg) cells were gated to selectively analyze cells expressing the p53 variant, that

is, cells with successful deletion of the GFP-expressing LSL cassette after AV-Cre infection. **d,** Heatmap illustrating N3a-induced changes in variant abundance in the annexin V-positive (pos) fraction (left) compared with the entire cell pool (right). Shown is the −log$_2$ fold change ($n$ = 3 biological replicates) relative to the annexin V-negative fraction (left) or DMSO-treated control cells (right). Lanes labeled as '16 d–4 d' represent the difference between the 4 d and 16 d timepoint, reflecting late N3a-induced changes in variant abundance. **e,** Scatter plot showing the correlation between the early (4 d) and late (between 4 and 16 d) occurring N3a-induced changes in variant abundance versus their enrichment in the apoptotic cell fraction. Shown is the mean ± s.d. enrichment ($n$ = 3 biological replicates) relative to the DMSO-treated control. All scatter plots show the Pearson correlation coefficient $\rho$ with $P$ value approximated using a two-tailed $t$-distribution and kernel density estimation plots on the side to illustrate the separation of variant classes. Dashed line, line of identity. FACS, fluorescence-activated cell sorting.

selective deprivation of glucose or glutamine (Fig. 3a). Under all conditions, the fitness effects mirrored those observed with N3a treatment, although the overall effects were less pronounced (Fig. 3b). This suggests that p53-independent mechanisms diluted the impact of p53 variants under these stress conditions. These results indicate that Mdm2 inhibitors, because of their selectivity for the p53 pathway, more effectively discriminate the functional differences among p53 variants than other p53-activating stimuli.

Next, we investigated whether known p53-reactivating compounds could rescue the tumor suppressive activity of p53 mutants (Supplementary Fig. 3). We treated the R175-mutant HCT116 cell pools with APR-246 and ZMC1, two compounds that have been reported to restore mutant p53 function[45,46]. However, neither compound, even when combined with N3a, selectively depleted R175H or other missense variants. This result indicates that these compounds cannot effectively reactivate R175 missense mutants to reduce proliferative

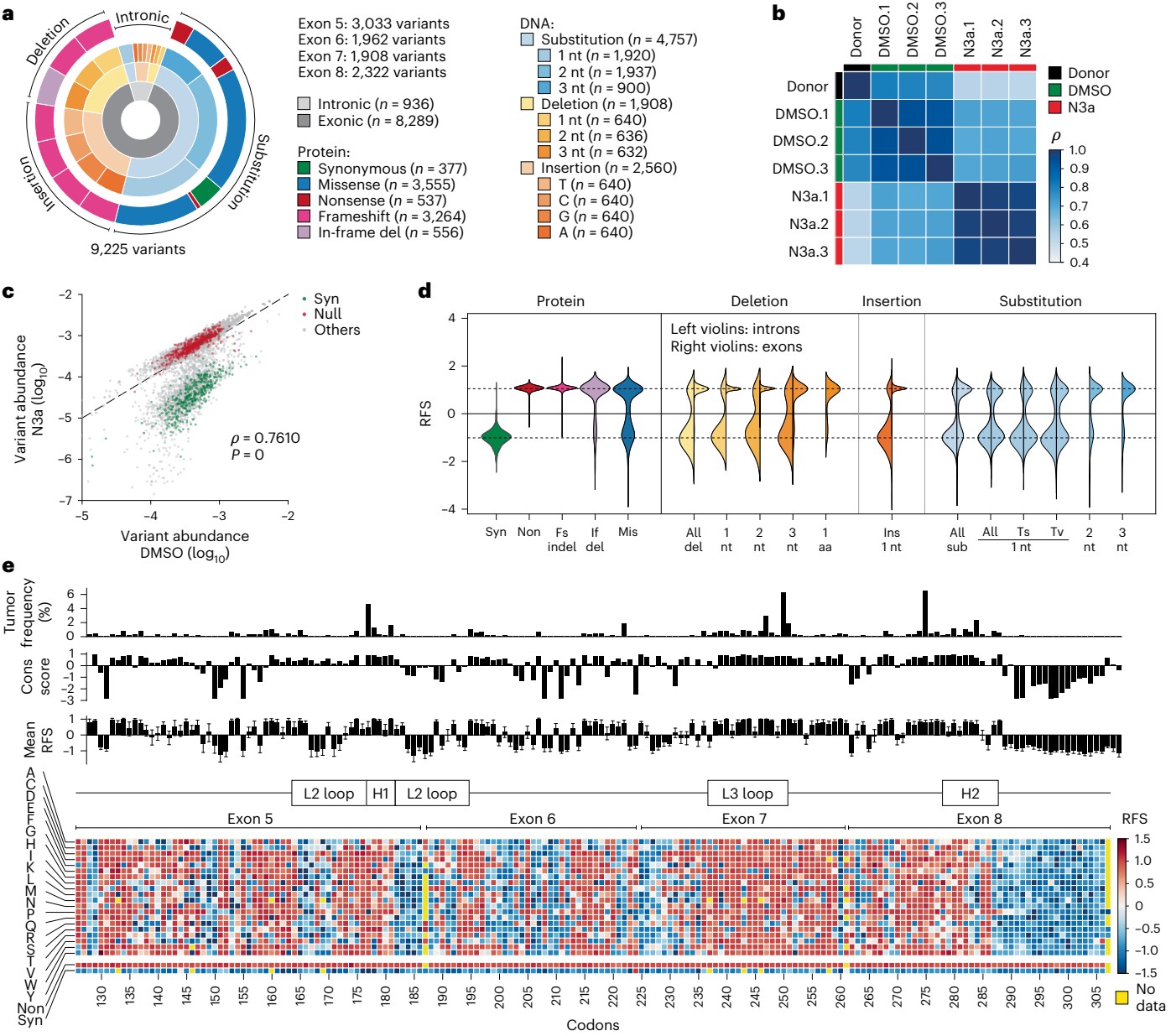

**Fig. 4 | *TP53* DBD variant screen. a**, Composition of the *TP53* DBD mutagenesis library. **b**,**c**, Quality control plots. **b**, Heatmap showing pair-wise correlation coefficients ($\rho$, Spearman) between sample replicates. **c**, Scatter plot illustrating separation of variants under p53-activating N3a treatment. Shown is the median abundance of all variants under N3a versus DMSO treatment ($n = 3$ biological replicates). Synonymous (syn) and nonsense (non) variants highlighted in green and red, respectively. $\rho$, Spearman correlation coefficient with $P$ value approximated using a two-tailed $t$-distribution. Dashed line, line of identity.

**d**, Distribution of RFSs for different variant classes. Left violin half shows distribution for intronic, right violin half for exonic variants. **e**, Heatmap showing the RFS for all mis, syn and non variants. Bar plots show for each codon the mutation frequency in the UMD *TP53* mutation database, the evolutionary conservation score and the RFS (mean ± s.d.) of all missense substitutions at this position. Fs, frameshift; if, in-frame; indel, insertion or deletion; nt, nucleotide; sub, substitution; Ts, transition; Tv, transversion.

fitness, supporting earlier studies that link their therapeutic effects to redox homeostasis rather than direct p53 reactivation[47–51].

Moreover, we noted variant-specific differences in response kinetics. Some variants, such as R175T, were rapidly depleted, coinciding with N3a-induced apoptosis, while others, such as R175S, showed slower depletion, likely due to cell-cycle arrest (Fig. 2e). To confirm this, we sorted apoptotic cells based on annexin V staining after 2 and 4 d of N3a treatment (Fig. 3c). Variants that depleted quickly were enriched in the apoptotic fraction (Fig. 3d,e), identifying apoptosis as the crucial mechanism reducing their fitness. In contrast, slowly depleted mutants were absent from the apoptotic fraction, supporting the idea that cell-cycle arrest, rather than apoptosis, drove their depletion.

Further experiments with single R175 variants confirmed this: R175T displayed robust apoptosis after N3a treatment, while R175S caused slower growth and increased p21 induction, consistent with cell-cycle arrest (Supplementary Fig. 4). The intermediate depletion kinetics of R175S reflect a separation-of-function phenotype, where the mutation compromises p53's apoptotic function more than its anti-proliferative activity. Since p53 protein levels heavily influence effector programs—higher levels often shifting the response from cell-cycle arrest to apoptosis[52]—accurately assessing these phenotypes requires physiologically controlled expression. CRISPR-based mutational scanning provides this control, allowing us to uncover mechanistic differences in variant function within their natural gene-regulatory context.

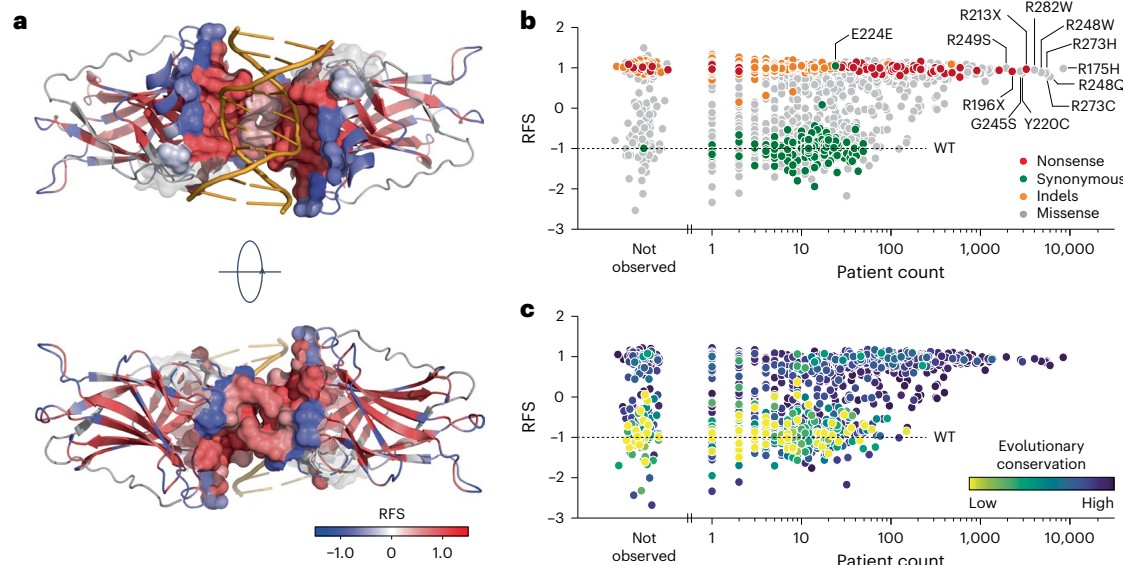

**Fig. 5 | RFS correlates with protein structure, mutation frequency and evolutionary conservation. a**, Structure of a DNA-bound p53 DBD dimer colored by RFS (PDB 3KZ8 (ref. 83)). The DBD-DNA and intra-dimer interaction interface within a distance of 10 Å is shown as a sphere model and superimposed on the cartoon model to highlight its sensitivity (red color, positive RFS values) to mutation. **b,c**, Scatter plots showing the correlation between RFS and aggregated variant count in patients with cancer listed in the UMD, IARC/NCI, TCGA and GENIE databases. Variants are colored by the indicated mutation types (**b**) or evolutionary conservation (**c**).

## Deep mutational scan of the p53 DBD

We extended our screen to a comprehensive library of 9,225 variants spanning the p53 DBD from exon 5 to 8 (amino acids 126 to 307), encompassing approximately 94.5% of all cancer-associated missense mutations (Fig. 4a and Supplementary Table 2). The library included all single-nucleotide substitutions (the most common *TP53* mutation type), as well as additional missense, nonsense and synonymous variants requiring two- or three-nucleotide changes, single-nucleotide insertions and 1–3-base pair (bp) deletions.

To overcome sequencing limitations, we divided the library into four sub-libraries, each covering a single exon with flanking intronic sequences, following previously published methods[27,53]. We co-transfected HCT116 LSL/Δ cells with the *TP53*-targeting Cas9 and each sub-library, followed by selection and Cre-induced activation of mutant expression. After treating cells with N3a or dimethyl sulfoxide (DMSO, as solvent control) for 8 d, we extracted genomic DNA, amplified the edited exon by PCR and analyzed variant frequencies by next-generation sequencing (NGS) (Extended Data Fig. 1a and Supplementary Table 2). We maintained coverage of at least 500 individually edited cells per variant, across three biological replicates. Control mutations, including nonsense (LOF) and synonymous (WT-like) variants, showed no notable abundance differences, confirming efficient donor library introduction without *TP53*-related bias (Fig. 4b and Extended Data Fig. 5a,b).

Following N3a treatment, the variant distribution shifted substantially, indicating functional differences. The correlation between the donor plasmid and cell libraries was strong for the control treatment but weakened markedly after N3a treatment (Fig. 4c and Extended Data Fig. 5a). Synonymous variants were depleted, while nonsense and frameshift mutations were enriched, creating a bimodal distribution that effectively separated LOF from WT-like variants (Extended Data Fig. 5b).

We standardized results across exons by converting enrichment scores (ESs) into relative fitness scores (RFSs), ranging from −1 (synonymous mutations) to +1 (nonsense mutations) (Fig. 4d and Extended Data Fig. 5c,d)[27]. Frameshift variants exhibited uniformly positive RFS values, similar to nonsense controls. In-frame deletions of three consecutive base pairs also yielded high RFS values, highlighting the

sensitivity of the p53 DBD to even single amino acid deletions. Substitution variants showed more variable effects, with transversions generally having a stronger impact than transitions. Overall, 55.2% of substitution variants displayed positive RFS values, indicating at least partial functional impairment of p53. Conversely, most intronic variants had negative RFS values, indicating preserved tumor suppressor activity.

We systematically replaced each residue with every possible amino acid to assess missense mutations (Fig. 4e). The screen returned reliable RFS values for 99% of the possible 3,458 missense variants, making it one of the most comprehensive studies of DBD variants to date (Supplementary Fig. 5). Missing variants mostly mapped to exon boundary-spanning codons (for example, G187, S261, A307) that were excluded from the library design since they could not be generated within a single exon. Hierarchical clustering by RFS values differentiated codons according to their vulnerability (Extended Data Fig. 5e,f). Hotspots such as G245, R248 and R249 were highly vulnerable to any substitution, while others such as R175 and R282 showed variable impairment depending on the amino acid change. Substitutions with biochemically similar amino acids clustered together based on functional effects, confirming that mutations with similar biochemical properties tend to cause less damage.

Mapping the median RFS values onto the three-dimensional protein structure revealed a significant correlation between higher RFS values and proximity to the DNA-binding surface (Fig. 5a and Extended Data Fig. 5g). Residues critical for stabilizing the hydrophobic core also showed high RFS values, while solvent-exposed residues were more tolerant to mutations (Extended Data Fig. 5g–i). Notably, residues involved in DNA contact (for example, R248) and those at the inter-dimer interface (for example, G199) were highly sensitive to mutations.

We compared our RFS values with the prevalence of over 150,000 *TP53* mutations in major cancer databases (Supplementary Table 3 and Supplementary Note 5). The most frequent hotspot mutations, nonsense and indel mutations, as well as other missense mutations with a patient count above 100, exhibited high RFS values, suggesting strong positive selection during tumorigenesis (Fig. 5b and Extended Data Fig. 6a). In contrast, missense variants with WT-like RFS values showed lower patient counts, resembling synonymous mutations and benign polymorphisms[54], likely representing passenger mutations. The

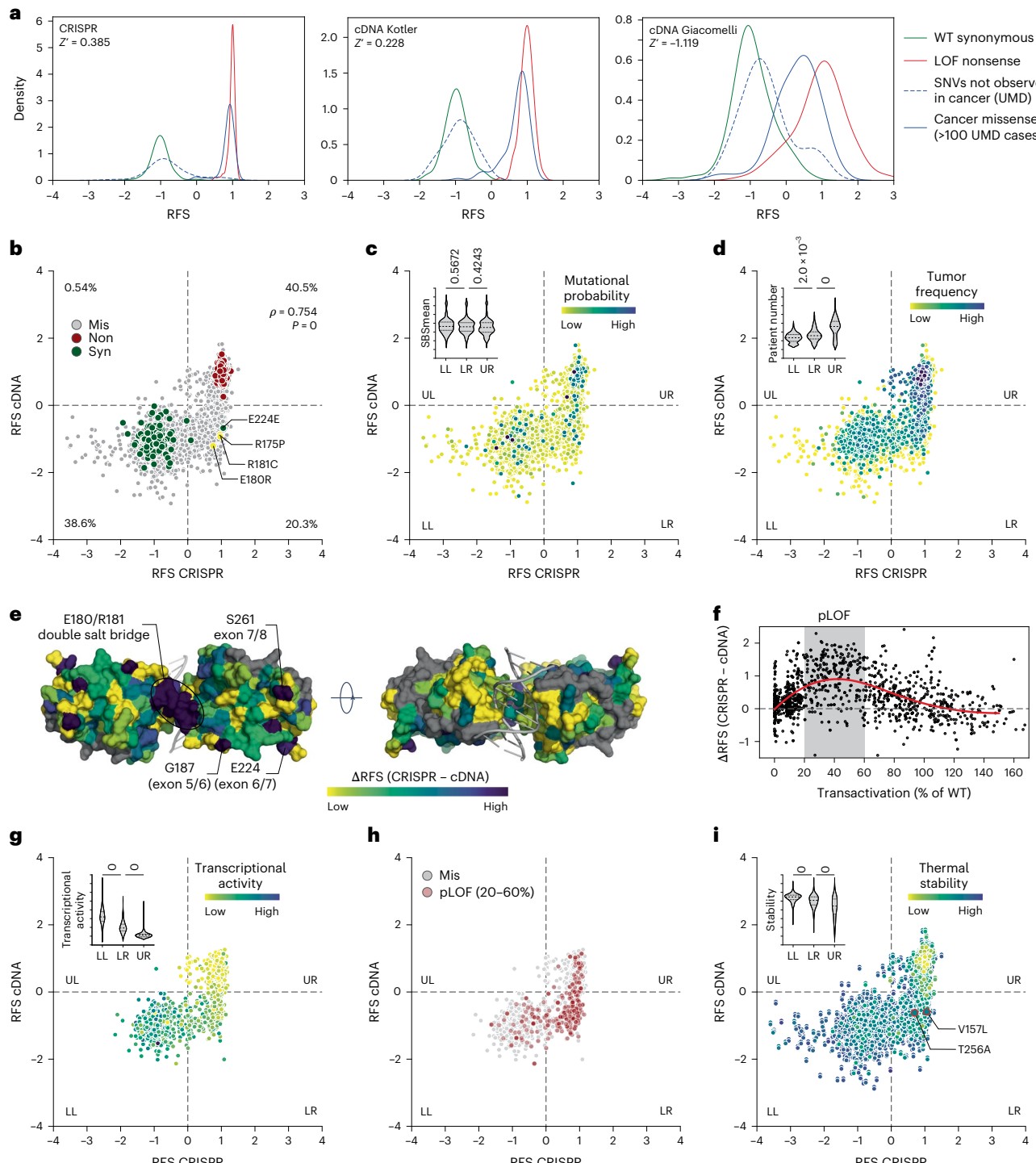

**Fig. 6 | CRISPR screen reveals pLOF variants. a,** Kernel density estimation plots showing the distribution of RFS scores for the indicated groups of variants in the CRISPR versus cDNA-based variant screens[18,20]. All results from previously reported cDNA screens were transformed to RFS by scaling the median of nonsense mutations to +1 and the median of synonymous mutations to −1. *Z*′ factors, a measure of statistical effect size, are stated as a quality parameter for the assay's ability to separate positive (LOF nonsense) and negative (synonymous) controls. **b**–**d,** Scatter plots illustrating correlation between RFS values obtained by CRISPR mutagenesis and cDNA overexpression[20]. Variants are categorized into four quadrants (LL, lower left; LR, lower right; UL, upper left; UR, upper right). Percentage of variants in each quadrant is given in **b**. *ρ*, Spearman correlation coefficient with *P* value approximated using a two-tailed *t*-distribution. Variants are colored by mutation type (**b**), average mutational probability (SBSmean) (**c**) or frequency in patients with cancer (**d**). Inserted violin plots illustrate the value distribution in the three main quadrants. **e,** Structure of a DNA-bound p53 DBD dimer (PDB 2AHI (ref. 84)) colored by the difference in RFS between the CRISPR and cDNA screen. Selected areas of high discrepancy are labeled. **f,** Scatter plot of the difference between CRISPR and cDNA screen versus the mean transcriptional activity of variants relative to WT p53 as measured in a yeast-based reporter system[17]. The area of 20–60% transcriptional activity (pLOF) is shaded in gray; red line, cubic spline curve. **g**–**i,** Scatter plots showing the correlation between RFS values obtained by CRISPR mutagenesis and cDNA overexpression[17] (**g**), classification as pLOF (**h**) or thermal stability as predicted by HoTMuSiC[67] (**i**). V157L and T256A are highlighted in **i** with red outline and increased dot size. Inserted violin plots illustrate the value distribution in the three main quadrants. All violin plots show *P* values from one-way ANOVA and a post hoc multiple comparisons test by Tukey.

strong correlation between codon-level RFS values and evolutionary conservation scores further confirmed that residues with high RFS values are under strong evolutionary selection (Fig. 5c, Extended Data Fig. 5j and Supplementary Table 4).

We observed a large number of high RFS missense mutations at evolutionarily conserved residues that were rarely or never reported in patients (Fig. 5b,c). Many of these variants were two- or three-nucleotide substitutions or single-nucleotide transversions, which are all less frequent in cancer cells compared with transitions (Extended Data Fig. 6a–h)[55]. When comparing variants with similar mutational probabilities based on COSMIC mutational signatures[56] (Extended Data Fig. 6i–l and Supplementary Table 5), those with a positive RFS consistently had significantly higher patient counts (Extended Data Fig. 6k,l). Thus, a positive RFS robustly identifies LOF variants under positive selection during tumor development.

We further assessed the ability of the RFS to classify variant pathogenicity using 1,256 ClinVar variants (≥1* review status, Supplementary Table 6)[57]. The RFS not only effectively distinguished nonsense from synonymous variants (Extended Data Fig. 5d), but also pathogenic from benign variants, achieving a precision–recall curve with an area under the curve of 0.999, an F1 score of 0.990, a precision/positive predictive value of 0.988 and a recall/sensitivity of 0.993 (Extended Data Fig. 7). Using ClinVar ≥1* variants as truth sets of pathogenic and benign variants[58,59], the RFS accurately classified >99% (398 of 401) of pathogenic/likely pathogenic controls as functionally abnormal and >98% (248 of 253) of benign/likely benign controls as functionally normal. The corresponding odds of pathogenicity (OddsPath) values were 50.2 and 0.0076, respectively, providing strong evidence for pathogenic (PS3) and benign (BS3) variant assessments according to the American College of Medical Genetics and Genomics (ACMG) and the Association for Molecular Pathology (AMP) guidelines[58,60,61]. This strength of evidence was consistent even with higher stringency thresholds, including ClinVar variants with two or more stars (Extended Data Fig. 7 and Supplementary Table 6).

### Increased sensitivity of CRISPR-based deep mutational scan for subtle LOF

We compared our CRISPR-based deep mutational scan with previous studies using lentiviral overexpression of mutant cDNA libraries[18,20], converting all data to RFS values (Fig. 6a and Supplementary Table 7). The CRISPR screen provided better separation between positive and negative controls, clearly distinguishing cancer-associated missense mutations from single-nucleotide variants (SNVs) not linked to cancer. In contrast, the cDNA screens showed substantial overlap between these groups, likely due to variable mutant expression from random genome integration of lentiviral constructs.

When comparing the CRISPR results with the cDNA-based study in ref. 20, both screens classified most variants similarly, but 20.3% of missense variants were differentially classified as LOF by CRISPR and WT-like by cDNA screening (Fig. 6b, lower-right quadrant). These lower-right variants had similar mutational probabilities but showed significantly higher patient counts than WT-like variants, suggesting positive selection during tumorigenesis (Fig. 6c,d and Extended Data Fig. 8).

Several lower-right variants, such as R175P, R181C and E180R, are tumorigenic in mice with a (partial) LOF phenotype[62–64]. Moreover, a notable region of discordant RFS values between the CRISPR and cDNA screen mapped to the intra-dimer interface, where mutations often cause pLOF effects[65] (Fig. 6e). To further validate CRISPR's sensitivity for detecting subtle LOF phenotypes, we compared the CRISPR RFS values with transcriptional activity from the yeast reporter assay in ref. 17, which is a gold standard for assessing the clinical impact of TP53 variants[66]. A moderate but highly significant negative correlation confirmed that positive RFS values are associated with low transcriptional activity (Supplementary Fig. 6). Variants with residual 20–60% of WT transcriptional activity showed the largest differences between CRISPR

and cDNA RFS values (Fig. 6f–h and Supplementary Table 7), further confirming the superior sensitivity of the CRISPR screen. All these observations were confirmed in a comparison with the cDNA screen in ref. 18 (Supplementary Fig. 7).

An analysis of protein stability estimates by HoTMuSiC[67] demonstrated that lower-right variants had higher thermal stability than upper-right quadrant LOF variants but lower stability compared with WT-like variants (Fig. 6i), indicating moderate destabilization that may impair function not as severely and irreversibly as in complete LOF variants. Two lower-right cancer variants, V157L and T256A, showed reduced thermostability in differential scanning fluorimetry assays but were less destabilized than other more frequent mutations (Supplementary Table 8). When introduced into HCT116 LSL/Δ cells by CRISPR-HDR, both mutations rendered cells resistant to N3a, similar to R175H and R175X (Supplementary Fig. 8). However, at 32 °C, responsiveness to N3a was restored, indicating moderate p53 destabilization. In addition, both variants were stabilized by arsenic trioxide, which allosterically reactivates several temperature-sensitive structural mutants[68,69] (Supplementary Table 8).

These findings highlight that even a subtle loss of p53 function from mild thermodynamic destabilization can clearly enhance proliferative fitness. This effect, missed by conventional cDNA expression screens, was correctly detected by the CRISPR screen, uncovering a set of dysfunctional missense variants with moderate destabilization and potential for pharmacological rescue.

### Widespread splicing alterations and NMD

DMS studies using cDNA overexpression are blind to mutation effects on RNA splicing, which can result in LOF through NMD. In our CRISPR-based screen, 55 of 56 previously reported splice-altering TP53 variants were enriched under N3a treatment, displaying positive RFS values indicative of LOF (Supplementary Table 9). Moreover, the most pronounced differences between the CRISPR and cDNA screens mapped to poorly conserved residues near exon boundaries (for example, G187, E224, V225 and S261), suggesting splicing disruption (Figs. 6e and 7a).

We sequenced cDNA from the cell libraries and correlated the abundance of variants at the cDNA level with their corresponding abundance in the genome (Fig. 7b, Supplementary Table 10 and Supplementary Note 6). Variants causing frameshift mutations in exons 5–8 led to premature termination codons, triggering NMD. Nonsense and frameshift mutations were significantly underrepresented at the mRNA level by ~30-fold (Fig. 7c–e). Additionally, several missense mutations near exon–intron junctions showed reduced mRNA levels and LOF, indicating splicing defects (Extended Data Fig. 9a,b). While many are rare double- or triple-nucleotide substitutions, some of these mutations, such as at codons G187, E224 and S261, are prevalent in cancer but had been classified as WT-like in all cDNA screens[17,18,20]. In our CRISPR screen, they were identified as LOF due to splicing defects (Extended Data Fig. 9c,d). To validate this, we introduced 'E224D' (NC_000017.11:g.7674859C>G) and 'E224=' (NC_000017.11:g.7674859C>T) into HCT116 LSL/Δ cells. Both mutations altered splicing, causing frameshift and premature termination, subjecting the mRNA to NMD and preventing p53 protein production (Fig. 7f–h), thereby rendering the cells resistant to N3a (Fig. 7i).

We also observed LOF variants in noncoding, exon-flanking intronic regions likely due to altered splicing. All mutations affecting the invariant GT and AG dinucleotides at intron ends resulted in LOF (Supplementary Fig. 9). SNVs at position 5 of intron 5 also had a deleterious impact, while similar substitutions in introns 6–8 were tolerated. The NC_000017.11:g.7673847A>C mutation in the 3′ region of intron 7, reported in a patient with pancreatic adenocarcinoma[70], caused aberrant splicing, leading to an in-frame insertion of three amino acids (Extended Data Fig. 9e,f). Unlike cDNA-based screens, the CRISPR screen therefore accurately discriminated functionally normal from abnormal variants in these intronic regions.

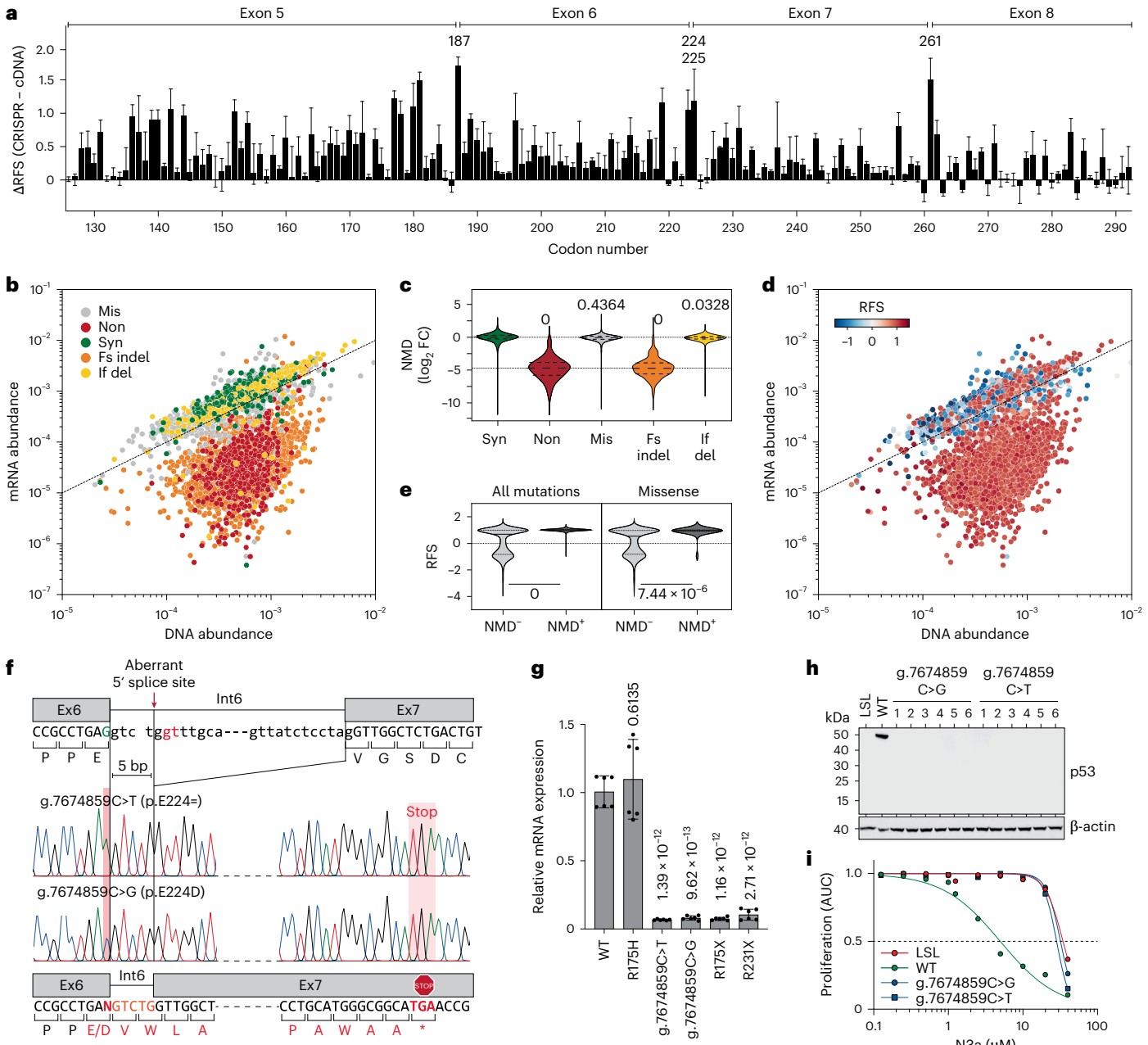

**Fig. 7 | Splicing and NMD. a**, Bar plot demonstrating large differences between CRISPR and cDNA screening results at exon borders (residues G187, E224, V225 and S261). Shown is the mean difference (±s.d.) of all missense variants at each codon. **b,d**, Scatter plots comparing the abundance of variants in the cell libraries at the level of genomic DNA and mRNA. Each dot represents the median abundance of a variant from $n = 3$ biological replicates. Variants are colored by mutation type (**b**) and by RFS (**d**). Dashed line, line of identity. **c**, Violin plot showing NMD as the $\log_2$ fold change in abundance at mRNA and DNA level by mutation type. One-way ANOVA with multiple comparison by Tukey. **e**, Distribution of RFS values in variants (all or missense) according to NMD status. Variants with a $\log_2$ fold change in abundance between mRNA and DNA <−2 were classified as NMD⁺. Two-sided Mann–Whitney test. **f**–**i**, LOF and NMD caused by g.7674859C>T (p.E224=) and g.7674859C>G (p.E224D) variants. **f**, Aberrant mRNA splicing revealed by Sanger sequencing of cDNA. **g**, Quantitative PCR with reverse transcription (RT–PCR) of indicated HCT116 mut/Δ cells. Shown is the *TP53* mRNA expression relative to WT as mean ± s.d. ($n = 6$ replicates). One-way ANOVA with Dunnett's multiple comparisons test. **h**, Western blot demonstrating lack of p53 protein expression in multiple HCT116 cell clones with g.7674859C>T/G variants. **i**, Resistance of g.7674859C>T/G clones to N3a. Proliferation was analyzed by real-time live-cell imaging. Shown is the area under the proliferation curve relative to untreated. p53-null (LSL, red) and WT (green) are shown as reference.

We also noted reduced mRNA levels for NC_000017.11:g.7675202 A>T, encoding the missense variant L137Q, and for NC_000017.11:g.767 4934T>A, encoding the synonymous variant G199=, suggesting splicing defects (Extended Data Fig. 9). While two other silent substitutions at the same position, NC_000017.11:g.7674934T>G/C, showed normal mRNA levels, g.7674934T>A and g.7675202A>T created cryptic splice sites, leading to aberrant transcripts. In HCT116 and H460 cells, both variants

lacked an anti-proliferative response to N3a and failed to induce p21 (Fig. 8a–d and Extended Data Fig. 10a–d). Sequencing revealed exon skipping and truncated transcripts (Fig. 8e–i and Extended Data Fig. 10e–i). The g.7674934T>A variant produced transcripts with premature termination codons, preventing p53 protein expression, while g.7675202A>T generated a shortened p53 protein with an in-frame deletion of amino acids 126−−137, despite being been classified as WT-like in cDNA screens.

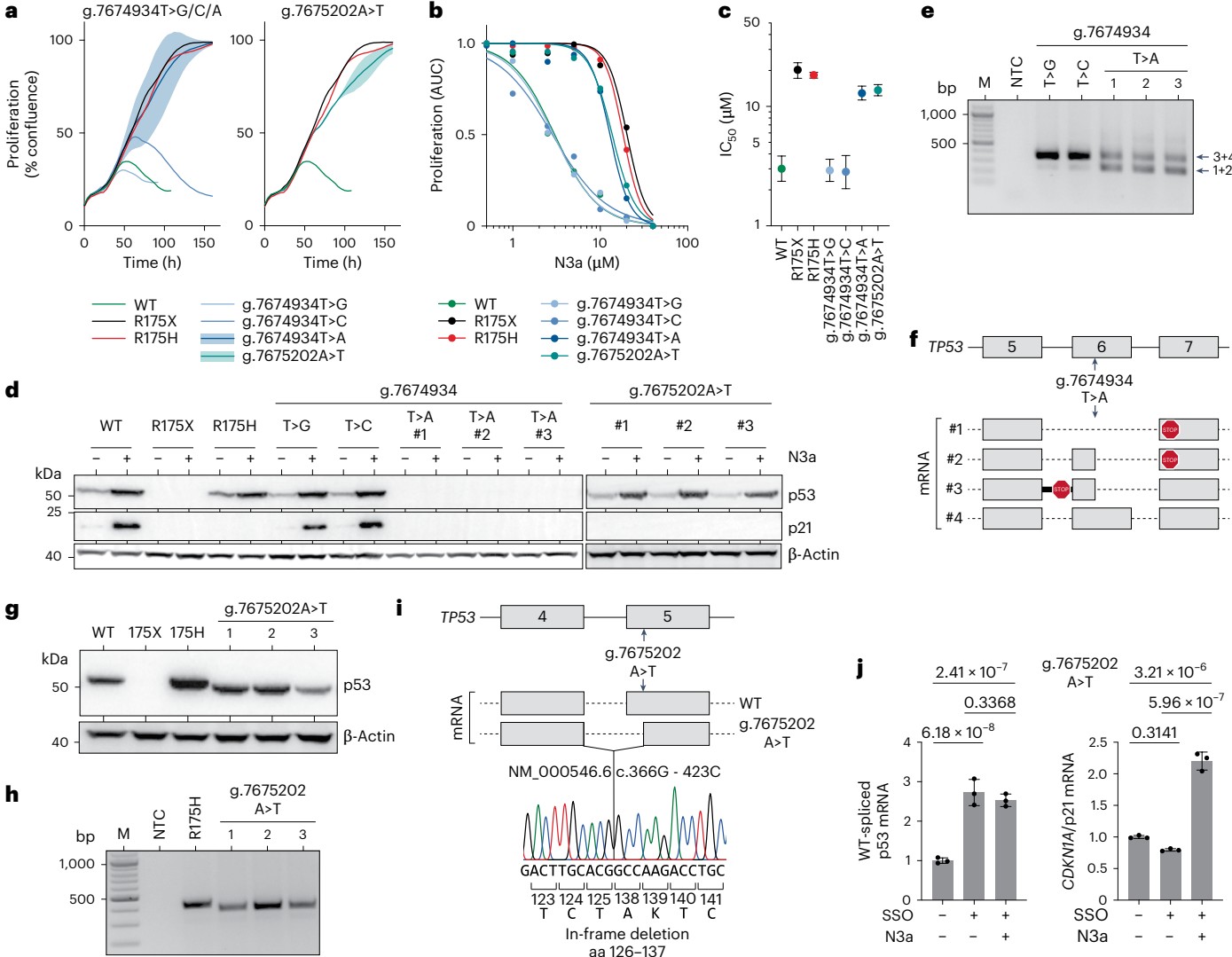

**Fig. 8 | Aberrant splicing due to exonic SNVs causes LOF. a–c**, Impact of codon 199 (NC_000017.11:g.7674934T>A/C/G) and codon 137 (NC_000017.11:g.7675202A>T) variants on the anti-proliferative activity of N3a in HCT116 cells. WT, missense (R175H) and nonsense (R175X) variants are shown for comparison. **a**, Proliferation in the presence of 10 μM N3a analyzed by real-time live-cell imaging. For the g.7674934T>A and g.7675202A>T genotypes, plots show the mean ± s.d. of $n = 3$ independent clones. **b**, Dose–response curves. Shown is the area under the proliferation curve (AUC) relative to untreated. **c**, $IC_{50}$ (with 95% CI). **d**, Western blot demonstrating mutant p53 and p21 protein expression in independent HCT116 clones in the absence and presence of N3a. **e,f**, cDNA analysis of g.7674934T>A/C/G clones. **e**, Agarose gel electrophoresis of RT–PCR products.

**f**, Scheme of mRNA transcripts detected by Sanger sequencing of RT–PCR amplicons. **g**, Western blot demonstrating reduced size of p53 protein in HCT116 clones with the g.7675202A>T genotype. **h,i**, cDNA analysis of g.7675202A>T clones. **h**, Agarose gel electrophoresis of RT–PCR products. **i**, Sequencing analysis of RT–PCR amplicons showing an in-frame deletion of 12 amino acids. **j**, Quantitative RT–PCR specific for the regularly spliced p53 and CDKN1A/p21 mRNA in HCT116 g.7675202A>T cells transfected with SSO and treated with N3a as indicated. Shown is the mRNA expression relative to untreated as mean ± s.d. ($n = 3$ replicates); two-way ANOVA with Tukey's multiple comparisons test. M, DNA size marker; NTC, no template control.

To explore the potential for correcting such splice defects, we used SSOs[71] designed to block the cryptic 3′ splice site in exon 5 created by the g.7675202A>T variant (Fig. 8j and Extended Data Fig. 10j). SSO transfection significantly increased the levels of the regularly spliced p53 mRNA and promoted p21 induction by N3a. This confirms that the LOF of the g.7675202A>T variant arises from aberrant splicing, not from a non-functional L137Q protein, and demonstrates proof-of-principle that cancer-associated p53 splice aberrations can be corrected using SSO technology. However, g.7675202A>T and g.7674934T>A were the only SNVs outside exon/intron borders to cause more than twofold mRNA reduction and LOF (Extended Data Fig. 9a), despite 355 other missense or synonymous SNVs creating cryptic splice sites. Thus, splice aberrations caused by exonic SNVs are less common than anticipated.

## Discussion

This study presents a comprehensive DMS of *TP53* using SGE by CRISPR-HDR, covering 94.5% of all cancer-associated *TP53* mutations. Our approach markedly outperforms previous multiplexed assays of variant effects studies based on cDNA overexpression[17,18,20], which struggled to clearly distinguish between nonsense and synonymous variants, as well as pathogenic and benign variants[58,59,61,72]. By introducing mutations at the endogenous *TP53* locus, we ensured physiological protein expression and highly reproducible results. This led to predictive values, sensitivity and specificity that surpassed cDNA-based classifiers and met strong PS3 and BS3 evidence levels in ACMG/AMP guidelines[59].

A key finding was that approximately 20% of the missense variants, previously classified as benign, were identified as LOF. These variants share a similar mutational probability with WT-like variants but occur

more frequently in tumors, suggesting they are positively selected during tumorigenesis. This aligns well with reports of tumorigenicity in mouse models for several of these variants[62–64], indicating that the deleterious impact of many *TP53* variants has been underestimated in earlier studies, likely due to nonphysiological expression levels from mutant cDNA overexpression.

Interestingly, many of the identified LOF variants were thermally destabilized by only a few degrees, markedly less than more frequent structural hotspot mutants such as Y220C[73,74]. This mild destabilization likely accounts for their residual transcriptional activity and lower frequency in patients with cancer. However, the temperature-sensitive phenotypes of these variants suggest that their folded, active conformation may be more easily restored by therapeutic interventions[75,76], such as targeted treatments with arsenic trioxide or antiparasitic antimonials[69,77]. Additionally, approaches such as hypothermia could provide further therapeutic benefit for patients harboring these mutations[78].

In addition to these findings, our study uncovered multiple splice-altering mutations, many of which had been previously overlooked. Large-scale RNA-seq studies of cancer samples have reported several examples of splice alterations within *TP53* (refs. 79–82), and 55 of the 56 reported splice-altering variants were also detected in our CRISPR-based screen. Two exonic variants, g.7675202A>T (L137Q) and g.7674934T>A (G199=), classified as benign by cDNA-based screens[17,18,20], were shown to disrupt normal splicing in our approach, leading to aberrant transcripts and LOF, promoting tumor cell fitness. These results underscore the importance of studying variant effects in a native genomic context. By using SSOs[71], we successfully masked the cryptic splice site created by the g.7675202A>T (L137Q) variant, restoring proper splicing and p53 function, and demonstrating the potential for therapeutic correction of splicing defects.

In summary, this DMS of *TP53* using CRISPR-HDR provides a comprehensive functional annotation of *TP53* variants, identifying even subtle LOF variants that were previously missed by cDNA-based screens. It also highlights temperature-sensitive variants amenable to pharmacological rescue and splice-altering variants that can potentially be corrected with SSOs. Importantly, we found no fitness advantage for missense over null mutations, reinforcing that GOF effects require secondary alterations. This study strongly enhances the translational value of *TP53* mutation databases, improving clinical variant interpretation for genetic counseling and personalized cancer therapy.

## Online content

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

## Methods

### Ethics and consent

This study complies with all relevant ethical regulations. Mouse experiments were performed in accordance with the German Animal Welfare Law (TierSchG) and received approval from the animal welfare committee of the local authority (Regierungspräsidium Gießen).

### Design of *TP53* variant library

The WT *TP53* sequence was derived from the human Ensembl genome, revision 96 (GRCh38). As the cloning procedure uses BbsI-mediated Golden Gate Cloning, BbsI-recognition sites present within *TP53* exon 4 to intron 9 were silently mutated and the resulting sequence was used as a template for library generation. Using transcript ENST00000269305 (RefSeq NM_000546), the sequences of exons 5, 6, 7 and 8, including 12 nucleotides of flanking intron sequence, were selected and subjected to in silico mutagenesis. Thirteen nucleotides of the introns flanking this 'mutatable' sequence were added so that this variable region was framed by short constant regions that would remain the same to all resulting synthetic oligonucleotides. To generate an exhaustive set of 'mutated' oligonucleotides that deviate from the WT sequence by a single mutation, the variable region was altered in the following way. Initially, each nucleotide was substituted with every other nucleotide, resulting in a comprehensive set of all SNVs. To include amino acid substitutions that cannot be achieved by a single-nucleotide substitution, we added double-nucleotide variants and triple-nucleotide variants to generate each possible amino acid substitution and nonsense mutation. In the case of multiple possible codon exchanges, we prioritized the change with the smallest hamming distance to the reference. To account for insertions, each possible single nucleotide was inserted at every position of the variable region, including the intronic region, resulting in a set of all possible insertions of size 1 bp. Finally, a deletion set was generated by deleting up to three nucleotides at every position of the variable region, thus creating a set of all possible deletions of sizes one to three.

### Generation of CRISPR-HDR donor vectors

Homology arm 1, ranging from exon 4 to intron 4 (chr17:7,675,788–7,676,168) of the *TP53* gene, was PCR-amplified from genomic DNA of HCT116 cells using the primers HA1_BsrGI_fw and HA1_BsaI_rev and cloned into the multiple cloning site 1 (MCS1) of the vector MCS1-EF1α-GFP-T2A-Puro-pA-MCS2-PGK-hsvTK (cat. no. HR700, System Biosciences) using BsrGI (New England Biolabs, R3575) and BsaI (New England Biolabs, R3733). Homology arm 2 ranged from intron 4 to intron 6 (for mutagenesis of exons 5 and 6, chr17:7,674,377–7,675,787) or from intron 4 to intron 9 (for mutagenesis of exons 7 and 8, chr17:7,673,145–7,675,787) with BbsI-recognition sites flanking the region to be mutated (R175: chr17:7,675,059 and 7,675,088; Ex5: chr17:7,675,036 and 7,675,254; Ex6: chr17:7,674,842 and 7,674,989; Ex7: chr17:7,674,164 and 7,674,308; Ex8: chr17:7,673,684 and 7,673,855) and was purchased as custom gene synthesis (GeneArt, Thermo Fisher) and cloned into MCS2 of HR700 using MluI (New England Biolabs, cat. no. R3198) and SalI (New England Biolabs, cat. no. R3138). In total, we generated five different HR700 donor vectors for cloning of libraries targeting R175, exons 5, 6, 7 and 8.

For generation of R175 plasmid libraries, complementary single-stranded oligonucleotides containing the desired mutations were purchased (Eurofins Genomics) and annealed individually to generate double-stranded DNA containing suitable overhangs. Double-stranded oligonucleotides were purified using a PCR purification kit (QIAGEN, cat. no. 28106) and cloned into HR700 vectors using BbsI-mediated Golden Gate Cloning.

For generation of exon-wide plasmid libraries, single-stranded oligonucleotide pools containing the desired mutations were purchased (oPools, Integrated DNA Technologies) and BbsI-recognition sites were introduced by PCR amplification ensuring a coverage of $1 \times 10^6$ for each mutation using the following primers: Exon5/6/7/8_BbsI_fw

and Exon5/6/7/8_BbsI_rev (Supplementary Table 11). Amplified oligos were purified using a PCR purification kit (QIAGEN, cat. no. 28106) and cloned into HR700 vectors using BbsI-mediated Golden Gate Cloning.

Plasmid libraries were transformed into MegaX DH10B T1R Electrocomp *E. coli* (Invitrogen, cat. no. C640003) and seeded on two (R175) or 30 (exon-wide libraries) 15-cm agar plates containing 50 μg ml⁻¹ kanamycin (Carl Roth, cat. no. C640003). After 16 h of growth at 37 °C, colonies were scraped off, pooled into 100 ml (R175) or 1.2 l of Lysogeny Broth (LB) medium and incubated for 4 h at 37 °C before extracting plasmid DNA using Nucleobond Xtra Midi kit (Macherey-Nagel, cat. no. C640003) according to the manufacturer's protocol.

Donor HDR plasmids for single mutations were generated using either annealed or PCR-amplified oligonucleotides as described above. Correctness and integrity of plasmids were validated using Sanger sequencing (LGC Genomics) or NGS. Plasmids for delivery of Cas9 and single guide RNAs (sgRNAs) were generated using BbsI-mediated Golden Gate cloning of annealed single-stranded oligonucleotides into pX330-U6-Chimeric_BB-CBh-hSpCas9 (pX330, gift from Feng Zhang, Addgene cat. no. 42230), pSpCas9(BB)-2A-Puro (PX459) V2.0 (pX459_puro, gift from Feng Zhang, Addgene cat. no. 62988), pSpCas9(BB)-2A-Hygro (pX459_hygro, gift from Ken-Ichi Takemaru, Addgene cat. no. 127763) or pSpCas9(BB)-2A-Blast (pX459_blast, gift from Ken-Ichi Takemaru, Addgene cat. no. 118055).

### Generation of screening cell lines

First, $2.5 \times 10^4$ HCT116 cells were transfected with 1.25 μg of pX330_sgIn5 (sgRNA 5'-TCA GTG AGG AAT CAG AGG CC-3') and 1.25 μg of HR700, which contained WT homology arms 1 and 2 flanking the LSL cassette, using Lipofectamine 2000 (Thermo Fisher Scientific, cat. no. 11668019) according to the manufacturer's protocol. Cells were selected with 1 μg ml⁻¹ puromycin (Invivogen, ant-pr) and single-cell clones were isolated. Single-cell clones were chosen based on N3a and puromycin resistance and analyzed by PCR for a Δ-allele with an inactivating deletion in intron 5 and a second allele containing the LSL cassette (HCT116 LSL/Δin5). The absence of the LSL cassette on the Δ-allele was confirmed using primers Intron4_fw and Exon7_rev. Sanger sequencing showed deletion of chr17:7,674,986–7,675,001. cDNA sequencing of the Δin5-allele revealed complete exclusion of exons 6 and 7. The aberrant joining of exons 5 and 8 resulted in a frameshift, creating an out-of-frame stop codon in exon 8. Premature stop codons in nonterminal exons trigger NMD, explaining the barely detectable p53 mRNA and protein levels expressed by the Δin5-allele. The presence of the LSL cassette was validated with two PCRs, one spanning the upstream end (Intron1_fw, GFP_rev) and the other spanning the downstream end (LoxP_fw, Exon7_rev). Finally, digital PCR for GFP (TaqMan Copy Number Assay, Applied Biosystems, cat. no. 4400291) was performed using QuantStudio 3D Digital PCR 20K Chip V2 (Applied Biosystems, cat. no. A26316) to confirm the presence of only a single copy of the LSL cassette in the genome. The respective single-cell clone of HCT116 LSL/Δin5 was then transfected with pX330_sgPuro (sgRNA 5'-CACGCCGGAGAGCGTCGAAG-3') to knock out the *pac* gene present in the LSL cassette. After validation of the puromycin sensitivity, HCT116 LSL/Δin5 cells were further transfected with pX459_hygro_sgIn7 (sgRNA 5'-CCACTCAGTTTTCTTTTCTC-3') to generate HCT116 LSL/Δin5+7 cells for mutagenesis of exons 7 and 8. After selection with 250 μg ml⁻¹ hygromycin (Invivogen, ant-hg), single-cell clones were screened via PCR (Δin5+7 allele: Intron4_fw, Exon8_fw; LSL allele: LoxP_fw, Exon8_fw). A respective single-cell clone of HCT116 LSL/Δin5+7 with distinguishable deletions (LSL allele: chr17:7,673,970-7,673,995; Δ-allele: chr17:7,673,986–7,674,259) on both alleles was chosen for further experiments.

H460 LSL/Δ/Δ cells were generated from NCI-H460 using the same procedure, with special attention given to the fact that this cell line has three *TP53* alleles, meaning it must contain two Δ alleles and one LSL allele.

## Generation and treatment of mutant cells and cell libraries

For generation of single mutants, $2.5 \times 10^4$ HCT116 LSL/ΔIn5, HCT116 LSL/ΔIn5+7 or H460 LSL/Δ/Δ cells were transfected with 1.25 µg of LSL allele-specific sgRNAs (pX459_blast_In5$^{LSL}$, sgRNA 5′-GTGAGGAATCAGAGGACCTG-3′ or pX459_blast_In7$^{LSL}$, sgRNA 5′-CTTTGGGACCTACCTGGAGC-3′) and 1.25 µg of the corresponding HR700 vector carrying the intended mutation using Lipofectamine 2000 according to the manufacturer's protocol. Transfected cells were selected with 20 µg ml$^{-1}$ blasticidin (Invivogen, ant-bl) for 3 d and 1 µg ml$^{-1}$ puromycin for 7 d, before single-cell clones were isolated and the presence of the mutation was validated through edit-specific PCR and Sanger sequencing. Finally, cells were infected with AV-Cre (ViraQuest, Ad-CMV-Cre, MOI20 for HCT116 cells, MOI250 for H460 cells) and expression of the mutant was confirmed via cDNA sequencing and western blot analysis.

For the generation of R175 libraries, $4 \times 10^6$ HCT116 LSL/ΔIn5 cells were transfected with 6.25 µg of pX459_blast_In5$^{LSL}$ and 6.25 µg of the HR700 vector library, respectively, using Lipofectamine 2000. For the generation of R175 libraries in H460 cells, $4 \times 10^6$ H460 LSL/Δ/Δ cells were transfected with 20 µg of pX459_blast_In5$^{LSL}$ and 20 µg of HR700 vector library using the Neon Transfection System (Thermo Fisher Scientific, cat. no. MPK10025). Transfected cells were selected for 3 d with 20 µg ml$^{-1}$ blasticidin and for 7 d with 1 µg ml$^{-1}$ puromycin. Then, $8 \times 10^6$ cells were infected with AV-Cre and, after 5 d, the cell library was divided and treated with 10 µM N3a (BOC Sciences, cat. no. B0084-425358), 75 nM RG7388 (MedChemExpress, cat. no. HY-15676), 10 µM RO-5963 (Calbiochem, cat. no. 444153), 1 µM MI-773 (Selleckchem, cat. no. S7649), 750 nM AMG 232 (MedChemExpress, cat. no. HY-12296) or the respective volume of DMSO (Carl Roth, cat. no. 4720) as solvent control for 8 d. For irradiation experiments, an X-RAD 320iX tube was used with settings of 320 kV voltage and a current of 8 mA, with a dose rate -1 Gy min$^{-1}$. Cells were further cultivated for 8 d after irradiation. 5-Fluorouracil (pharmacy of the Universitätsklinikum Gießen and Marburg) was administered at a concentration of 5 µM for 24 h or 48 h, and cells were further cultivated for 8 d after treatment. For mutant p53 reactivation studies, cell libraries were treated with either 12.5 µM or 25 µM APR-246 (Sigma-Aldrich, cat. no. SML1789) or 0.01 µM or 0.04 µM ZMC-1 (Abcam, NSC319726, cat. no. A24132) alone or in combination with 10 µM N3a or DMSO for a total of 8 d. Starvation experiments were performed to investigate the effect of nutrient deprivation on cell growth. Three different conditions were used to induce starvation: HBSS (Merck, cat. no. H8264) for 3 d, DMEM without glucose (Gibco, cat. no. 11966025) for 1 d and DMEM without glutamine (Gibco, cat. no. 11960044) for 7 d. DMEM without glucose and DMEM without glutamine were supplemented with 10% (v/v) dialyzed FBS (Gibco, cat. no. 26400044). Following starvation, cells were allowed to recover and expand for either 1 d in the case of the −glucose condition or 7 d in the case of the HBSS or −glutamine conditions.

For generating exon-wide mutant cell libraries, $5.4 \times 10^8$ HCT116 LSL/ΔIn5 or HCT116 LSL/ΔIn5+7 cells were transfected with 1.125 mg of pX459_blast_In5$^{LSL}$ or pX459_blast_In7$^{LSL}$, and 1.125 mg of the corresponding HR700 vector library, using Lipofectamine 2000. Transfected cells were selected with 20 µg ml$^{-1}$ blasticidin for 3 d and 1 µg ml$^{-1}$ puromycin for 7 d. Then, $1.2 \times 10^8$ cells were infected with AV-Cre and, after 5 d, the cell library was divided and treated with 10 µM N3a or DMSO for 8 d. Recombination was monitored through flow cytometry analysis of GFP expression. Genomic editing was performed a single time, with cells transfected once using each exon library and subsequently selected with blasticidin and puromycin. The entire functional assay, which included Cre transfection and the selection with either N3a or DMSO, was conducted in triplicate for each exon library. These triplicates were performed sequentially on different days, rather than in parallel.

## Genomic DNA analysis of mutant cells and libraries

Genomic DNA of mutant cells was isolated using the DNA Blood Mini Kit (QIAGEN, cat. no. 51106) following the manufacturer's protocol, and a nested PCR strategy was used to selectively amplify either edited or edited and Cre-recombined alleles (Extended Data Fig. 1a). The input amount of genomic DNA and number of PCR reactions were adjusted to achieve a minimum average coverage of 500 cells per variant. For first step PCR, the following primers were used before AV-Cre recombination: LoxP_fw, Intron9_rev; and after AV-Cre recombination: Intron4_fw, Intron9_rev. The PCR products were pooled, purified and diluted 1:1,000 for the second, editing-specific PCR step. Editing specificity was achieved by using primers binding to sequences that are created in intron 5 or intron 7 by homologous recombination with the HDR donor: Exon5/6/7/8_Edspec_fw, Exon5/6/7/8_Edspec_rev. The PCRs were performed with Q5 High-Fidelity DNA Polymerase (New England Biolabs, cat. no. M0491) following the manufacturer's protocol. PCR products were purified using a PCR purification kit according to the manufacturer's protocol. PCR amplicons were purified with AMPure XP beads (Beckman Coulter, cat. no. A63880) and sequencing libraries were prepared from 5 ng of the purified amplicon using the NEBNext Ultra DNA Library Prep Kit for Illumina (New England Biolabs, cat. no. E7370L) according to the manufacturer's protocol. The quality of sequencing libraries was validated with a Bioanalyzer 2100 using the Agilent High Sensitivity DNA Kit (Agilent, cat. no. 5067-4626). The pooled sequencing libraries were quantified and sequenced on either the MiSeq (v.2 or v.2 nano, 2 × 250 cycles, or v.3, 2 × 300 cycles, depending on library complexity) or the NovaSeq 6000 (SP flow cell 2 × 250 cycles) platform (Illumina).

Sequences were obtained via targeted paired-end sequencing. Sequenced reads were demultiplexed using mmdemultiplex (v.0.1). Overlapping paired-end reads were trimmed of adapter/primer sequences using CutAdapt[85] (v.3.5) and merged into a single sequence using NGmerge[86] (v.0.3), taking advantage of the overlapping reads to reduce sequencing errors. The occurrence of each synthetic sequence was counted from merged reads via exact matching[31], since the minimal hamming distance between synthetic sequences was 1. WT and nonmatching reads were discarded.

To calculate the relative frequencies (variant abundances), the read count was divided by the total number of matched reads. From this ratio, we obtained the ES as the $\log_2$ fold change of the variant abundance in treated versus control conditions. However, this ES is dependent on the relative amounts of WT-like and LOF variants in a cell population, which vary between different libraries. To obtain a score that is comparable across different libraries and screens, the ES was further normalized into an RFS by the following formula:

$$\text{RFS}_{\text{Ex}}(\text{ES}) = \left( \frac{\text{ES} - \bar{x}_{\text{ex}}^{\text{non}}}{\bar{x}_{\text{ex}}^{\text{non}} - \bar{x}_{\text{ex}}^{\text{syn}}} \right) \times 2 + 1$$

with $\bar{x}_{\text{ex}}^{\text{non}}$ denoting the median of the scores for all nonsense mutations in a specific exon (ex) and $\bar{x}_{\text{ex}}^{\text{syn}}$ denoting the median of all synonymous mutations in this exon. RFS scores were calculated for each replicate, then, as our total score, we obtained the median (RFS$_{\text{median}}$) over all three replicates.

In addition, we used the Enrich2 package (v.1.2.0)[87] as an orthogonal method to calculate scores from the raw variant counts. Specifically, we configured Enrich2 in count mode by inputting our variant counts as 'Identifiers Only' SeqLib, selected 'log ratios' as the scoring method based on the Enrich2 manual recommendations and used DMSO-treated samples as T0 and N3a-treated samples as T1, enabling a comparison between treated and untreated cells. For normalization, we applied the 'library size (full)' option. The standard error (SE_enrich2) was then used to calculate 95% confidence intervals. The resulting Enrich2 scores (score_enrich2), along with their confidence intervals, were transformed with the same method applied to calculate RFS values, which used the median of nonsense and synonymous variants. This yielded the transformed_score_enrich2 and transformed_SE_enrich2. To determine whether each variant's transformed_score_enrich2 was

significantly higher than the population of synonymous variants, we conducted a one-sided *z*-test for each variant under the null hypothesis that the variant's score is equal to or lower than the weighted mean of the synonymous variants. The resulting *P* values were adjusted for multiple hypothesis testing using the Benjamini–Hochberg correction.

## cDNA analysis of mutant cells and libraries

Total RNA was isolated using the RNeasy Mini Kit (QIAGEN, cat. no. 74106) and reverse transcribed using the SuperScript VILO cDNA Synthesis Kit (Invitrogen, cat. no. 11754250). PCR was performed with the primers: Exon2_fw, Exon11_rev. PCR products were purified and sequenced using Sanger sequencing. For sequencing of G199G cDNA variants, PCR products were cloned into pCR-Blunt II-TOPO Vector (Invitrogen, cat. no. 450245) according to the manufacturer's protocol. For cDNA sequencing of mutant cell libraries, RNA was reverse transcribed and amplified in five reactions with 1 µg of RNA template each using SuperScript IV One-Step RT–PCR System with ezDNase (Invitrogen, cat. no. 12595025) and the primers: Intron4_fw, Intron8_rev. The amplified PCR products were pooled, providing an estimated variant coverage of 50–250×, purified and diluted 1:1,000 for a second step of PCR, using Q5 High-Fidelity DNA Polymerase and exon-specific primers: Exon5/6/7/8_cDNA_fw, Exon5/6/7/8_cDNA_rev. Library preparation, sequencing and analysis followed the same protocol as for genomic DNA. Merged reads were trimmed to only include exonic regions.

## Data analysis and software

Pathogenicity classifications from ClinVar were intersected with the list of 9,225 variants in the CRISPR DBD library, yielding 1,367 unique variants present in both datasets that were further subgrouped by germline review status (≥1*, ≥2* or 3*/variant curation expert panel (VCEP)) and mutation type ('molecular consequence' all or missense) (Supplementary Table 6). We visualized the distribution of RFS values in these subgroups for the ClinVar germline pathogenicity classes, and calculated precision–recall curves, receiver operating characteristic curves and OddsPath[58,59].

To analyze distance relationships within the p53 DBD, we generated a contact map for Protein Data Bank (PDB) 2AHI using ProteinTools (https://proteintools.uni-bayreuth.de/)[88]. The map represents the distances between all amino acid pairs in a matrix form. The distance from the DNA-binding surface (TOP) was defined as the mean distance from the residues 248, 273, 277 and 280; the distance from the opposite pole (BOTTOM) as the mean distance from residues 153, 225 and 260; and the distance from the core (CENTER) as the mean distance from residues 195, 236 and 253. HoTMuSiC[67] was used to predict thermal destabilization of variants and solvent accessibility of residues based on PDB entry 2AHI.

To compare CRISPR and cDNA-based variant screens, we used datasets from ref. 20 and ref. 18. Both cDNA datasets were transformed to RFS as defined above by scaling the median of nonsense variants to +1 and the median of synonymous variants to −1. Our analysis of yeast reporter data from ref. 17 used the mean transcriptional activity (in % of WT) across the eight different reporter constructs.

The plots and statistical analyses in this study were created using Microsoft Excel 2019 (v.2301), GraphPad Prism (v.9.4.1) or Python (v.3.9.12), with libraries: Matplotlib (v.3.5.1), Seaborn (v.0.11.2), SciPy (v.1.7.3) and Statsmodels (v.0.13.2). Graphics were assembled in Adobe Illustrator (v.26.5.2). Quantification of western blots was performed with ImageJ (v.1.54g).

## Cell culture

See Supplementary Method 1.

## RNA analysis

See Supplementary Methods 2–4.

## Protein analysis

See Supplementary Methods 5 and 6.

## Cellular phenotype analysis

See Supplementary Methods 7–9.

## Animal experiments

See Supplementary Method 10.

## Statistics and reproducibility

The results presented in the graphs represent the mean or median values obtained from *n* biological replicates, as indicated. The error bars in the figures indicate the standard deviation, unless stated otherwise. The difference between two sets of data was assessed through either a two-sided unpaired *t*-test or a Mann–Whitney test if the data were not normally distributed. To analyze multiple groups, a one-way analysis of variance (ANOVA) was used in combination with a multiple comparisons test. For three or more groups that had been divided into two independent variables (such as treatment and genotype), a two-way ANOVA was used in combination with a multiple comparisons test. The ANOVA results and selected pair-wise comparisons are reported in the figures and Source Data files. A *P* value less than 0.05 was considered statistically significant.

Since methionine and tryptophan are each encoded by only a single codon, synonymous variants for these amino acids were excluded from the variant library design. Additionally, due to the exon-wise generation of mutome data, mutations spanning exon boundaries were also excluded. Variants with a mean cDNA read count below 5 were excluded from the analysis for NMD. Animal group sizes were determined by performing a power analysis based on an anticipated effect size of 1.5, with a power of 80% and a significance level of 0.05 (one-sided *t*-test). However, no statistical method was applied to predetermine the sample size for the other experiments. The experiments were not randomized, and investigators were not blinded to group allocation or outcome assessment.

All blots or gel images show results representative of two independent experiments.

## Reporting summary

Further information on research design is available in the Nature Portfolio Reporting Summary linked to this article.

## Data availability

All data generated or analyzed during this study are included in this published article (and its Supplementary Information and Source Data files). All function scores are included in Supplementary Tables 1 and 2, including NGS read counts. Sequencing raw data were deposited at EMBL BioStudies (https://www.ebi.ac.uk/biostudies/), accession numbers: E-MTAB-12734 (bulk RNA-seq), E-MTAB-13904 (single-cell RNA-seq), E-MTAB-14322 (*TP53* R175 SGE experiments), E-MTAB-12857 (*TP53* exon 5–8 SGE genomic DNA sequencing) and E-MTAB-12861 (*TP53* exon 5–8 SGE cDNA sequencing). To evaluate the correlation between RFS value and variant frequency in cancer patient samples, we obtained *TP53* variant frequency data from: UMD TP53 Mutation Database (release 2017_R2, https://p53.fr/tp53-database)[89], NCI/IARC The TP53 Database[90] (release R20, July 2019, https://tp53.isb-cgc.org/), the 'curated set of nonredundant studies' from the TCGA and the AACR project GENIE[91] from cBioPortal[92] (http://www.cbioportal.org/, downloaded 20 December 2022). To evaluate the mutational probability of variants, we used Mutational Signatures (v.3.3, June 2022) downloaded from COSMIC (https://cancer.sanger.ac.uk/signatures/)[56]. We calculated a mean Single Base Substitution (SBSmean) signature by averaging signatures SBS1 to SBS21 weighted by their prevalence in cancer samples, as reported in ref. 93 (Supplementary Table 3). Pathogenicity classifications for 3,417 *TP53* variants were extracted from the ClinVar database with '*TP53*' as search term (https://www.ncbi.nlm.nih.gov/clinvar/, downloaded 27 July 2024). The evolutionary conservation profile for p53 was obtained from the ConSurf-Database

(https://consurf.tau.ac.il/consurf_index.php, downloaded 17 February 2022)[94]. RFS values were mapped onto the p53 DBD structure using PyMOL (v.2.5.2) with Protein Data Bank (PDB) entries 2AHI (ref. 84) (https://www.rcsb.org/structure/2AHI) and 3KZ8 (ref. 83) (https://www.rcsb.org/structure/3KZ8). For comparisons with previous TP53 DMS studies, we used enrichment data of p53 variants measured in the p53-null H1299 cell line (ref. 20, Supplementary Table 2, RFS_H1299), and enrichment results from the A549 p53-knockout cell line (ref. 18, A549_p53NULL_Nutlin-3_Z-score). Yeast reporter data for transcriptional activity of p53 variants from ref. 17 were downloaded from the NCI/IARC TP53 Database (release R20, July 2019, https://tp53.isb-cgc.org/). Source data are provided with this paper.

## Code availability

Code is available on GitHub (https://github.com/IMTMarburg/TP53_SGE) and via Zenodo at https://doi.org/10.5281/zenodo.13983866 (ref. 95).

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

## Acknowledgements

We thank S. Bischofsberger, A. Grzeschiczek, B. Geißert, A. Filmer and A. Schneider for technical and experimental support. We acknowledge support by the Medical Core Facilities for Genomics, Flow Cytometry and Irradiation. D.-I.B and A.C.J. are grateful for support by the Structural Genomics Consortium (SGC), a registered charity (no. 1097737) that received funds from Bayer AG, Boehringer Ingelheim, Bristol Myers Squibb, Genentech, Genome Canada through the Ontario Genomics Institute (grant no. OGI-196), the EU/EFPIA/OICR/McGill/KTH/Diamond Innovative Medicines Initiative 2 Joint Undertaking (EUbOPEN grant no. 875510), Janssen, Merck KGaA, Pfizer and Takeda. A.C.J. is funded by the Deutsche Forschungsgemeinschaft (DFG grant no. JO 1473/1-3) and the German Cancer Aid (TACTIC). T. Stiewe is funded by grants from BMBF (grant no. 031L0063), Deutsche Forschungsgemeinschaft (DFG grant nos. TRR81/3 109546710 A10, STI 182/13-1, STI 182/15-1, GRK2573), the German Center for Lung Research (DZL), State of Hesse (LOEWE iCANx), von Behring-Röntgen Stiftung (grant nos. 65-0004 and 66-LV06) and Deutsche José Carreras Leukämie Stiftung e.V. (grant no. 09R/2018). R.S. is supported by grants from the Institute for Lung Health (ILH) and Deutsche Forschungsgemeinschaft (DFG grant nos. SA 1923/7-1, CRC1213 A10N). The UMD database developed by T. Soussi is supported by grants from Hadassah France. The funding bodies were not involved in the design of the study; in the collection, analysis and interpretation of data; and in writing the manuscript.

## Author contributions

The study was conceptualized by T. Stiewe with support from J.S.F., M.K., R.S., T. Soussi and A.C.J. The wet-lab experiments were performed by J.S.F., M.K., D.D., O.P., P.H., A.B., M. Noeparast, E.P., M. Neumann, D.-I.B., K.K., N.M. and I.B. The experiments were supervised by M.W., S.E., A.C.J. and T. Stiewe. Next-generation sequencing was performed by J.T.-S., A.N., T.P. and M.B. The data were curated, analyzed and visualized by K.H., M.M., J.S.F., M.K., T. Soussi and T. Stiewe. Funding was acquired by R.S., A.C.J. and T. Stiewe. The original draft was written by J.S.F. and T. Stiewe, and all co-authors reviewed and edited the manuscript.

## Competing interests

The authors declare no competing interests.

## Additional information

**Extended data** is available for this paper at https://doi.org/10.1038/s41588-024-02039-4.

**Correspondence and requests for materials** should be addressed to Thorsten Stiewe.

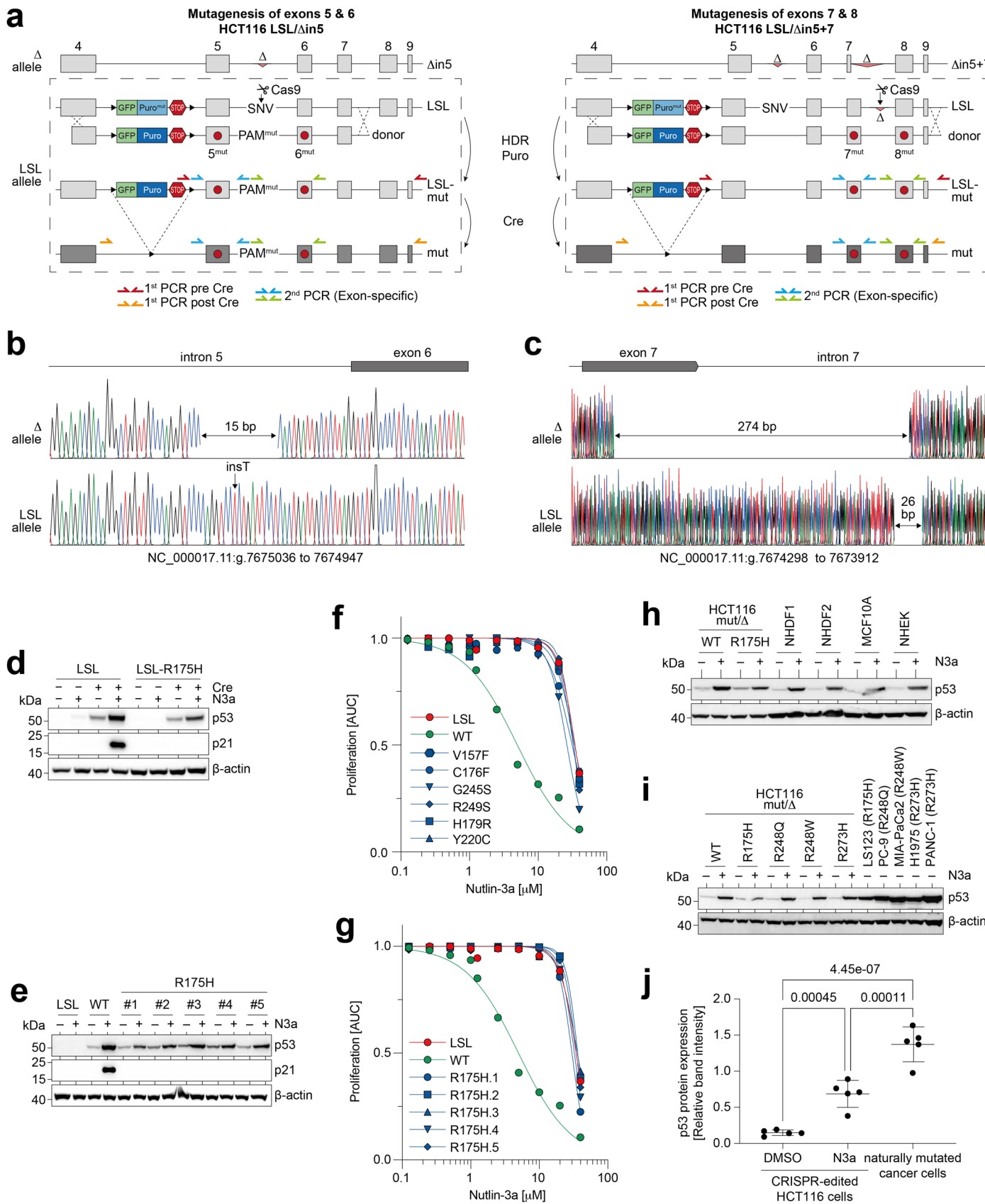

**Extended Data Fig. 1 | See next page for caption.**

**Extended Data Fig. 1 | Generation and functional characterization of single *TP53*-mutant HCT116 cell clones. a**, CRISPR/Cas9-targeting of *TP53* in HCT116 cells. Shown are the two *TP53* alleles and their modifications. The Δ allele contains inactivating intronic deletions (**b** and **c**). The second allele contains a loxP-flanked transcriptional stop (LSL) cassette, expressing GFP and a non-functional puromycin N-acetyltransferase (Puro^mut) resistance gene, and harbors an SNV for allele-specific Cas9-targeting. Donor vectors contain an intact puromycin resistance gene allowing selection of HDR-edited cells. To prevent re-cutting and enable selective amplification of edited alleles (LSL-mut), exon 5/6 donors contained a PAM-inactivating mutation. An intron-7 deletion on the LSL allele eliminated the need for an additional exon 7/8 donor mutation. Adenoviral Cre was used to excise the LSL-cassette and activate expression, yielding HCT116 mut/Δ cells. Selective NGS of the edited and Cre-recombined allele was ensured by nested PCR using the indicated primer pairs. **d** and **e**, Western blot of p53 and p21 expression in the indicated cell lines treated with Cre and N3a as indicated. **f** and **g**, Proliferation of *TP53*-mutant cell clones in the presence of N3a analyzed by real-time live-cell imaging. Shown is the area under the proliferation curve (AUC) relative to untreated. p53-null (LSL, red) and wild type (WT, green) are shown for reference. **h-j**, p53 protein expression in edited HCT116 and H460 cells, normal human diploid fibroblasts (NHDF, two donors), mammary epithelial cells (MCF10A), normal human epidermal keratinocytes (NHEK), and patient-derived p53-mutant cell lines. **j**, Quantification of p53 normalized to β-actin in (**i**). Mean±SD (n=5 cell lines per group); one-way ANOVA with Holm-Šídák's multiple comparisons test.

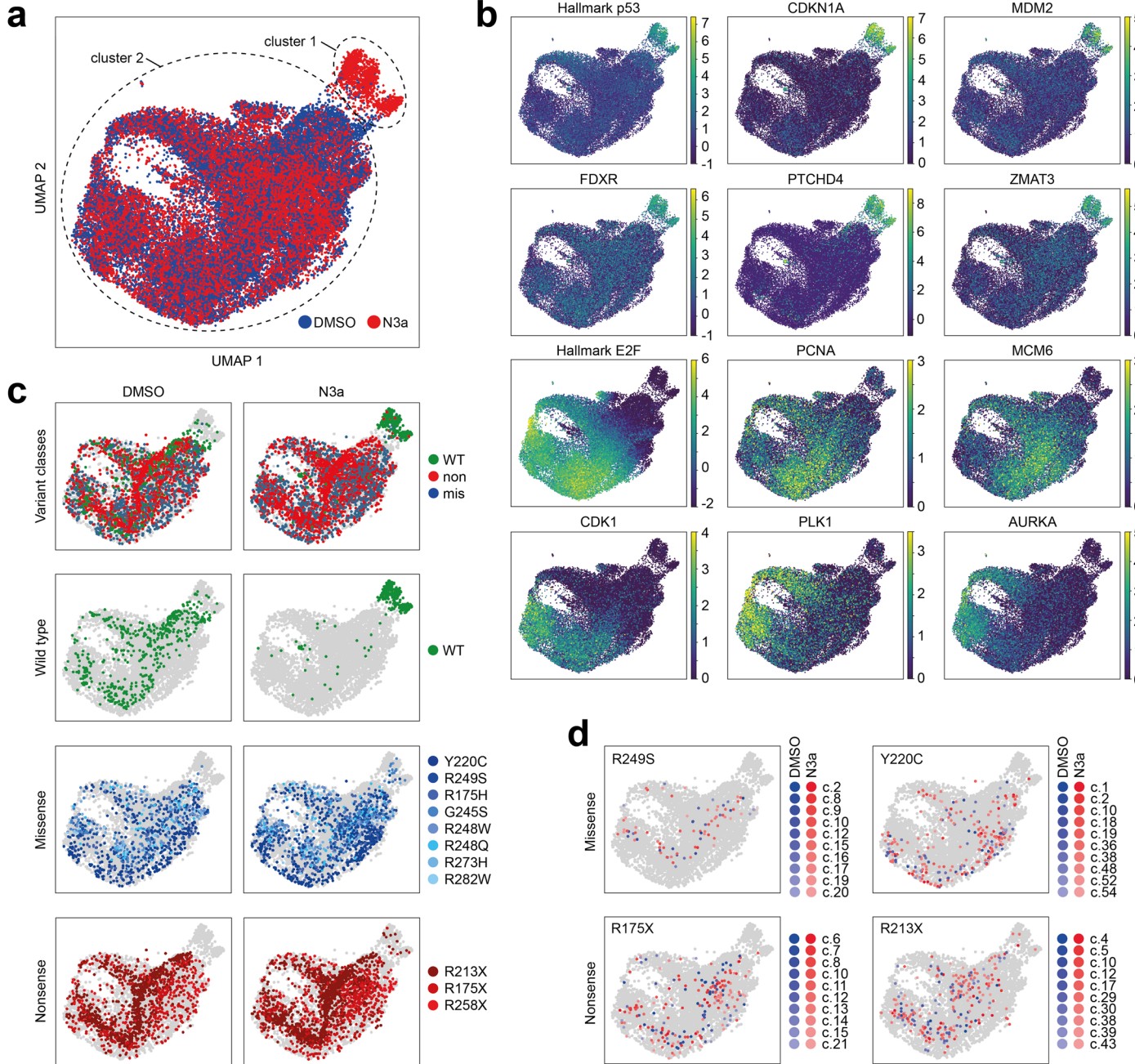

**Extended Data Fig. 2 | Clonal variance analysis by single-cell RNA sequencing.** Dimensionality reduction analysis (Uniform Manifold Approximation and Projection, UMAP) was used to visualize the overall distribution of a DMSO- and a second N3a-treated cell pool, each containing 12 different *TP53* variants, including 8 missense, 3 nonsense, and wild type (WT), with each variant represented by 10 independent single-cell clones. **a**, UMAP plot colored by treatment (cell pool) with two main cell clusters highlighted. **b**, Expression of p53-related genes/signatures (top) and cell cycle-related genes/signatures (bottom). **c**, Cells colored by variant class or variant genotype. **d**, Cells with indicated variant genotypes colored by clone ID and treatment.

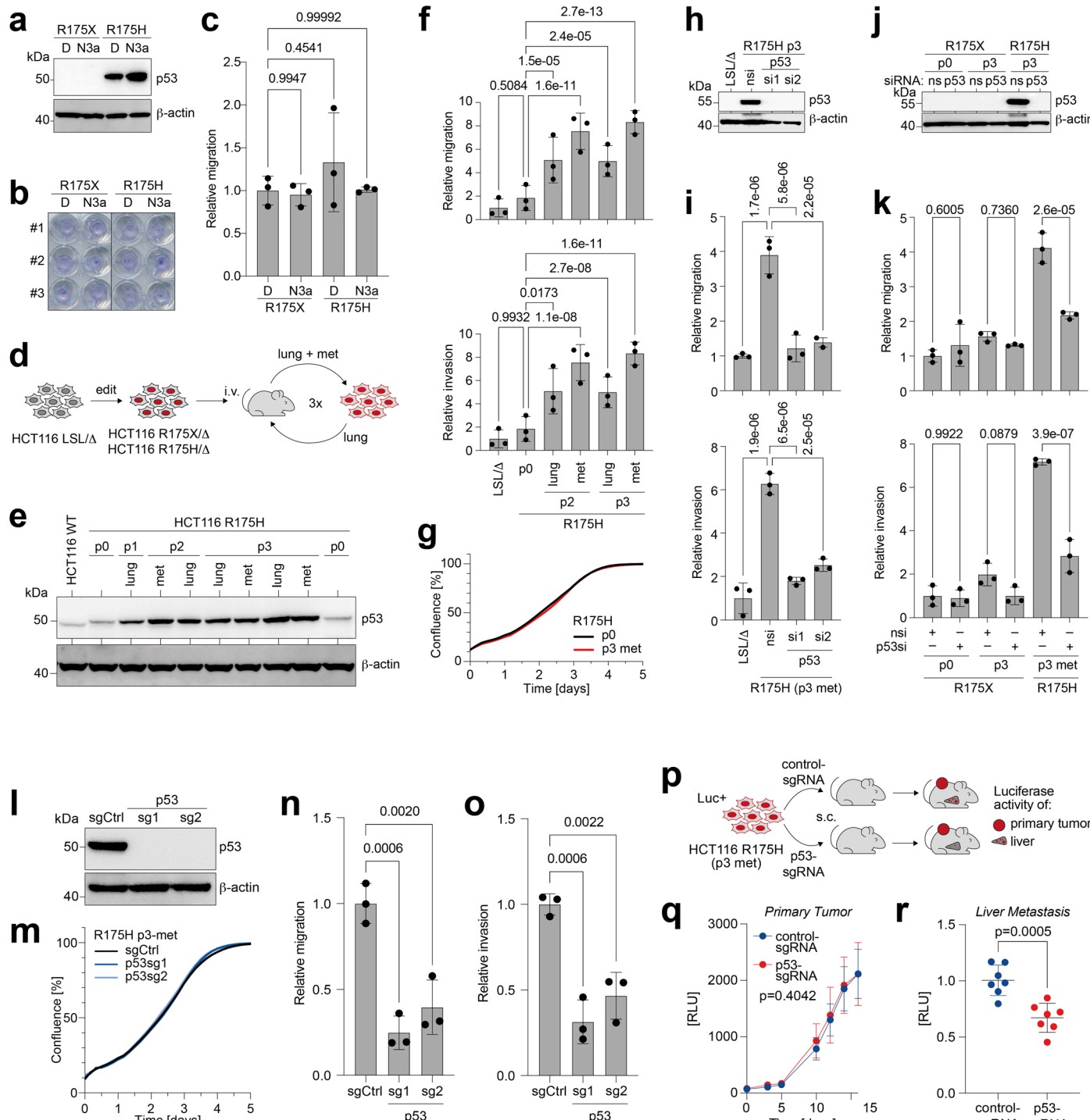

**Extended Data Fig. 3 | R175H enables development of pro-metastatic properties. a-c,** Transwell migration assays of indicated HCT116 R175H/Δ and R175X/Δ cells treated with N3a or DMSO (D). **a,** Western blot. **b,** Images of migration assays stained with crystal violet. **c,** Quantification of migration. Mean ±SD (n=3 biological replicates) relative to DMSO-treated R175X cells; one-way ANOVA with Dunnett's multiple comparisons test. **d-e,** *In vivo* tumor progression model. **d,** HCT116 R175H/Δ and R175X/Δ cells were grown in mice after intravenous injection. Tumors from lungs and metastatic sites were explanted, expanded in cell culture and re-injected for up to 3 mouse passages (p0-3). **e,** Western blot. **f,** Transwell assays for migration and invasion after mouse passaging. Mean ±SD (n=3 biological replicates) relative to original LSL/Δ cells; two-way ANOVA with Dunnett's multiple comparisons test. **g,** Proliferation curves of R175H p0 and p3-met cells measured by real-time live-cell imaging. Shown is the mean confluence of n=3 experiments. **h-k,** Transwell migration and invasion.

nsi, non-silencing siRNA; p53si, p53-targeting siRNA. Mean ±SD (n=3 biological replicates); one-way ANOVA with Šídák's multiple comparisons test. **h** and **j,** p53 Western blots of cells in (**i**) and (**k**). **l-r,** HCT116 R175H/Δ p3-met cells with CRISPR-knockout of p53^R175H. **l,** Western blot. **m,** Proliferation after transduction with p53-targeting or control Cas9-nucleases. **n-o,** Quantification of migration (**n**) and invasion (**o**). Mean ±SD (n=3 biological replicates) relative to sgCtrl-cells; one-way ANOVA with Dunnett's multiple comparisons test. **p-r,** *In vivo* metastasis assay. **p,** HCT116 R175H/Δ p3-met cells were dual-labelled with firefly and secreted Gaussia luciferase, transfected with CRISPR nucleases, and subcutaneously injected into immunodeficient mice. **q,** Primary tumor growth based on secreted Gaussia luciferase levels. Mean ±SD (n=7 mice per group). p-value of group factor from two-way ANOVA. **r,** Liver metastasis based on firefly luciferase activity in whole liver homogenate. Mean ±SD relative to control-sgRNA (n=7 mice per group); two-sided unpaired t-test. RLU, relative light units.

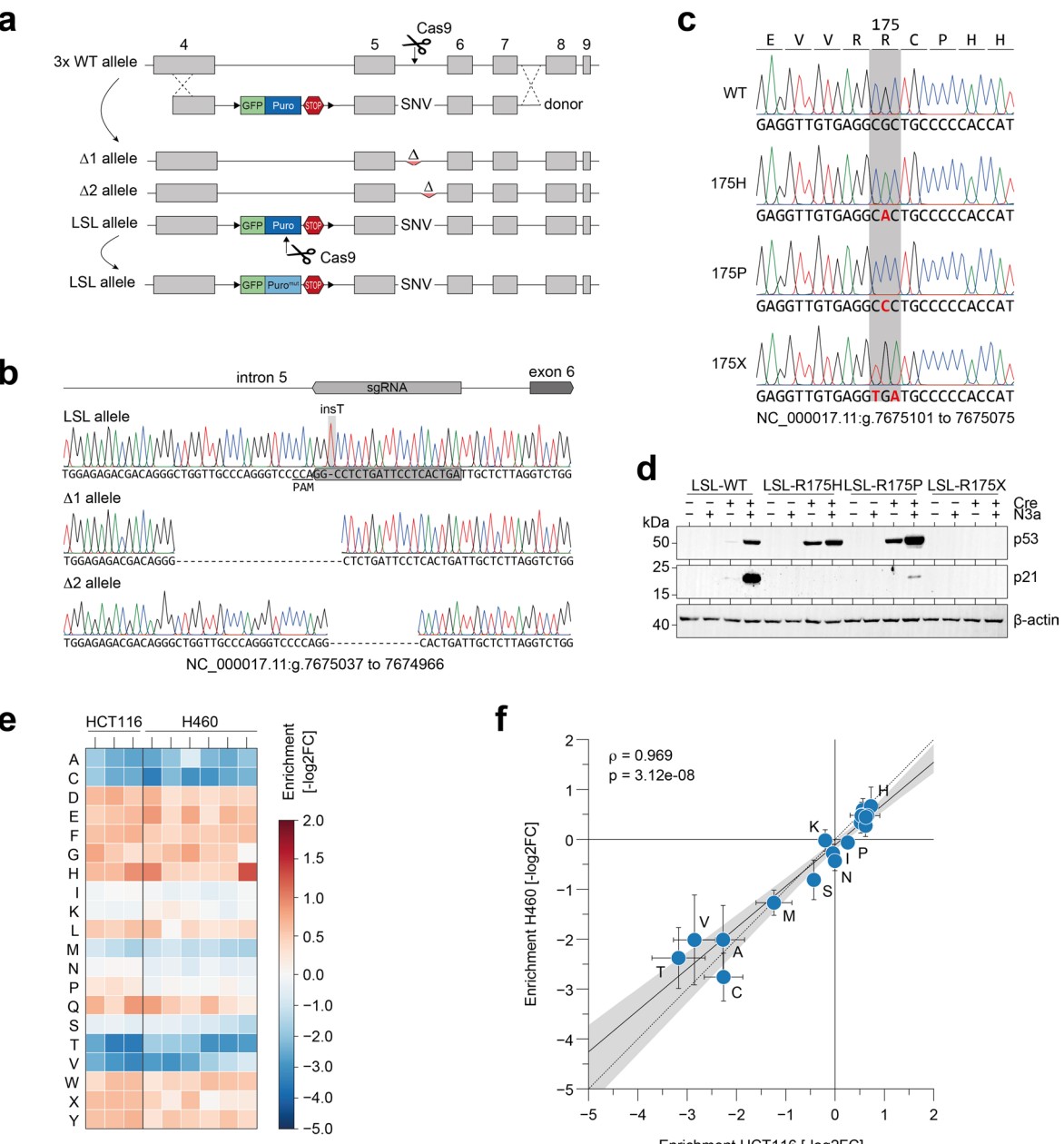

**Extended Data Fig. 4 | R175 mutagenesis screen in H460 cells. a,** Scheme depicting the generation of the H460 LSL/Δ/Δ cell line. **b,** Sanger sequencing results of the three *TP53* alleles in H460 LSL/Δ/Δ cells. **c,** Sanger sequencing results of the LSL allele at codon 175 for R175-edited/mutated H460 cell clones. **d,** Western blot of mutated H460 cell clones ± Cre and N3a. **e,** Heatmap depicting changes in variant abundance following 8 days of 10 μM N3a treatment.

Shown is the -log2 fold change versus the mean of the DMSO-treated controls (HCT116 n=3; H460 n=6 biological replicates). **f,** Scatter plot illustrating the correlation between N3a-induced variant enrichment in HCT116 and H460 cells. Shown is the mean ±SD enrichment (-log2 FC, n=3 biological replicates) and Pearson correlation coefficient ρ with p-value approximated using a two-tailed t-distribution. Dashed line, line of identity.

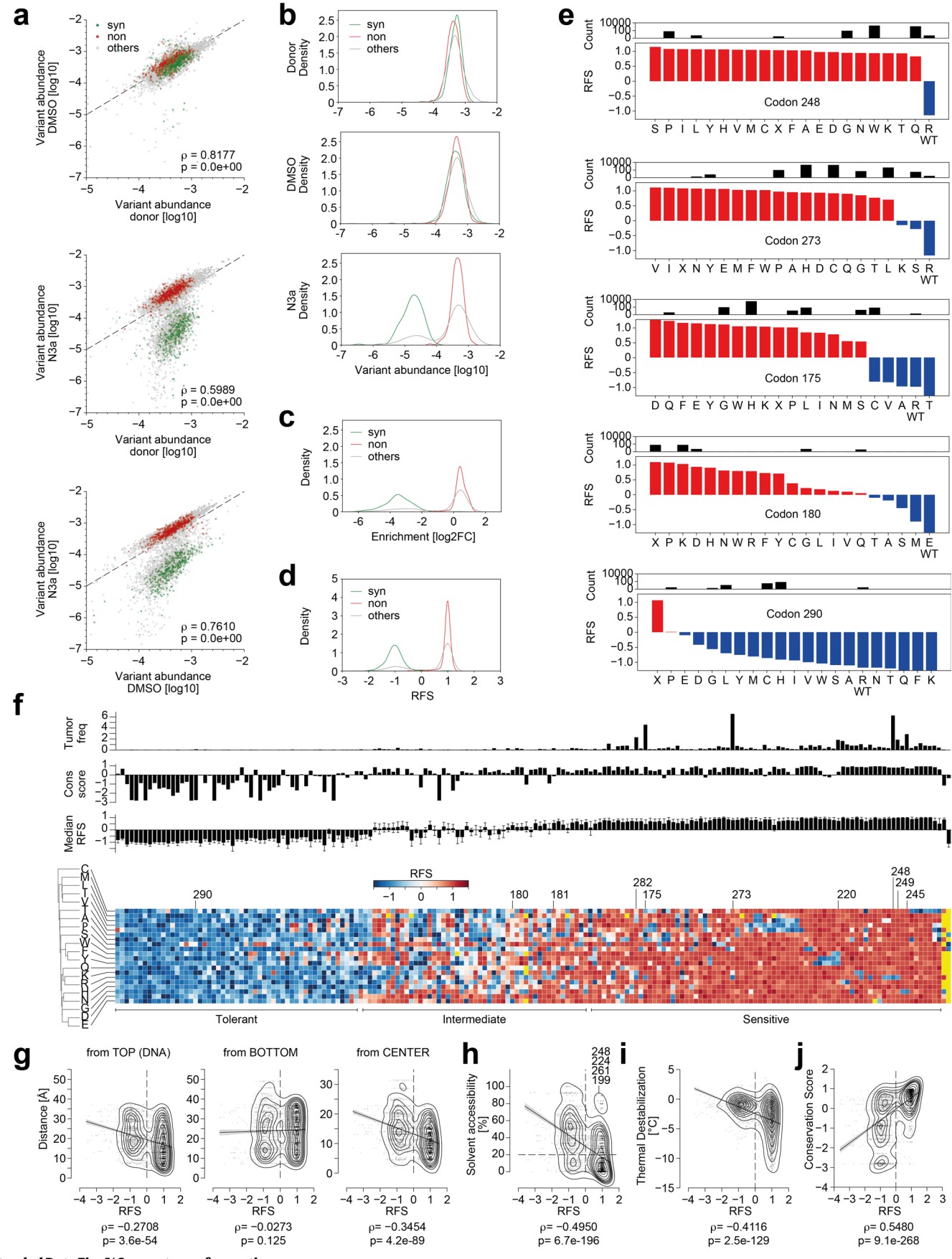

**Extended Data Fig. 5 | See next page for caption.**

**Extended Data Fig. 5 | *TP53* DBD variant screen. a**, Quality control plots illustrating the correlation of variant abundance between donor and DMSO- or N3a-treated cell libraries. Shown is the median abundance (n=3 biological replicates). Syn, synonymous; non, nonsense. ρ, Spearman correlation coefficient with p-value approximated using a two-tailed t-distribution. Dashed line, line of identity. **b-d**, Kernel density estimation (KDE) plots of (**b**) variant abundance in indicated donor and cell libraries, (**c**) enrichment (log2 fold change) under treatment, and (**d**) RFS. **e**, Bar plot showing the median RFS values of all perturbations at exemplary codons (blue, negative RFS indicative of WTp53-like activity; red, positive RFS indicative of loss of WTp53 function). Black bars indicate the patient counts in the UMD *TP53* mutation database. **f**, Hierarchically clustered heatmap showing the RFS for all missense variants. Bar plots show for each codon the mutation frequency in the UMD *TP53* mutation

database, the evolutionary conservation score, and the median±SD RFS at this position. **g**, Scatter plots showing the correlation between RFS and distance of the altered residue from the TOP (DNA-binding surface), BOTTOM (protein pole opposite from the DNA-binding surface) and CENTER of the p53 DBD. **h-j**, Scatter plots showing the correlation between RFS and (**h**) solvent accessibility of the altered residue, (**i**) thermal destabilization of the variant, and (**j**) the conservation score of the altered residue. In **h**, solvent-accessible residues with nevertheless high RFS values are indicated. R248 is a DNA-contact residue, E224 and S261 are located at exon borders and affect splicing, G199 is located at the inter-dimer interface and also critical for splicing. **g-j**, All plots show variants as individual datapoints, kernel density estimates, and regression lines with 95% confidence intervals. ρ, Spearman correlation coefficient with p-value approximated using a two-tailed t-distribution.

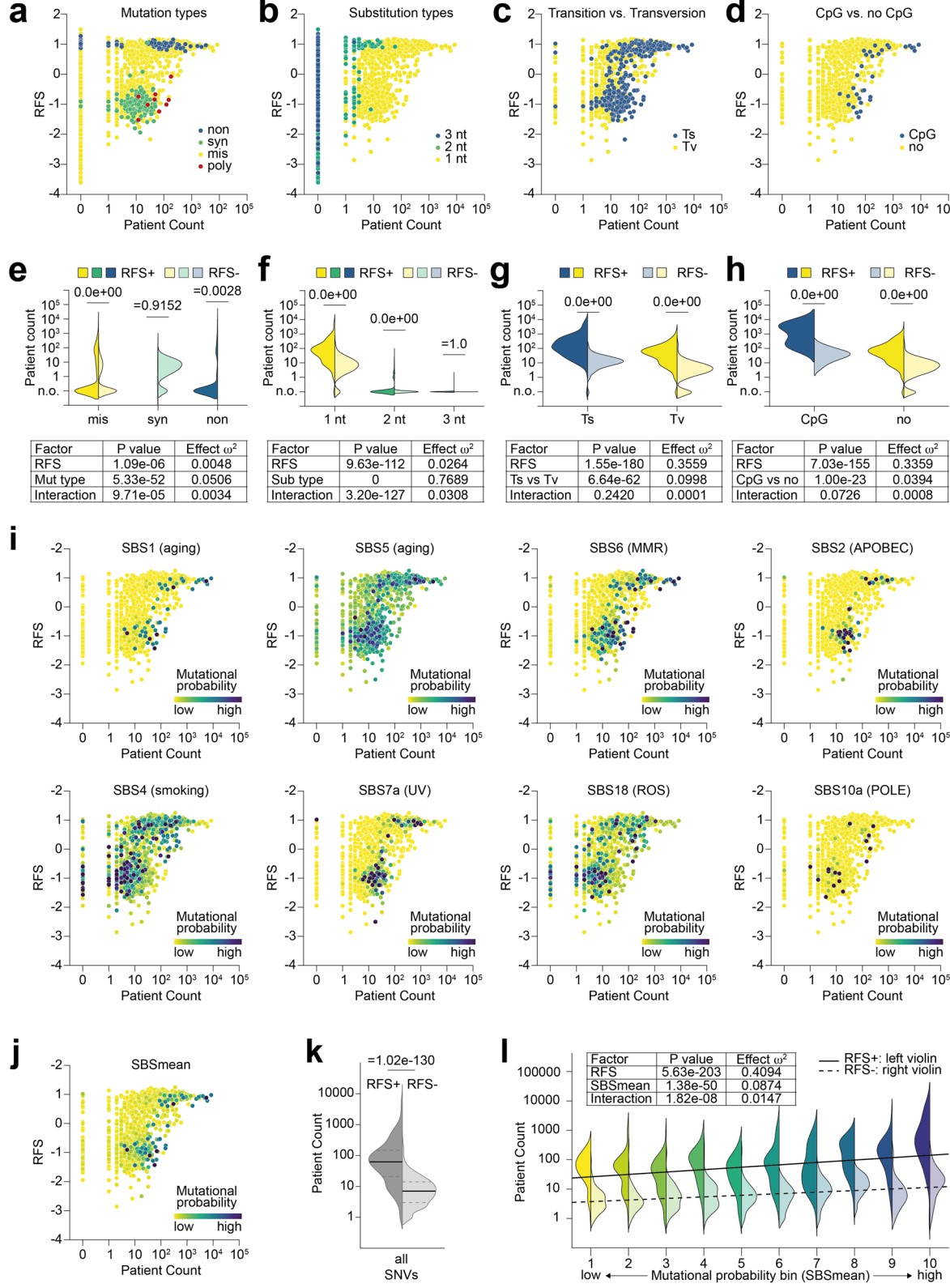

**Extended Data Fig. 6 | See next page for caption.**

**Extended Data Fig. 6 | RFS and mutational probability. a-d**, Scatter plots of RFS versus patient count (sum of all records in the UMD, IARC/NCI, TCGA, and GENIE databases). **a**, Variants are colored by mutation class. Labelled in red are functionally neutral genetic variants (polymorphisms, poly; Doffe et al.[54]). **b**, Missense variants colored by number of substituted nucleotides. **c**, Single-nucleotide missense variants colored as transition Ts (A-G, C-T) or transversion Tv (A-C, A-T, C-A, C-G) mutations. **d**, Single-nucleotide missense variants colored as CpG or non-CpG mutations. **e-h**, Violin plots showing the distribution of patient counts for the mutation types depicted in **a-d** stratified by RFS as RFS+ (RFS>0) or RFS- (RFS<0). n.o., not observed. Tables report the two-way ANOVA p-value and effect size ($\omega^2$) for each factor and their interaction. Selected post-hoc multiple comparison test results (Tukey) are shown directly in the plot. **i-j**, Scatter plots of RFS versus patient count (sum of all records in the UMD, IARC/NCI, TCGA, and

GENIE databases) colored by mutational probability according to the indicated COSMIC mutational signatures (v3.3 - June 2022). SBSmean (**j**) denotes an averaged mutational signature calculated by weighting the most common mutational signatures based on their occurrence in the TCGA pan-cancer cohort. **k** and **l**, Violin plots comparing the distribution of patient counts for single-nucleotide substitutions stratified by RFS. **k**, All single-nucleotide substitutions and p-value from a two-sided Mann-Whitney test. Lines show the median and the 25% and 50% quartiles. **l**, Single-nucleotide substitutions binned by increasing mutational probability using the 'SBSmean' signature. Two-way ANOVA p-value and effect size ($\omega^2$) for each factor ('RFS' and 'SBSmean bin') and their interaction are reported in the table, indicating a strong effect of RFS on patient count mostly independent of mutational probability.

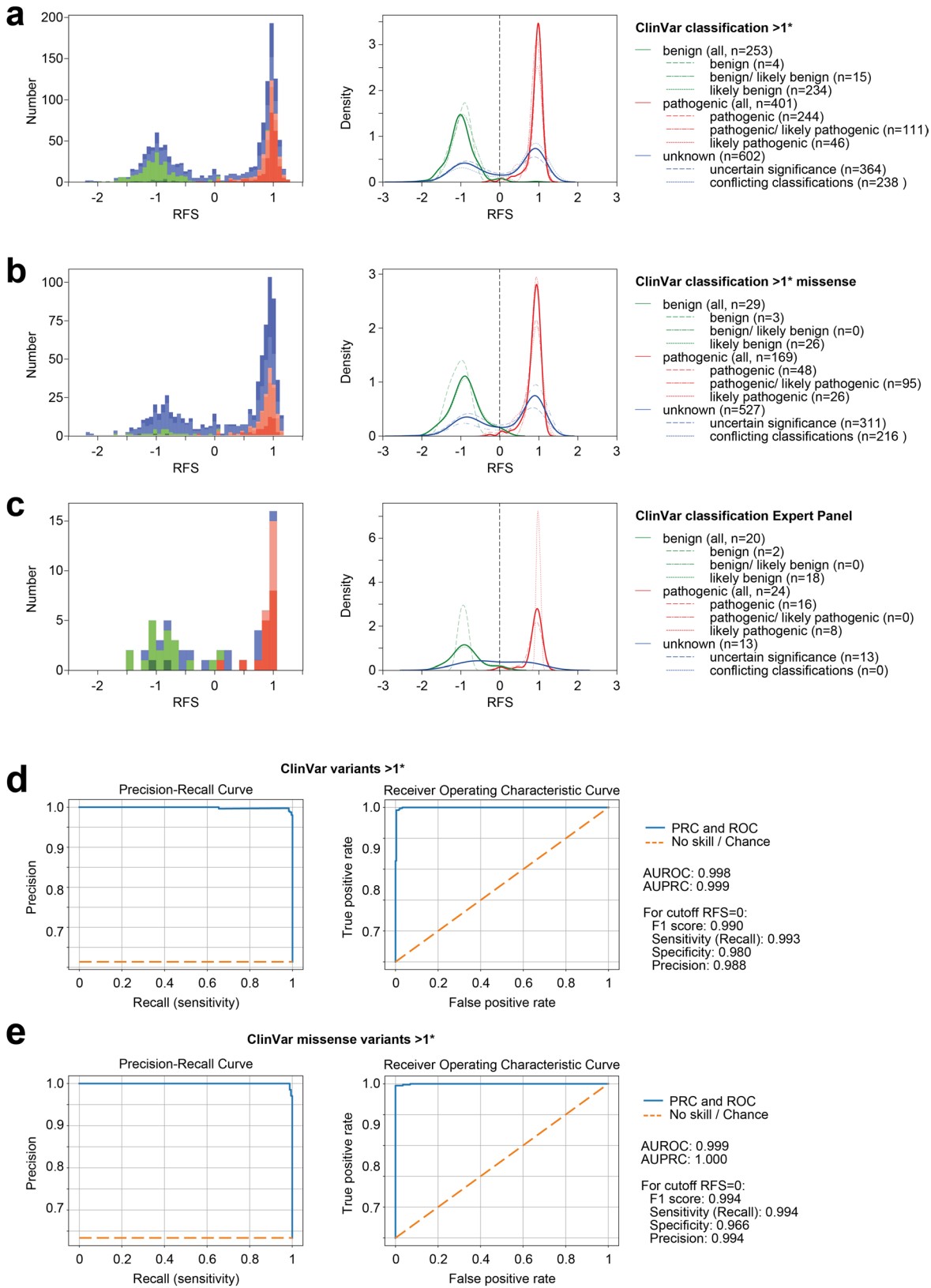

**Extended Data Fig. 7 | Clinical variant interpretation. a-c**, Distribution of RFS values for ClinVar variants colored by pathogenicity classification. *Left*, stacked histograms; *right*, kernel density estimation plots. **a**, ClinVar variants with ≥1* review status. **b**, Missense ClinVar variants with ≥1* review status. **c**, ClinVar variants classified by the *TP53* variant curation expert panel (VCEP). **d** and **e**, Precision-Recall (*left*) and Receiver-Operating Characteristic curves (*right*) for (**d**) ClinVar variants with ≥1* review status and (**e**) missense ClinVar variants with ≥1* review status. AUPRC and AUROC, area under the Precision-Recall and Receiver Operating Characteristic curves, respectively.

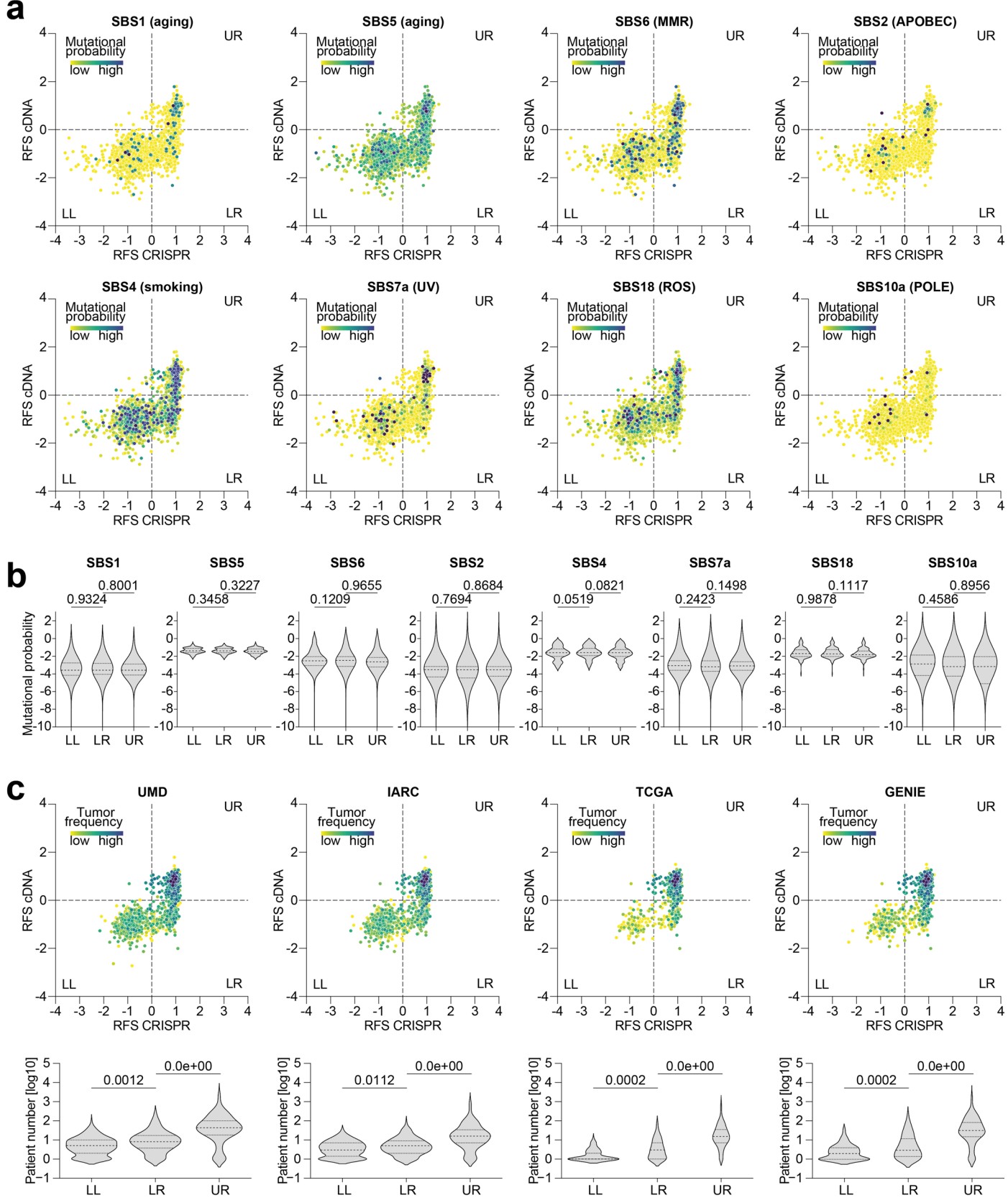

**Extended Data Fig. 8 | See next page for caption.**

**Extended Data Fig. 8 | Mutational probability and tumor frequency.**
**a**, Scatter plots illustrating correlation between RFS values obtained by CRISPR mutagenesis and cDNA overexpression (Kotler et al.[20]). Variants are colored based on their mutational probability according to the indicated COSMIC mutational signatures (v3.3 - June 2022). **b**, Violin plots depict the distribution of mutational probabilities among the variants located in the three main quadrants LL, LR and UR. Reported are p-values from Tukey's post-hoc multiple comparisons tests performed after one-way ANOVA. **c**, Scatter plots illustrate the correlation between RFS values obtained by CRISPR mutagenesis and cDNA overexpression (Kotler et al.[20]). Variants are colored by their frequency in patients based on the indicated mutation databases (UMD, IARC/NCI, TCGA, GENIE). Violin plots depict the distribution of variant patient counts in the three main quadrants LL, LR, and UR (p-values from one-way ANOVA and Tukey's post-hoc multiple comparisons tests). All violin plots show the median and the 25% and 50% quartiles.

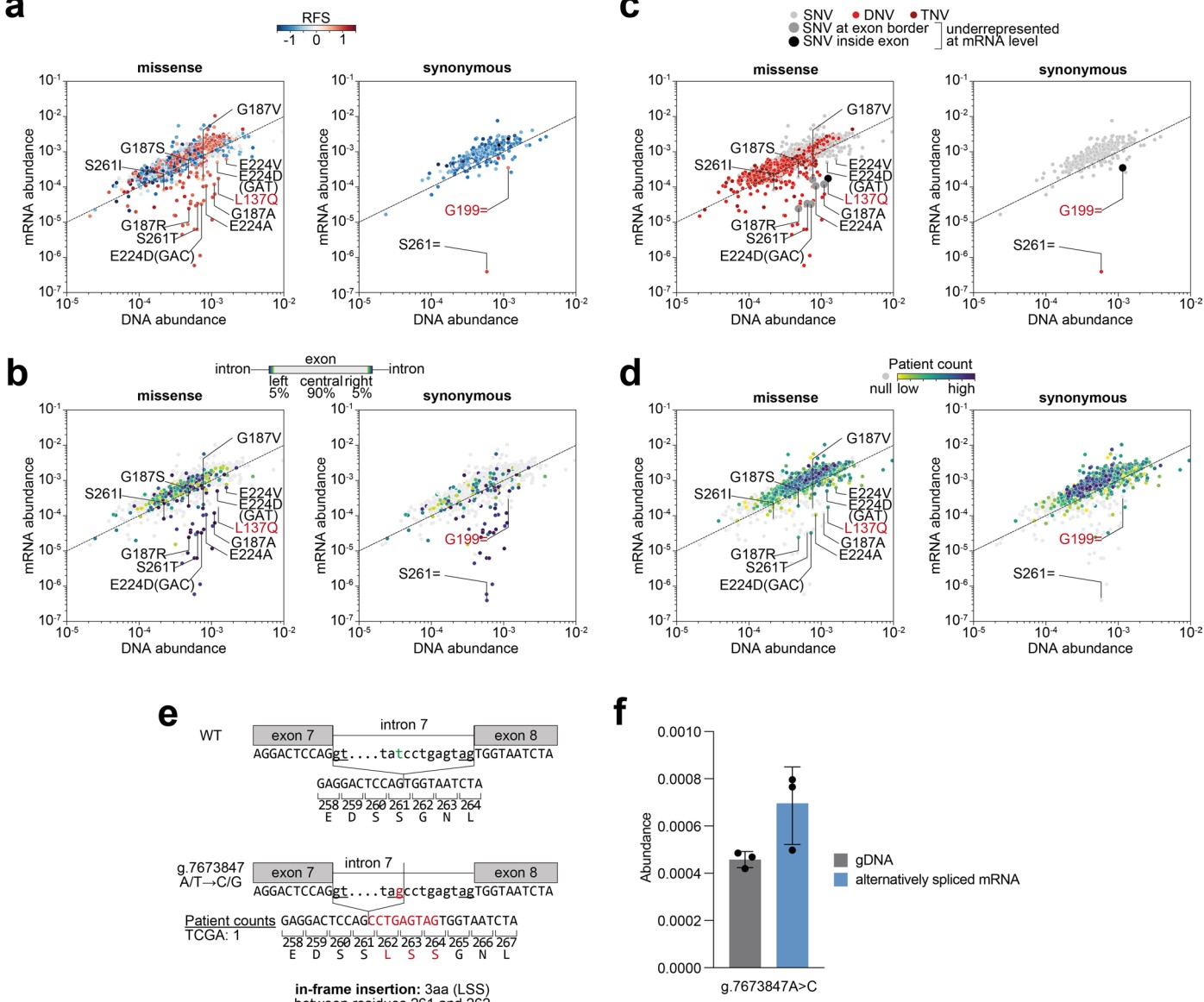

**Extended Data Fig. 9 | Missense and synonymous mutations resulting in splice alterations.** Scatter plots comparing the abundance of missense (left panels) and synonymous (right panels) variants in the cell libraries at the level of genomic DNA and mRNA. Each dot represents the median abundance of a variant from n=3 biological replicates. Variants are colored by RFS in **a**, by distance from the exon border in **b**, substitution type in **c**, and patient count in **d**. LOF variants underrepresented at the mRNA level are individually labeled (black font for variants at the exon border, red font for variants inside the exon). Dashed line, line of identity. **e-f**, Intronic variant NC_000017.11: g.7673847A>C. **e**, Schematic depiction of splicing alterations. **f**, Abundance of the g.7673847A>C variant at the genomic DNA level is similar to the abundance of the aberrantly spliced mRNA. Shown is the mean ±SD of n=3 biological replicates.

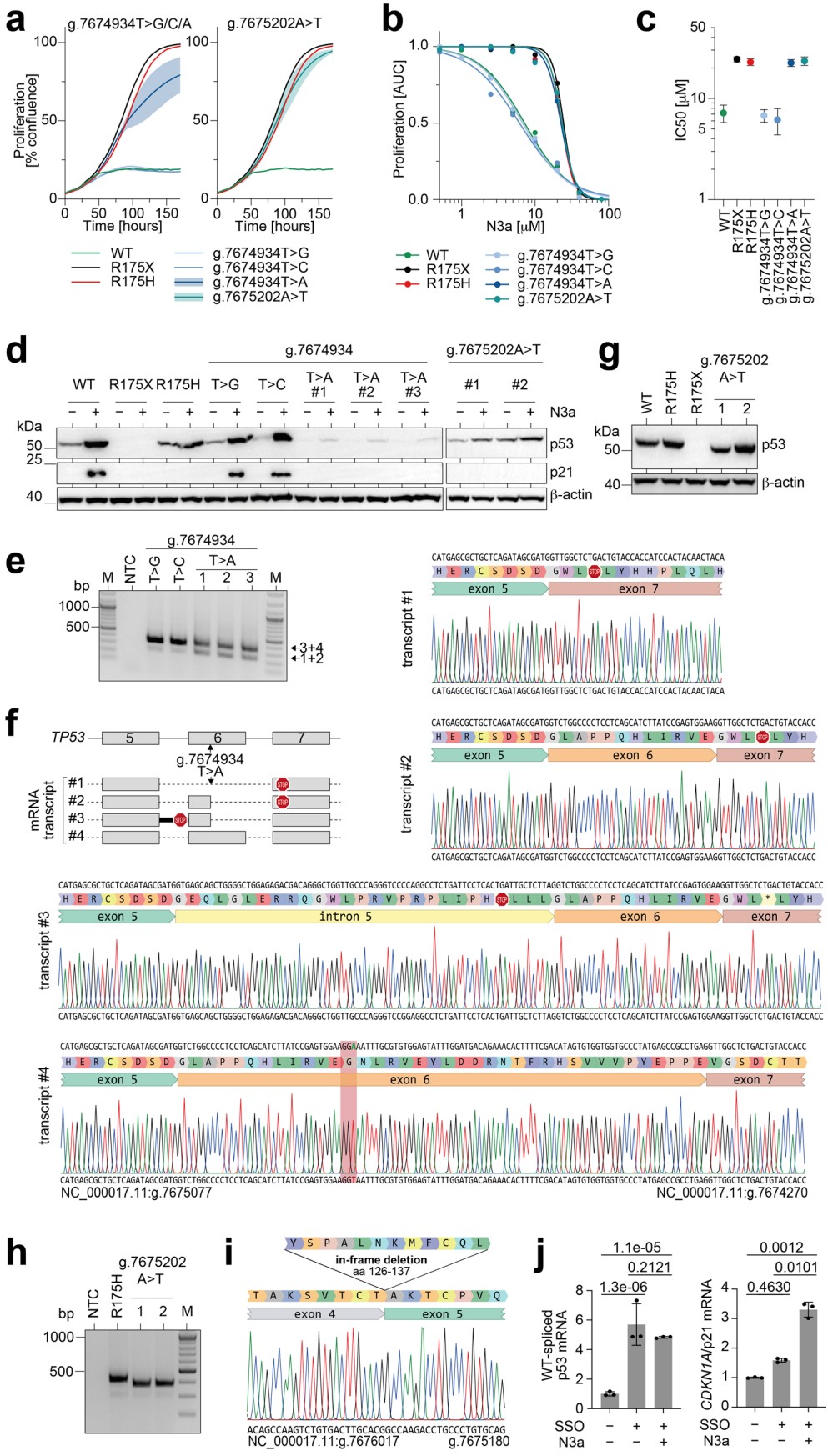

**Extended Data Fig. 10 | See next page for caption.**

**Extended Data Fig. 10 | Aberrant splicing due to exonic SNVs causes LOF.**
**a-c**, Impact of codon 199 variants NC_000017.11:g.7674934T>A/C/G and codon 137 variant NC_000017.11: g.7675202A>T on the anti-proliferative activity of N3a in H460 cells. Wild-type (WT), missense (R175H), and nonsense (R175X) variants are shown for comparison. **a**, Proliferation curves under treatment with 10 μM N3a. For the g.7674934T>A and g.7675202A>T genotypes, plots show the mean +SD of independent clones. **b**, Dose-response curves of N3a calculated from proliferation curves by non-linear regression of area under the curve (AUC) values. **c**, IC$_{50}$ values with 95% confidence interval for N3a calculated from dose-response curves in **b**. **d**, Western blot demonstrating mutant p53 and p21 protein expression in independent H460 clones in the absence and presence of N3a. **e** and **f**, cDNA analysis of g.7674934T>A/C/G clones. **e**, Agarose gel electrophoresis of reverse transcription (RT)-PCR products. **f**, mRNA transcripts detected by Sanger sequencing of RT-PCR amplicons. **g**, Western blot demonstrating reduced size of the p53 protein in H460 clones with the g.7675202A>T genotype. **h** and **i**, cDNA analysis of g.7675202A>T clones. **h**, Agarose gel electrophoresis of RT-PCR products. **i**, Sequencing analysis of RT-PCR amplicons showing an in-frame deletion of 12 amino acids. **j**, Quantitative RT-PCR specific for the regularly spliced p53 and *CDKN1A*/p21 mRNA in H460 g.7675202A>T cells transfected with splice-switching oligonucleotide (SSO) and treated with N3a as indicated. Shown is the mRNA expression relative to untreated as mean±SD (n=3 biological replicates); two-way ANOVA with Tukey's multiple comparisons test.

# Reporting Summary

## Statistics

For all statistical analyses, confirm that the following items are present in the figure legend, table legend, main text, or Methods section.

| n/a | Confirmed | |
|---|---|---|
| ☐ | ☒ | The exact sample size (*n*) for each experimental group/condition, given as a discrete number and unit of measurement |
| ☐ | ☒ | A statement on whether measurements were taken from distinct samples or whether the same sample was measured repeatedly |
| ☐ | ☒ | The statistical test(s) used AND whether they are one- or two-sided *Only common tests should be described solely by name; describe more complex techniques in the Methods section.* |
| ☒ | ☐ | A description of all covariates tested |
| ☐ | ☒ | A description of any assumptions or corrections, such as tests of normality and adjustment for multiple comparisons |
| ☐ | ☒ | A full description of the statistical parameters including central tendency (e.g. means) or other basic estimates (e.g. regression coefficient) AND variation (e.g. standard deviation) or associated estimates of uncertainty (e.g. confidence intervals) |
| ☐ | ☒ | For null hypothesis testing, the test statistic (e.g. *F*, *t*, *r*) with confidence intervals, effect sizes, degrees of freedom and *P* value noted *Give P values as exact values whenever suitable.* |
| ☒ | ☐ | For Bayesian analysis, information on the choice of priors and Markov chain Monte Carlo settings |
| ☒ | ☐ | For hierarchical and complex designs, identification of the appropriate level for tests and full reporting of outcomes |
| ☐ | ☒ | Estimates of effect sizes (e.g. Cohen's *d*, Pearson's *r*), indicating how they were calculated |

*Our web collection on statistics for biologists contains articles on many of the points above.*

## Software and code

Policy information about availability of computer code

| Data collection | ImageLab (v6.0.1)<br>IncuCyte S3 Software (v2018A)<br>BD FACSDiva (v6.1.3)<br>BD Accuri C6 Plus (v1.0.23.1)<br>Summit (v6.3.1)<br>Simplicity (v4.2)<br>Epson Scan (v3.24G)<br>Gen5 (v3.08) |
|---|---|
| Data analysis | GraphPad Prism (9.4.1)<br>Adobe Photoshop CS6 (v13.0.1)<br>Microsoft Excel 2019 (v2301)<br>LightCycler 480 Software (v1.5.0.39)<br>mmdemultiplex (v0.1): https://github.com/MarcoMernberger/mmdemultiplex.git<br>CutAdapt (v3.5)<br>NGmerge (v0.3)<br>STAR (v2.7.10a)<br>UMI-tools (v1.1.1)<br>DEseq2 (v1.34.0)<br>Enrich2 (v1.2.0)<br>GSEA (v4.2.2) |

ScanPy (v1.9.0)
IncuCyte S3 Software (v2018A)
FlowJo (v10.8.1)
ConSurf: https://consurf.tau.ac.il/consurf_index.php (Ben Chorin et al., 2020; doi: 10.1002/pro.3779)
PyMOL (v2.5.2)
ProteinTools: https://proteintools.uni-bayreuth.de (Ferruz et al., 2021; doi: 10.1093/nar/gkab375)
HoTMuSiC: https://soft.dezyme.com (Pucci et al., 2016; doi: 10.1038/srep23257)
Python (v3.9.12) with Matplotlib (v3.5.1), Seaborn (v0.11.2), SciPy (v1.7.3), Statsmodels (v0.13.2)
Adobe Illustrator (26.5.2)

Analysis code is available on GitHub: https://github.com/IMTMarburg/TP53_SGE

For manuscripts utilizing custom algorithms or software that are central to the research but not yet described in published literature, software must be made available to editors and reviewers. We strongly encourage code deposition in a community repository (e.g. GitHub). See the Nature Portfolio guidelines for submitting code & software for further information.

# Data

Policy information about availability of data

All manuscripts must include a data availability statement. This statement should provide the following information, where applicable:
- Accession codes, unique identifiers, or web links for publicly available datasets
- A description of any restrictions on data availability
- For clinical datasets or third party data, please ensure that the statement adheres to our policy

Raw data generated or analyzed for the present study are available as Source Data files. Variant library designs are available as Supplementary Tables S1-S10. For comparison with cDNA-based mutome screens data was used from Kotler et al. 2018 (Supplemental Table 2, RFS_H1299) and from Giacomelli et al., 2018 (Supplementary Table 3, A549_p53NULL_Nutlin-3_Z-score). Publicly available datasets used during this study were UMD TP53 Mutation Database (2017_r2), NCI/IARC The TP53 Database (R20, July 2019), curated set of non-redundant studies (TCGA) and the AACR project GENIE (downloaded from cBioPortal on Dec 20, 2022), Mutational Signatures (v3.3, June 2022, downloaded from COSMIC), ClinVar (downloaded from https://www.ncbi.nlm.nih.gov/clinvar/ on July 27, 2024), and PDB RCSB PDB 2AHI and 3KZ8.
RNA and DNA sequencing data was deposited at EMBL BioStudies, accession numbers E-MTAB-12734 (bulk RNA-seq), E-MTAB-13904 (single-cell RNA-seq), E-MTAB-14322 (TP53 R175 SGE experiments), E-MTAB-12857 (TP53 exon5-8 SGE genomic DNA sequencing), and E-MTAB-12861 (TP53 exon5-8 SGE cDNA sequencing).

# Research involving human participants, their data, or biological material

Policy information about studies with human participants or human data. See also policy information about sex, gender (identity/presentation), and sexual orientation and race, ethnicity and racism.

| | |
|---|---|
| Reporting on sex and gender | N/A |
| Reporting on race, ethnicity, or other socially relevant groupings | N/A |
| Population characteristics | N/A |
| Recruitment | N/A |
| Ethics oversight | N/A |

Note that full information on the approval of the study protocol must also be provided in the manuscript.

# Field-specific reporting

Please select the one below that is the best fit for your research. If you are not sure, read the appropriate sections before making your selection.

☒ Life sciences          ☐ Behavioural & social sciences          ☐ Ecological, evolutionary & environmental sciences

For a reference copy of the document with all sections, see nature.com/documents/nr-reporting-summary-flat.pdf

# Life sciences study design

All studies must disclose on these points even when the disclosure is negative.

| | |
|---|---|
| Sample size | For each experiment, sample size was chosen to obtain sufficient number of experiments and samples to calculate statistical significance. |
| | For SGE experiments, we aimed to investigate all cancer-relevant variants across the parts of the TP53 gene, which encode the DNA-binding domain where >90% of all cancer mutations are located. We therefore included all single-nucleotide variants in exons 5-8 with 12 base pairs of flanking intronic sequence. We further added double and triple nucleotide variants to include all possible amino acid substitutions as well as nonsense and synonymous variants at each codon. Finally, we added all possible single nucleotide insertions and 1-3 base pair deletion, |

yielding a sample size of 9,225 variants.

Plasmid preparation and transfection of CRISPR-constructs for generating the mutome library were performed ensuring a minimal coverage of 500 cells/variant. Mutome screens were performed ensuring a minimal coverage of 1000 cells/variant (genomic DNA) or 50 cells/variant (cDNA).

| Data exclusions | No data could be generated for synonymous mutations at codons for methionine or tryptophan, as no alternative codon is possible. Due to the exon-wise generation of mutome data, some mutations spanning over exon borders couldn't be generated.<br>Variants with lower mean cDNA read counts than 5 were excluded from the analysis. |
|---|---|
| Replication | All experiments were conducted in three replicates, which are all shown in the manuscript and showed excellent correlation. For western blot analysis, representative images of two independent experiments are shown. Data on single cell clones were replicated with 2-10 independent cell clones. |
| Randomization | We performed no randomization in this study because all analyzed TP53 variants were combined with respective negative and positive controls into complex libraries for multiplexed assays of variant effects (MAVE). In these MAVE experiments, variants were introduced into the genome of cells randomly by transfecting cells with the complex library. The resulting cell pools, which contained the variants and controls, were treated and analyzed as pools, and data on individual variants were extracted from these pools. While gDNA libraries and time-point replicates for target regions were grouped together in the same sequencing runs for consistency, the allocation of sequencing runs after grouping was dictated by the availability of gDNA libraries and the timing of experiment readiness, which was essentially random. No additional randomization was performed. |
| Blinding | Blinding was not performed in this study. TP53 variants, along with respective negative and positive controls, were combined into complex libraries for multiplexed assays of variant effects (MAVE). Variants were introduced randomly into the genome via transfection with the pooled library, meaning the identity of the variant in each cell was unknown to the investigator. Consequently, formal blinding was unnecessary as the experimental setup inherently concealed variant identity during analysis. |

# Reporting for specific materials, systems and methods

We require information from authors about some types of materials, experimental systems and methods used in many studies. Here, indicate whether each material, system or method listed is relevant to your study. If you are not sure if a list item applies to your research, read the appropriate section before selecting a response.

## Materials & experimental systems

| n/a | Involved in the study |
|---|---|
| ☐ | ☒ Antibodies |
| ☐ | ☒ Eukaryotic cell lines |
| ☒ | ☐ Palaeontology and archaeology |
| ☐ | ☒ Animals and other organisms |
| ☒ | ☐ Clinical data |
| ☒ | ☐ Dual use research of concern |
| ☒ | ☐ Plants |

## Methods

| n/a | Involved in the study |
|---|---|
| ☒ | ☐ ChIP-seq |
| ☐ | ☒ Flow cytometry |
| ☒ | ☐ MRI-based neuroimaging |

## Antibodies

| Antibodies used | p53 (Santa Cruz Biotechnology, sc-126)<br>p21 (Santa Cruz Biotechnology, sc-6246)<br>β-actin (Abcam, ab6276)<br>goat anti-mouse IgG Fc HRP antibody (Invitrogen, A16084)<br>goat anti-rabbit IgG F(ab') HRP antibody (Amersham, NA9340)<br>goat anti-mouse Alexa-488 conjugate (Invitrogen, A-11029) |
|---|---|
| Validation | p53 (Santa Cruz Biotechnology, sc-126): validated by manufacturer by western blot analysis of p53 expression in A549, Daudi and NTERA-2 cell lysates.<br>p21 (Santa Cruz Biotechnology, sc-6246): validated by manufacturer by western blot analysis of p21 Waf1/Cip1 expression in NIH/3T3 cell lysates.<br>β-actin (Abcam, ab6276): validated by manufacturer by western blot analysis of β-actin expression in β-actin HAP1, HeLa, Jurkat, A431, HEK-293, COS-7, NIH/3T3, PC-12 Rat2, CHO, MDBK and MDCK cell lysates.<br>goat anti-mouse IgG Fc HRP antibody (Invitrogen, A16084): validated by manufacturer by western blot analysis on whole cell extracts of K-562 and U-87 MG using anti-SOD1 antibody (Product # MA1-105, 2 µg/mL) and goat anti-mouse IgG Fc HRP antibody<br>goat anti-rabbit IgG F(ab') HRP antibody (Amersham, NA9340): validated by manufacturer by western blot analysis using anti-beta galactosidase antibody (Cappel) and anti-rabbit IgG, HRP F(ab')2 fragment antibody<br>goat anti-mouse Alexa-488 conjugate (Invitrogen, A-11029): validated by manufacturer by immunofluorescence analysis of HeLa cells stained with alpha Tubulin monoclonal Antibody (#A11126) and goat anti-mouse IgG (H+L) Alexa-488 conjugate. |

# Eukaryotic cell lines

Policy information about cell lines and Sex and Gender in Research

| | |
|---|---|
| Cell line source(s) | HCT116: ATCC (CCL-247)<br>NCI-H460: ATCC (HTB-177)<br>HEK 293T: ATCC (CRL-3216)<br>MCF10A: (CRL-10317)<br>LS-123: (CCL-255)<br>MIA-PaCa2: (CRL-1420)<br>H1975: (CRL-5908)<br>PANC-1: (CRL-1469)<br>PC9: ECACC (90071810)<br>AmphoPack-293: Takara Bio Inc. (CVCL_WI47)<br>Normal human diploid fibroblasts (NHDF) and normal human epidermal keratinocytes were obtained from healthy donors |
| Authentication | HCT116 were authenticated by whole genome sequencing.<br>NCI-H460 were authenticated by STR profiling.<br>LS-123, MIA-PaCa2, H1975, PANC-1, PC9, and MCF10A were authenticated by profiling their TP53 mutation status.<br>HEK 293T and AmphoPack-293 were not formally authenticated, but validated functionally by their lentiviral packaging capacity. |
| Mycoplasma contamination | All cell lines were tested negative for mycoplasma contamination prior to and in the course of experiments. |
| Commonly misidentified lines<br>(See ICLAC register) | No commonly misidentified cell lines were used in this study. |

# Animals and other research organisms

Policy information about studies involving animals; ARRIVE guidelines recommended for reporting animal research, and Sex and Gender in Research

| | |
|---|---|
| Laboratory animals | Rag2tm1.1Flv;Il2rgtm1.1Flv male and female mice at a minimal age of eight weeks |
| Wild animals | The study did not involve wild animals. |
| Reporting on sex | This information has not been collected. Both male and female mice were used for this study, as TP53 mutations in cancers are similarly frequent in males and females. |
| Field-collected samples | The study did not involve field-collected samples. |
| Ethics oversight | Mouse experiments were performed in accordance with the German Animal Welfare Law (TierSchG) and received approval from the animal welfare committee of the local authority (Regierungspräsidium Gießen). |

Note that full information on the approval of the study protocol must also be provided in the manuscript.

# Plants

| | |
|---|---|
| Seed stocks | N/A |
| Novel plant genotypes | N/A |
| Authentication | N/A |

# Flow Cytometry

## Plots

Confirm that:

☒ The axis labels state the marker and fluorochrome used (e.g. CD4-FITC).

☒ The axis scales are clearly visible. Include numbers along axes only for bottom left plot of group (a 'group' is an analysis of identical markers).

☒ All plots are contour plots with outliers or pseudocolor plots.

☒ A numerical value for number of cells or percentage (with statistics) is provided.

## Methodology

| | |
|---|---|
| Sample preparation | As described in the methods section, cells and media supernatants were collected, pelleted, and resuspended in Annexin-V-APC conjugate (MabTag, AnxA100) diluted in Annexin V binding buffer (BD Biosciences, 556454) according to the manufacturer's protocol. The suspension was incubated in the dark for 20 minutes at RT, washed in Annexin V binding buffer and analyzed by flow cytometry on a BD LSR II Flow Cytometer or a BD Accuri C6 Plus Flow Cytometer. Sorting of Annexin V stained cells was performed using a Beckman Coulter MoFlo Astrios sorter . |
| Instrument | BD LSR II Flow Cytometer, BD Accuri C6 Plus Flow Cytometer, Beckman Coulter MoFlo Astrios sorter |
| Software | BD FACSDiva, BD Accuri C6 Plus (v1.0.23.1), Summit (v6.3.1), FlowJo (v10.8.1) |
| Cell population abundance | does not apply |
| Gating strategy | 1.) SSC-H vs. FSC-H gating to exclude debris<br>2.) FSC-H vs. FSC-A gating to exclude doublets<br>3.) APC-A (annexin-V) vs. FITC-A (GFP) gating to quantify / sort GFP negative and annexin-V positive / annexin-V negative cells. Unrecombined, completely recombined and unstained cells were used as controls to set up the gating.<br>See Supplementary Figure 10. |

☒ Tick this box to confirm that a figure exemplifying the gating strategy is provided in the Supplementary Information.

