## [Peer Review File · Nature Genetics]

Deep CRISPR mutagenesis characterizes the functional diversity of TP53 mutations

Corresponding Author: Professor Thorsten Stiewe

Version 0:

Decision Letter:

19th Jun 2024

Dear Professor Stiewe,

First, please accept my sincere apologies for the delay in getting back to you. Thank you for your patience.

Your Article, "Functional diversity of the TP53 mutome revealed by a deep CRISPR mutagenesis scan" has now been seen by 2 referees. You will see from their comments below that while they find your work of interest, some important points are raised. We are interested in the possibility of publishing your study in Nature Genetics, but would like to consider your response to these concerns in the form of a revised manuscript before we make a final decision on publication.

We therefore invite you to revise your manuscript taking into account all reviewer and editor comments. Please highlight all changes in the manuscript text file. At this stage we will need you to upload a copy of the manuscript in MS Word .docx or similar editable format.

*2) If you have not done so already please begin to revise your manuscript so that it conforms to our Article format instructions, available

[here](http://www.nature.com/ng/authors/article_types/index.html).

*3) Include a revised version of any required Reporting Summary: <https://www.nature.com/documents/nr-reporting-summary.pdf>

Link Redacted

We hope to receive your revised manuscript within four to eight weeks. If you cannot send it within this time, please let us know.

Nature Genetics is committed to improving transparency in authorship. As part of our efforts in this direction, we are now requesting that all authors identified as 'corresponding author' on published papers create and link their Open Researcher and Contributor Identifier (ORCID) with their account on the Manuscript Tracking System (MTS), prior to acceptance. ORCID helps the scientific community achieve unambiguous attribution of all scholarly contributions. You can create and link your ORCID from the home page of the MTS by clicking on 'Modify my Springer Nature account'. For more information please visit <http://www.springernature.com/orcid>.

Sincerely,

Safia Danovi, PhD
Senior Editor, Nature Genetics
ORCID: 0009-0007-7822-5479

Referee expertise:

Please note that both reviewers have signed their reports

Referee #1: cancer functional genomic, TP53

Referee #2: cancer functional genomics

Reviewers' Comments:

Reviewer #1:

Remarks to the Author:

In this manuscript, Funk et al. set out to perform a deep mutational scanning of p53's DBD. They have performed an impressive and well-controlled large-scale knock-in CRISPR-mediated HR screen of 3400 alleles and measured their effect using NGS and allele abundance as read out. First, they set up the screen by generating a panel of 14 of the most common single TP53 mutations in HCT116 cell line and characterize their LOF phenotype upon N3a treatment using RNAseq, GSEA, qRT-PCR and proliferation. Next, they perform a R175 saturation screen, which revealed a nice separation-of-function phenotype into variants that cause cell death versus anti-proliferative functions. Lastly, they perform a saturation screen of the whole TP53 DBD including the exon/intron junctions. Comparing their knock-in screen with previous cDNA DMS screens revealed increased sensitivity, likely due to the physiological expression levels, identifying many more variants as potentially pathogenic than the cDNA screens, which is clinically relevant. They could also identified several interesting alterations that affect splicing and NMD, which did not surface in previous cDNA screens.

As the authors chose to inactivate one p53 allele, they obviously could not assess the dominant negative function of all the variants. The potential dominant negative function of several p53 mutants is arguably very important (potentially more important than potential GOF properties reported for some point mutation, which is still somewhat contentious in the field). I feel this major limitation was sufficiently addressed in the discussion and does not distract from the overall impact of the manuscript. Similarly, the authors do address the limitation of functionally assessing GOF properties, albeit I feel that further considerations could be addressed during revisions (see comments below).

Overall, this is a very elegant study delineating how large-scale CRISPR-mediated knock-in mutational scanning approaches can delineate novel insights into protein function in a clinically relevant manner. The data and the methodology are very strong and of highest quality. The manuscript is a very well written (albeit the discussion and some of the result sections are quite lengthy and sometimes repetitive) and it is a very original and innovative study. Some comments and suggestions can be found below.

kind regards,

Daniel Schramek

Most of my comments are rather minor in nature. There are two somewhat more important comments the authors should address:

1, The authors write: 'Given that GOF properties of a missense mutant are dependent on its constitutive stabilization^{15,23,26}, mutational scans in the HCT116 cell model enable a specific interrogation of LOF effects without

being confounded by GOF activities.'

While the data clearly shows the relatively low levels of the endogenous p53 in HCT116 cells and the equally low levels of the knock-in variants as well as the increased levels upon N3a treatment, I think this statement is misleading. In normal non-transformed cells (primary fibroblasts, primary keratinocytes, MCF10A cells etc.), p53 levels are usually undetectable by WB and as such, the HCT116 cells likely show higher levels than non-transformed cells. In addition, the N3a treatment clearly shows that higher p53 levels can be achieved in the HCT116 cells. In fact, the LOF activity can also only be elucidated upon artificially elevating the p53 levels using N3 and robust enrichment/depletion of variants is only observed upon artificial stabilization of p53 levels using N3a. Why would the same not apply to GOF? As such, it is impossible to a priori rule out that the variants could not elicit GOF properties upon N3a and I think the authors might want to rephrase or explain this better. There are several reasons to consider:

a, While the authors show a comparison of p53 levels in cancer cells versus the baseline p53 levels in the engineered HCT116 cells (Extended data Fig. 1f), it would be beneficial to compare the p53 levels of cancer cells versus the N3a-stimulated p53 levels in HCT116 cells. Given the presented data in Extended data Fig. 1d-g, those levels are likely quite comparable.

b, It would be nice to show how the p53 levels in HCT116 cells compare to the baseline and N3a stimulated p53 levels in non-transformed cells. Again, given the data, it is quite likely that baseline p53 expression in HCT116 cells is higher than in non-transformed cells (e.g. MCF10a, primary keratinocytes, etc).

c, While the authors are correct that many GOF p53 mutations show stabilization in tumors and that people ascribe much of the oncogenic effects to these high levels, Li-Fraumeni patients with germline p53 mutations (which are under homeostasis undetectable by WB and IF/IHC) as well as mutant p53 knock-in mice (which under homeostasis in non-tumour tissue also exhibit undetectable p53 levels by WB and IF/IHC) clearly show that certain p53 GOF mutations have an effect even when they are initially lowly expressed.

d, Furthermore, looking carefully at panel 1d, it is quite apparent that only wt and R181L show effects upon N3a treatment, in line with the p21 WB data in 1c. It is also clear that none of the other alleles change upon N3a treatment, but it is also clear that many alleles separate from the untreated wt alleles and the untreated and N3a-treated KO alleles along the PC2 axis! Do the authors just ascribe this to experimental noise (if so, please state in the text or legend) or is this a sign that some alleles indeed show some GOF activity associated with an altered transcriptional profile? The scRNAseq data in Extend Data Fig 2. would rather argue that there is no GOF effect (at least transcriptionally) under baseline and N3a-stimulated conditions, given that the mis and null alleles are largely overlapping. I think this the absence (or presence) of a GOF effects in these experiments should be at least noted in the result section and/or discussion.

e, Lastly, GOF activities are hard to delineate in culture – as the authors correctly point out in the discussion. Did the authors try some xenograft experiments and or migration/invasion assays? I am not clear why this system should not also be a terrific way to test whether these alleles expressed from their endogenous locus show GOF activity. In fact, this would be really important given the ongoing discussions about p53 GOF in the field and would certainly add a lot of weight and importance to the paper esp. because it would offer a glimpse into the behavior of human mutants under endogenous transcriptional control. One could presumably even KO MDM2 to ensure high enough mutant p53 levels (akin to the N3a treatment) for functional tests in vivo or in vitro using e.g. migration/invasion assays.

For these reasons, I think that the statement 'Given that GOF properties of a missense mutant are dependent on its constitutive stabilization^{15,23,26}, mutational scans in the HCT116 cell model enable a specific interrogation of LOF effects without being confounded by GOF activities.' should be revised. All assays in the manuscript are using 'constitutive stabilization' (by N3a), so how can GOF properties (if they exist) be excluded? In fact, this system could/should be used to explore these phenomena or – if the authors feel that it is out of the scope of the current paper - at least discussed as a possibility for exploring/testing GOF phenotypes in the future.

Given the recent heated discussions in the field, one could also limit the exploration of GOF activity to certain known/suspected alleles – e.g. R175H, R248W and R273H.

2, In a similar notion, the authors ascribe all the R175 alleles as LOF or pLOF alleles.

While the Syn alleles have clearly negative effects on cell growth and thus strongly deplete, 'mis variants showed a highly variable mutation-specific response'. Now, it is a little unclear to me how the enrichment score is calculated: the authors write: The ES is 'defined as the negative log2-transformed fold-change compared to the control treatment' – what is meant by 'the control treatment'? In the figure legend, the authors write: 'Enrichment or depletion is shown as the -log2 fold change versus the mean of the early untreated replicates.'

What is an 'early' untreated timepoint (ie. less than 2 days after transduction to compare to the 3d nutlin time point?) Given the importance of these experiments, please be more specific.

In any case, the heatmap clearly indicates that syn alleles decrease over time while most of the mis alleles including the most common R175H allele increase over time to a similar degree as the null alleles. Again, it might be nice to point out that – at least by this measure, none of the R175 mis alleles show a GOF phenotype (despite high N3a-induced mutant p53 levels).

Minor comments:

1, The engineering of the HCT116 cells to allow for the HRD-mediated swapping of p53 variants, is super elegant, but rather complicated to follow for non-CRISPR/HRD aficionados.

For example, it is unclear to me why mutation of the branch point in a p53 intron leads to the loss of the allele? I thought that mutating of the branch point within an intron disrupts the formation of the lariat structure, leading to errors in splicing. Failure to form a lariat usually result in the retention of the intron or the inclusion of part of the intron in the final mRNA molecule. Why loss of the gene expression? Is it non-sense mediated decay? Is this known for this branch point in intron 5? Maybe a reference or some explanation in the text (or at least in the methods or supplement) would help less-expert readers to understand this better.

Similarly, I think it would help many readers if the authors explain in more details the process of HRD. For example, the authors write: 'The LSL allele was specifically cleaved by CRISPR/Cas9 nucleases targeting the intronic regions deleted on the allele.' Presumably, the cutting of the LSL allele is increasing the HRD frequency, right? It would be beneficial to mention/explain this for the general audience.

2, The authors should comment on/explain the discrepancy between the frequency of successfully targeted clones assessed by PCR versus sequencing.

3, The authors write: 'Focusing on missense mutations, we explored the functional consequences of replacing each residue with every other amino acid. The screen returned reliable RFS values for 3,425 (99.05%) of all possible 3,458 missense mutant proteins, making this the most comprehensive study of the DBD mutome to date'. Giacomelli et al. (Nat Gen 2018) performed a TP53 saturation mutagenesis screens in an isogenic pair of TP53-wild-type and -null cell lines using nutlin and etoposide as p53 trigger to delineate LOF, GOF and dominant negative features of 8,258 p53 mutant alleles, so that each of the 20 natural amino acids and a stop codon would be represented at each codon position across the entire p53 gene (not 'just' the DBD). As such, while the current screen is more sensitive, how can it be more comprehensive? Please omit, rephrase or explain.

4, It would be nice to label the missense R175 variants that shift upon N3a similarly to wtp53 in the 3rd blot of panel 2b and panel 2d.

5, Figure Panel 2e – there is a black line before the 25d untreated timepoint, which is confusing. Please explain or remove. Also, why is the 25d timepoint placed at the end of the untreated timepoint? Why is this 25d timepoint showing higher depletion/enrichment or at least more variable results than the 2 week and 4 week timepoint? Presumably the 25d untreated timepoint was chosen to match the 25d treated timepoint?

6, Fig. 7J – it is obvious that SSO treatment stabilizes the mRNA of p53 and also leads to an induction of p21 upon co-treatment of N3a. However, it is unclear to what extent this happens. It would be nice to have wt p53 and wt p53 associated p21 levels in the same experiment.

Reviewer #2:

Remarks to the Author:

Overall, the first author and team should be commended for an incredible amount of work. If the replicates are real biological replicates and not PCR replicates, this will be an addition to the existing trove of variant effect data for TP53. Its impact will depend on the ability to validate the data for clinical use which is not something the authors attempted here.

My major concern is that throughout this paper, the authors wrote the manuscript in a way that makes it seem as though the methods were developed in their lab, for this work, for the first time. This is far from reality. Deep mutational scans (DMS) that employ CRISPR-mediated homology-directed repair (CRISPR-HDR) to install edits are called saturation genome editing (SGE) – a method developed 10 years ago by Jay Shendure's lab (Findlay et al. Nature 2014). There are many citations missing for both SGE and other CRISPR based methods to install and measure functional effects of variants (listed below). Additionally, nearly the entire protocol (minus the upstream cell line engineering and cre step) and analysis performed here is standard operating procedure for SGE specifically and DMS generally – splitting up variant libraries into Illumina-sized chunks, using synthetic oligos to make variant libraries, and the exon normalization method, yet none are cited. I'm hopeful that these missing citations were not an intentional omission of the literature. And for the students' sake, I am hopeful y'all didn't actually reinvent all these wheels. There are plenty of resources available through the Atlas of Variant Effects Alliance.

A second major complaint is that the words pathogenic and likely pathogenic are protected terms. You cannot deem a variant pathogenic or likely pathogenic based solely on the outcome of a functional assay. Functional data must be translated into evidence for clinical variant classification and combined with other sources of evidence to classify variants as pathogenic, likely path, VUS, likely benign or benign (Richards ACMG/AMP guidance and Brnich ClinGen SVI update). Please see ClinGen's guidance for interpreting variants in TP53 (ClinGen CSPEC, <https://cspec.genome.network/cspec/ui/svi/doc/GN009>). This manuscript must be heavily edited to avoid labeling variants

pathogenic unless already classified as such. Please see additional guidance specifically for publishing clinically relevant mutational scans (Gelman 2020)

Third, one of the regions of TP53 where variants are the hardest to classify is the tetramerization domain. This is not covered in this study.

Fourth, a complete data set is not included in the supplemental tables and the data and code have not been submitted to a stable repository.

Specific comments:

Mutome is not (and should never be) a word.

The first sentence of the abstract – over 2000 mutations described in patients... Where is this number coming from? If ClinVar, it is 2000 variants described in individuals, if TCGA 2000 mutations described in tumors. Either way, the sentence should be edited.

The final paragraph of the introduction is missing all the citations for saturation genome editing by HDR, base and prime editing, see below. This list is not comprehensive, there could be some I've missed.

Omitted saturation genome editing citations:

Findlay et al. Nature 2014 (method development)

Findlay et al. Nature 2018 (BRCA1)

Meitlis et al. AJHG 2020 (ssODN-based HDR, CARD11)

Radford et al. Nature Comms 2023 (DDX3X)

Sahu et al. BioRxiv 2023 (ssODN-based HDR, BRCA2)

Huang et al. BioRxiv 2023 (BRCA2)

Findlay lab, BioRxiv 2023 (VHL)

Omitted saturation prime editing citations:

Erwood et al. Nature Biotech (NPC and BRCA2 2022)

Omitted saturation base editing citation:

Hanna et al. Cell 2021 (multiple genes)

Etc

Figure 1 DEF?

Why is WT clone 2 showing some LOF? Can you find a structural rearrangement by sequencing the locus?

Saturation genome editing specifically and DMS generally are accepted methods for understanding the effect of genetic variation. Most of figures 2 and 3 do not add much information beyond the full mutational scan.

Typo line 220

The APR-246 or ZMC1 experiments are a good use of the technology.

It is nice that there is validation across 2 cell lines for a small number of variants. However, they are both cancer cell lines and therefore less informative than a second cell line that is closer to normal.

Page 14 Lines 279-281. This is an overreach. Many DMS experiments regardless of the way the variants are introduced identify mechanism and separation of function variants.

Lines 295-300 page 15. Describing both Tile-seq and standard saturation genome editing experimental design. Both are published methods and should be cited (see Findlay et al and the Roth lab's publications).

Page 16 lines 317-319 – Relative fitness scores are nearly the same as a procedure also developed by Greg Findlay for SGE (Findlay et al 2018)

Page 16 lines 330-331 – kind of. This library contains many extraneous variants that aren't terribly clinically useful (multiple nucleotide aa changes, all the indels). And the Giacomelli and Boettcher papers scanned the whole protein.

Page 18 lines 375-380 – if I recall correctly, an analysis of mutational propensity was also performed by Giacomelli et al. and they found something similar. Citation?

Page 18 lines 395-388 – unless you're going to calculate a positive and negative predictive value or likelihood ratio for identifying truly pathogenic variants (that have a label in ClinVar), this is an overreach.

Figure 6. Could use a more in-depth comparison to Kato. Kato has been the gold standard for assessing variants in TP53 for clinical variant interpretation.

Also, this is a bit unfair to Giacomelli, they did their screens in 2 different conditions. You're only comparing to one of them.

Fayer et al. AJHG 2021 found that combining the 4 datasets from Giacomelli and Boettcher gave them more predictive power to correctly predict variants of known effect (ClinVar variants).

Page 20, Line 411, overreach. "Likely pathogenic" is a protected term as clinical variant class, maybe "potentially pathogenic"?

Page 21, lines 448-451, overreach, this is not the first mutational scan to correctly observe splice defects, see missing saturation genome editing citations above

Page 22, line 472, What you describe is validation of the damage or splice defect not pathogenicity. Determining pathogenicity requires a completely different procedure (see Clingen CSPEC for TP53)

Page 23, lines 494-496, nope. You cannot say this. Determining pathogenicity requires a completely different procedure (see Clingen CSPEC for TP53)

Page 24, line 497 "intraexonic" is confusing term. Aren't they just exonic?

Page 25, lines 507 -510 Again, what you describe is validation of the damage or splice defect not pathogenicity. Determining pathogenicity requires a completely different procedure (see Clingen CSPEC for TP53)

Page 26 you can't just call any variant pathogenic based on the outcome of the screen. g.7675202A>T, could potentially be pathogenic. Is it seen in tumors? Individuals with Li-Fraumeni syndrome?

Line 556 - more frequent in tumors not patients

Line 557 – nope. You cannot say this. Determining pathogenicity requires a completely different procedure (see Clingen CSPEC for TP53)

Lines 659 – 662 nope. You cannot say this. Determining pathogenicity requires a completely different procedure (see Clingen CSPEC for TP53)

Methods:

Lines 826 – 832. What was the replicate structure for these experiments? Were they complete/biological replicates e.g. transfected into 3 separate populations of cells or partial/technical replicates, a single transfection with 3 separate extraction/PCR reactions? This is important to describe accurately.

Line 843-844: What is meant by editing-specific primers? Would these only pick up edited alleles? How much of the final cell libraries are unedited?

Alignment by exact matching analysis was done in the Sanger's DDX3X paper (Radford et al.).

863-872 Score not sufficiently explained in methods. I think I know what was done, but it should be explained in more detail so people could replicate the score.

I can't find the complete data set in the supplementary tables. Table S2 has the variant identifiers and variant sequence, table S7 has missense variants and crispr_rfs scores. But I cannot find a table with both of those together for the complete data set of variants. Can a single table be generated with all the variant identifiers and scores along with scores from individual replicates, variants counts?

The contents of the supplementary tables are insufficiently described.

Is the code available on Github or another stable repository?

Data variants and scores must be uploaded to MaveDB and sequence reads uploaded to SRA before publication.

Lea Starita, University of Washington

Version 1:

Decision Letter:

Our ref: NG-A64951R

12th Sep 2024

Dear Dr Stiewe,

Thank you for submitting your revised manuscript "Functional diversity in the TP53 mutational landscape revealed by a deep CRISPR mutagenesis scan" (NG-A64951R). It has now been seen by the original referees and their comments are

below. The reviewers find that the paper has improved in revision, and therefore we'll be happy in principle to publish it in Nature Genetics, pending minor revisions to satisfy the referees' final requests and to comply with our editorial and formatting guidelines.

Sincerely,

Safia Danovi, PhD
Senior Editor, Nature Genetics
ORCID: 0009-0007-7822-5479

Reviewer #1 (Remarks to the Author):

The authors did a very thorough job in addressing all my comments. I am esp. impressed with Extended Fig 3, which offers a very elaborate view in p53 R175H GOF ability, which apparently only manifests upon serial transplantation in mice, which is very interesting. The additional explanations added to the manuscript will certainly help readers understand the intricacies of this complex but beautiful screening paradigm and understand the benefits of this screening paradigm. The authors also provided a nice heatmap comparison between their study and previous mutational scanning efforts (page 9 of the rebuttal letter), which might be nice to include into the extended data. Overall, I think this manuscript will be an important study with regards to the technical advances and the new insights into p53 biology and thus appeal to the broad readership of Nature Genetics.

Reviewer #2 (Remarks to the Author):

I am satisfied with the edits.

Response to Referees

We thank both reviewers for the thorough and constructive review. The comments were extremely helpful for us to improve the quality of our manuscript.

Reviewer #1:

Remarks to the Author:

In this manuscript, Funk et al. set out to perform a deep mutational scanning of p53's DBD. They have performed an impressive and well-controlled large-scale knock-in CRISPR-mediated HR screen of 3400 alleles and measured their effect using NGS and allele abundance as read out. First, they set up the screen by generating a panel of 14 of the most common single TP53 mutations in HCT116 cell line and characterize their LOF phenotype upon N3a treatment using RNAseq, GSEA, qRT-PCR and proliferation. Next, they perform a R175 saturation screen, which revealed a nice separation-of-function phenotype into variants that cause cell death versus anti-proliferative functions. Lastly, they perform a saturation screen of the whole TP53 DBD including the exon/intron junctions. Comparing their knock-in screen with previous cDNA DMS screens revealed increased sensitivity, likely due to the physiological expression levels, identifying many more variants as potentially pathogenic than the cDNA screens, which is clinically relevant. They could also identified several interesting alterations that affect splicing and NMD, which did not surface in previous cDNA screens.

As the authors chose to inactivate one p53 allele, they obviously could not assess the dominant negative function of all the variants. The potential dominant negative function of several p53 mutants is arguably very important (potentially more important than potential GOF properties reported for some point mutation, which is still somewhat contentious in the field). I feel this major limitation was sufficiently addressed in the discussion and does not distract from the overall impact of the manuscript. Similarly, the authors do address the limitation of functionally assessing GOF properties, albeit I feel that further considerations could be addressed during revisions (see comments below).

Overall, this is a very elegant study delineating how large-scale CRISPR-mediated knock-in mutational scanning approaches can delineate novel insights into protein function in a clinically relevant manner. The data and the methodology are very strong and of highest quality. The manuscript is a very well written (albeit the discussion and some of the result sections are quite lengthy and sometimes repetitive) and it is a very original and innovative study. Some comments and suggestions can be found below.

*kind regards,
Daniel Schramek*

We are grateful to your extremely helpful review! Many thanks!

Most of my comments are rather minor in nature. There are two somewhat more important comments the authors should address:

1, The authors write: 'Given that GOF properties of a missense mutant are dependent on its constitutive stabilization^{15,23,26}, mutational scans in the HCT116 cell model enable a specific interrogation of LOF effects without being confounded by GOF activities.'

While the data clearly shows the relatively low levels of the endogenous p53 in HCT116 cells and the equally low levels of the knock-in variants as well as the increased levels upon N3a treatment, I think this statement is misleading. In normal non-transformed cells (primary fibroblasts, primary keratinocytes, MCF10A cells etc.), p53 levels are usually undetectable by WB and as such, the HCT116 cells likely show higher levels than non-transformed cells. In addition, the N3a treatment clearly shows that higher p53 levels can be achieved in the HCT116 cells. In fact, the LOF activity can also only be elucidated upon artificially elevating the p53 levels using N3 and robust enrichment/depletion of variants is only observed upon artificial stabilization of p53 levels using N3a. Why would the same not apply to GOF? As such, it is impossible to a priori rule out that the variants could not elicit GOF properties upon N3a and I think the authors might want to rephrase or explain this better.

We thank you for this comment on GOF – a very controversial topic in the p53 field. We have investigated GOF mechanisms of p53 mutants in the past, for example PMID 37563605 and 27956623, and therefore did not a priori rule out GOF properties. In fact, the **new Extended Data Fig. 3** shows that at least the R175H variant can induce pro-metastatic GOF effects in the HCT116 model after long-term *in vivo* propagation. These new experiments are described in more detail in response to your point 1e.

However, in our multiplexed assays for cellular fitness we observed no differences between p53-null variants (nonsense and frameshift variants) and deleterious missense variants. If missense variants exhibit GOF effects in HCT116 cells, they are not captured by our functional fitness assay.

We did not mean to argue that there is no GOF, but rather that our assay specifically measures LOF – not GOF. We have rephrased our statement to clearly state that the specificity for LOF effects refers to our functional fitness assay, not missense mutations in general or the cellular model itself.

There are several reasons to consider:

a, While the authors show a comparison of p53 levels in cancer cells versus the baseline p53 levels in the engineered HCT116 cells (Extended data Fig. 1f), it would be beneficial to compare the p53 levels of cancer cells versus the N3a-stimulated p53 levels in HCT116 cells. Given the presented data in Extended data Fig. 1d-g, those levels are likely quite comparable.

We have added a new Western blot including a quantification (**new Extended Data Fig. 1i-j**) that compares not only baseline, but also N3a-induced levels of mutant p53 proteins in HCT116 cells to the panel of p53-mutant cancer cells. While N3a clearly increased the mutant p53 expression in HCT116 cells, the obtained expression levels were significantly lower than baseline levels in cancer cells with naturally acquired p53 mutations.

b, It would be nice to show how the p53 levels in HCT116 cells compare to the baseline and N3a stimulated p53 levels in non-transformed cells. Again, given the data, it is quite likely that baseline p53 expression in HCT116 cells is higher than in non-transformed cells (e.g. MCF10a, primary keratinocytes, etc).

We have added a new Western blot (**new Extended Data Fig. 1h**) that compares baseline and N3a-induced WT and R175H p53 protein levels in HCT116 cells to those in normal human diploid fibroblasts (NHDF) from two different donors, MCF10a mammary epithelial cells, and normal human epidermal keratinocytes (NHEK). We readily detected baseline p53 levels in both non-transformed and HCT116 cells. These levels did not show considerable differences between the cell types. Our results align with our experience that baseline p53 levels in non-transformed cells, while weak, are clearly detectable using high-affinity antibodies such as the well-characterized monoclonal anti-p53 antibody DO-1.

c, While the authors are correct that many GOF p53 mutations show stabilization in tumors and that people ascribe much of the oncogenic effects to these high levels, Li-Fraumeni patients with germline p53 mutations (which are under homeostasis undetectable by WB and IF/IHC) as well as mutant p53 knock-in mice (which under homeostasis in non-tumour tissue also exhibit undetectable p53 levels by WB and IF/IHC) clearly show that certain p53 GOF mutations have an effect even when they are initially lowly expressed.

The cancer susceptibility of Li-Fraumeni patients and knock-in mice with p53 missense mutations is undisputed, but by itself no evidence for GOF activity, because the LOF caused by these mutations is sufficient to explain cancer susceptibility. Additional GOF can only be claimed if missense mutations induce a higher cancer susceptibility than nonsense/frameshift mutations or a p53 knock-out in mice.

In Li-Fraumeni syndrome, the aspect of GOF is difficult to address, as germline carriers of *TP53* mutations are heterozygous for the mutations. Earlier tumor onset in missense versus nonsense/frameshift mutations has been reported (PMID: 23538418) and may be due to GOF activity, but can also be explained by the dominant-negative activity, which is present in missense but absent in nonsense/frameshift mutants.

GOF can only be cleanly distinguished from the dominant-negative effect in a homozygous setting, which requires mouse models. In homozygous mutant mice, most p53 missense mutations (for example, R172H and R270H, corresponding to the human hotspot mutations R175H and R273H) do not accelerate tumorigenesis or induce more tumors than a p53 knockout (PMID: 15607981, 15607980). The survival curves of these mice are indistinguishable from knock-out mice (PMID: 18483220). As the mutant protein in these mice is not stable, this underlines the hypothesis that there is no GOF activity in primary non-transformed cells where the mutant is not stabilized.

In contrast, tumors with some missense mutations show a more aggressive and metastatic phenotype, suggesting GOF activity, but these tumors also have constitutively stabilized mutant p53. Together this supported the notion that GOF activity only becomes evident after the mutant protein has become stabilized. This concept has been experimentally confirmed by Terzian et al. (PMID: 18483220), who showed that a Mdm2 or p16INK4a knock-out stabilizes mutant p53 in non-transformed cells along with an earlier age of tumor onset and more metastatic tumors than observed in either p53 knockout or p53 missense mutant mice without additional Mdm2 or p16 knock-out.

In conclusion, data from homozygous-mutant mice, in which GOF can be cleanly separated from dominant-negative activity, argue that GOF activity relies on the prior stabilization of the mutant protein.

d, Furthermore, looking carefully at panel 1d, it is quite apparent that only wt and R181L show effects upon N3a treatment, in line with the p21 WB data in 1c. It is also clear that none of the other alleles change upon N3a treatment, but it is also clear that many alleles separate from the untreated wt alleles and the untreated and N3a-treated KO alleles along the PC2 axis! Do the authors just ascribe this to experimental noise (if so, please state in the text or legend) or is this a sign that some alleles indeed show some GOF activity associated with an altered transcriptional profile? The scRNAseq data in Extend Data Fig 2. would rather argue that there is no GOF effect (at least transcriptionally) under baseline and N3a-stimulated conditions, given that the mis and null alleles are largely overlapping. I think this the absence (or presence) of a GOF effects in these experiments should be at least noted in the result section and/or discussion.

While it is true that the various mutants show some separation along the PC2 axis, it is important to note the different scales of the PC2 and PC1 axes, as well as the lower contribution of PC2 to the variance in the population (18% versus 28%). Consequently, the effect of PC2 is much lower than that of PC1. We believe this variation likely represents experimental noise, especially since the RNA-seq was performed with one or two single-cell clones for each mutant. Exactly to address the extent of clonal variation, we conducted the scRNA-seq experiment which included 10 single-cell clones for each mutant.

We have rephrased this part of the results section to better explain why we conducted scRNAseq to address this issue.

e, Lastly, GOF activities are hard to delineate in culture – as the authors correctly point out in the discussion. Did the authors try some xenograft experiments and or migration/invasion assays? I am not clear why this system should not also be a terrific way to test whether these alleles expressed from their endogenous locus show GOF activity. In fact, this would be really important given the ongoing discussions about p53 GOF in the field and would certainly add a lot of weight and importance to the paper esp. because it would offer a glimpse into the behavior of human mutants under endogenous transcriptional control. One could presumably even KO MDM2 to ensure high enough mutant p53 levels (akin to the N3a treatment) for functional tests in vivo or in vitro using e.g. migration/invasion assays.

We thank the reviewer for this comment, which touches an important aspect of GOF effects. To investigate pro-metastatic properties, we used R175H-edited HCT116 cells as our model, due to the vast literature on pro-metastatic GOF activities of this mutant. For direct comparison, we engineered a stop mutation at the same site (R175X), which completely abolished p53 protein expression.

First, we compared R175H and R175X cells in migration assays using a Boyden-chamber transwell setup. However, we did not observe any differences in migration under either baseline or N3a treatment conditions (**new Extended Data Fig. 3a-c**). This clearly indicated that acute stabilization of endogenous R175H is not sufficient to increase migration.

Next, we performed a xenograft model of experimental metastasis. We injected the two cell types intravenously and assessed colonization in the lungs (and liver as a secondary metastatic site). Tumor cells were recovered from the lungs and re-injected intravenously into new mice for three rounds of passages. Western blot analysis showed that baseline R175H protein levels progressively increased with each passage (**new Extended Data Fig. 3d-e**). Concurrently, we observed an increase in migration and invasion for R175H (but not R175X) cells in transwell assays, with no change in

proliferation kinetics. This increase in migration and invasion was abolished by siRNA knock-down or CRISPR knock-out of the R175H mutant, indicating that these pro-metastatic properties are mediated by R175H (**new Extended Data Fig. 3f-o**).

Lastly, when late-passage R175H cells with or without R175H knock-out were transplanted subcutaneously into mice, the R175H knock-out diminished liver metastasis but did not affect primary tumor growth (**new Extended Data Fig. 3p-r**).

These new experiments demonstrate that an R175H mutation, even when acutely stabilized with N3a, is not sufficient for pro-metastatic activity. However, it can slowly drive tumor cell evolution to a pro-metastatic state that depends on high-level stable expression of the R175H protein. It remains unclear whether this is due to a slow R175H-induced reprogramming process over time or selective *in vivo* pressures that favor subclones with high-level stable R175H expression and R175H-mediated metastatic properties.

For these reasons, I think that the statement 'Given that GOF properties of a missense mutant are dependent on its constitutive stabilization^{15,23,26}, mutational scans in the HCT116 cell model enable a specific interrogation of LOF effects without being confounded by GOF activities.' should be revised. All assays in the manuscript are using 'constitutive stabilization' (by N3a), so how can GOF properties (if they exist) be excluded? In fact, this system could/should be used to explore these phenomena or – if the authors feel that it is out of the scope of the current paper - at least discussed as a possibility for exploring/testing GOF phenotypes in the future.

Given the recent heated discussions in the field, one could also limit the exploration of GOF activity to certain known/suspected alleles – e.g. R175H, R248W and R273H.

The data presented in **new Extended Data Fig. 3a-c** indicate that treatment with N3a is not equivalent to 'constitutive stabilization.' While the underlying reasons remain speculative, it is known that mutant p53 can be stabilized by numerous factors as delineated systematically in your recent work (PMID: 38580884). Several studies have shown that constitutive stabilization of mutant p53 is mediated, among other factors, by epichaperomes, that are formed in malignant cells, consist of heat shock protein 90 and associated proteins, and shield mutant p53 from ubiquitination by Mdm2, but also other E3 ligases such as CHIP, which are not blocked by N3a (PMID: 37339579). It is therefore plausible that the pro-metastatic properties are not merely a consequence of mutant p53 protection from Mdm2-dependent degradation but also other ligases, or may involve the binding of mutant p53 to epichaperomes, that differ in their activity and composition between different tumor cell types. Mutant p53 stabilization by N3a-treatment is therefore not the same as the constitutive stabilization that is observed in naturally mutated cancer cells.

We have found no evidence that any of the missense variants behave differently from pure LOF (nonsense/frameshift) variants in our fitness assay. Based on these findings, we believe that our HCT116 assay, which measures cellular fitness following specific p53-pathway activation with N3a, primarily measures the LOF caused by a mutation. This is not meant to imply that variants cannot exert GOF activities; rather, our assay setup does not capture them.

The data shown in **new Extended Data Fig. 3** demonstrate, that mutants can exert GOF effects in HCT116 cells following long-term selection in mice. Adapting this to a high-throughput assay may be possible, but would extend far beyond the scope of this manuscript. We acknowledge the importance of this question and will freely provide all reagents to interested researchers to support further studies into GOF properties.

2, In a similar notion, the authors ascribe all the R175 alleles as LOF or pLOF alleles. While the Syn alleles have clearly negative effects on cell growth and thus strongly deplete, 'mis variants showed a highly variable mutation-specific response'. Now, it is a little unclear to me how the enrichment score is calculated: the authors write: The ES is 'defined as the negative log2-transformed fold-change compared to the control treatment' – what is meant by 'the control treatment'? In the figure legend, the authors write: 'Enrichment or depletion is shown as the -log2 fold change versus the mean of the early untreated replicates.'

What is an 'early' untreated timepoint (ie. less than 2 days after transduction to compare to the 3d nutlin time point?) Given the importance of these experiments, please be more specific.

The enrichment score (ES) is calculated in two steps: first, by dividing the abundance of a variant in the test population by its abundance in a control (or reference) population. Second, this ratio (or fold change) is log₂-transformed, and the -log₂FC is reported as the enrichment score. For an N3a-treated test population, the control or reference is a cell population treated with DMSO, the solvent in which N3a is diluted. In the methods section, we use 'control treatment' as a general term because the specific control condition depends on the experiment.

- In the assay with the large DBD library, cells were treated with N3a (dissolved in DMSO) for 8 days, and control cells were treated with DMSO for the same duration (Fig. 4). The same applies to experiments with other Mdm2 or Mdmx inhibitors (Fig. 2f-h). In the experiment with different N3a doses, control cells were treated with the same volume of DMSO as used for the highest N3a dose (Extended Data Fig. 4a). In the APR246 (Extended Data Fig. 4b-c), ZMC1 (Extended Data Fig. 4d-e), and 5FU experiment (Fig. 3a), controls were untreated.
- In the irradiation experiment, control cells were unirradiated (0Gy) (Fig.3a).
- In the starvation experiments (Fig. 3a), control cells were non-starved cells, cultured in regular cell culture medium containing glucose and glutamine.
- In the apoptosis experiment (Fig. 3d), variant enrichment in the annexin-V positive cell fraction was calculated using the annexin-V negative cell fraction as the reference.
- In the time course experiment with 'untreated' cells (Fig. 2e, left panel), we used the 2-week time point as control (reference) to calculate enrichment scores. In the time course with Nutlin-3a (Fig. 2e, right panel), we used the 25-day DMSO-treated cells as control/reference. Since these were two separate experiments, the panels were separated by a line in the original figure. To clarify this, we have now split the two heatmaps and corrected the 25-day 'untreated' label to 'DMSO'. We have also moved the 25-day DMSO-treated next to the 25-day N3a-treated cells.

We have added the detailed information on the control/reference samples to the legends of Fig. 2, 3 and Extended Data Fig. 4.

In any case, the heatmap clearly indicates that syn alleles decrease over time while most of the mis alleles including the most common R175H allele increase over time to a similar degree as the null alleles. Again, it might be nice to point out that – at least by this measure, none of the R175 mis alleles show a GOF phenotype (despite high N3a-induced mutant p53 levels).

We have added the following statement to point this out: "As observed with single mutants (Fig. 1), none of the R175 missense variants significantly enhanced cellular fitness beyond the effect of nonsense mutations, indicating, at least by this measure, the absence of a discernible GOF phenotype."

Minor comments:

1, The engineering of the HCT116 cells to allow for the HRD-mediated swapping of p53 variants, is super elegant, but rather complicated to follow for non-CRISPR/HRD aficionados.

For example, it is unclear to me why mutation of the branch point in a p53 intron leads to the loss of the allele? I thought that mutating of the branch point within an intron disrupts the formation of the lariat structure, leading to errors in splicing. Failure to form a lariat usually result in the retention of the intron or the inclusion of part of the intron in the final mRNA molecule. Why loss of the gene expression? Is it non-sense mediated decay? Is this known for this branch point in intron 5? Maybe a reference or some explanation in the text (or at least in the methods or supplement) would help less-expert readers to understand this better.

The mutation of the branch point in intron 5 indeed leads to missplicing. Despite the barely detectable p53 mRNA levels in HCT116 LSL/ Δ cells, we successfully amplified the cDNA for sequencing. This revealed the complete exclusion of exons 6 and 7, resulting in the aberrant joining of exons 5 and 8. This missplicing creates a frameshift, producing an out-of-frame stop codon in exon 8. Such premature stop codons in non-terminal exons typically trigger nonsense-mediated decay (NMD), explaining the extremely low levels of p53 mRNA observed in these cells. We have added this information to the methods section.

Similarly, I think it would help many readers if the authors explain in more details the process of HRD. For example, the authors write: 'The LSL allele was specifically cleaved by CRISPR/Cas9 nucleases targeting the intronic regions deleted on the Δ allele.' Presumably, the cutting of the LSL allele is increasing the HRD frequency, right? It would be beneficial to mention/explain this for the general audience.

Yes, the DNA double strand break (DSB) induced by cutting the LSL allele stimulates repair by the cellular DSB repair machinery. One of the key pathways for DSB repair is homology-directed repair (HDR), which utilizes available homologous sequences for repair. When co-transfecting donor vectors containing homologous TP53 gene sequences that span the introduced DSB, the cellular HDR machinery can use this exogenous DNA as a repair template. When these donor vectors include variants of interest, these variants may be introduced into the endogenous gene locus during the repair process. We have extended the description of this process in the introductory results section.

2, The authors should comment on/explain the discrepancy between the frequency of successfully targeted clones assessed by PCR versus sequencing.

We assessed the success of gene editing by CRISPR-HDR using two methods: PCR and sequencing. PCR measures the integration of the donor vector into the endogenous TP53 gene locus. Integration via homologous recombination can take place anywhere within the homology arms. This means that the donor DNA can integrate partially without necessarily including the mutation of interest. Therefore, not all PCR-positive clones will contain the desired mutation. Sequencing precisely verifies the presence of the specific mutation encoded by the donor DNA. This method provides an accurate assessment of whether the mutation has been correctly introduced at the target site. Consequently, the percentage of clones confirmed by

sequencing is lower than the percentage of PCR-positive clones. We have expanded the description of Fig. 1b in the results section to better explain this (expected) discrepancy.

3, The authors write: 'Focusing on missense mutations, we explored the functional consequences of replacing each residue with every other amino acid. The screen returned reliable RFS values for 3,425 (99.05%) of all possible 3,458 missense mutant proteins, making this the most comprehensive study of the DBD mutome to date'. Giacomelli et al. (Nat Gen 2018) performed a TP53 saturation mutagenesis screens in an isogenic pair of TP53-wild-type and -null cell lines using nutlin and etoposide as p53 trigger to delineate LOF, GOF and dominant negative features of 8,258 p53 mutant alleles, so that each of the 20 natural amino acids and a stop codon would be represented at each codon position across the entire p53 gene (not 'just' the DBD). As such, while the current screen is more sensitive, how can it be more comprehensive? Please omit, rephrase or explain.

All previous datasets, including the yeast reporter study by Kato, the cDNA-overexpression scans by Giacomelli, Kotler, and Boettcher, as well as the newer base or prime editor scans, have their individual limitations. To illustrate this, we plotted the heatmaps of the functional activity scores for the DNA binding domain, which is the focus of our manuscript. Yellow color shows missing values. The study by Boettcher analyzed the dominant-negative effect with a reporter assay but did not address the LOF and was therefore excluded.

Hanna et al., Cell (2021)

Gould et al., Nat Biotechnol (2024)

Kato et al., Proc National Acad Sci (2003)

Kotler et al., Mol Cell (2018)

Giacomelli et al., Nat Genet. (2018)

This manuscript

The comparison clearly shows that the yeast study by Kato and the more recent base and prime editing studies by Hanna and Gould lack comprehensiveness. The heatmaps have large gaps, severely limiting their utility for structure-function analysis. While the studies by Giacomelli and Kotler were designed to be similarly comprehensive for the DBD region and even include regions outside the DNA binding domain, especially the Kotler study suffered from drop-outs resulting in missing activity scores for numerous variants. The Giacomelli study, which is indeed similarly comprehensive as ours for the DBD region, suffers from poor discriminatory power and does not cleanly separate nonsense and synonymous variants nor known pathogenic from known benign variants (Fayer et al., 2021). Reviewer #2 effectively demonstrated that combining multiple of these datasets can resolve some quality issues and improve the evidence strength for variant interpretation (Fayer et al., 2021). However, the combination of these cDNA-overexpression studies cannot overcome the inherent inability of these assays to correctly assess, for example, variant effects on splicing.

Based on this reassessment, we have rephrased the statement as follows:

“... making this one of the most comprehensive studies of DBD variants to date.” The differences in accuracy and resulting implications for clinical variant interpretation are discussed at a later point in the manuscript. Please also see our comments to the questions of Reviewer #2.

4, It would be nice to label the missense R175 variants that shift upon N3a similarly to wtp53 in the 3rd blot of panel 2b and panel 2d.

Done.

5, Figure Panel 2e – there is a black line before the 25d untreated timepoint, which is confusing. Please explain or remove. Also, why is the 25d timepoint placed at the end of the untreated timepoint? Why is this 25d timepoint showing higher depletion/enrichment or at least more variable results than the 2 week and 4 week timepoint? Presumably the 25d untreated timepoint was chosen to match the 25d treated timepoint?

The black line was meant to indicate two separate experiments. The time course of untreated cells was one experiment, the time course of N3a-treated cells another. We have now split the heatmap clearly into two distinct panels. As elaborated in response to point 2, enrichment of untreated cells (left panel) was calculated relative to the 2-week untreated time point. Enrichment in N3a-treated cells (right panel) was calculated relative to cells that were treated with DMSO for 25 days, as this was the longest duration of N3a-treatment. We have moved the DMSO 25-day time point to the end of the panel to enable direct comparison of DMSO- and N3a-treated 25-day time points.

6, Fig. 7J – it is obvious that SSO treatment stabilizes the mRNA of p53 and also leads to an induction of p21 upon co-treatment of N3a. However, it is unclear to what extent this happens. It would be nice to have wt p53 and wt p53 associated p21 levels in the same experiment.

As shown in Fig. 1f, N3a-treated HCT116 with wild-type p53 induce p21 by 10-25x. HCT116 with the R181L variant induce p21 by 2-3x, which is sufficient to induce cell cycle arrest and delay tumorigenesis. The SSO+N3a-treatment of HCT116 with the g.7675202 ('L137Q') variant resulted in at least 2x increased p21 levels. While this is not as high as the induction seen with wild-type p53, it is comparable to the levels observed with the R181L variant. However, since typically only 50-60% of HCT116 cells are successfully transfected, the actual p21 induction by SSO+N3a is likely underestimated. Consequently, our experiment provides initial proof-of concept.

Optimal SSO efficiency requires rigorous optimization of SSO design, including testing different sequences and chemical modifications. While this future work will be necessary to advance this concept into a therapeutic strategy, it is beyond the scope of the current manuscript.

We are grateful to your extremely helpful review! Many thanks!

Reviewer #2:

Remarks to the Author:

Overall, the first author and team should be commended for an incredible amount of work. If the replicates are real biological replicates and not PCR replicates, this will be an addition to the existing trove of variant effect data for TP53. Its impact will depend on the ability to validate the data for clinical use which is not something the authors attempted here.

Thank you for your positive feedback on our work. As detailed later, the cell editing was performed once, but the selection and functional assays with this cell library were indeed conducted in real biological replicates, not just PCR replicates. To enhance the clinical relevance of our study, we have incorporated additional analysis using the ClinVar database.

My major concern is that throughout this paper, the authors wrote the manuscript in a way that makes it seem as though the methods were developed in their lab, for this work, for the first time. This is far from reality. Deep mutational scans (DMS) that employ CRISPR-mediated homology-directed repair (CRISPR-HDR) to install edits are called saturation genome editing (SGE) – a method developed 10 years ago by Jay Shendure’s lab (Findlay et al. Nature 2014). There are many citations missing for both SGE and other CRISPR based methods to install and measure functional effects of variants (listed below). Additionally, nearly the entire protocol (minus the upstream cell line engineering and cre step) and analysis performed here is standard operating procedure for SGE specifically and DMS generally – splitting up variant libraries into Illumina-sized chunks, using synthetic oligos to make variant libraries, and the exon normalization method, yet none are cited. I’m hopeful that these missing citations were not an intentional omission of the literature. And for the students’ sake, I am hopeful y’all didn’t actually reinvent all these wheels. There are plenty of resources available through the Atlas of Variant Effects Alliance.

We apologize for any oversight in our citations and acknowledge the importance of properly crediting previous work. As researchers focused on p53, we did not intend to imply that these methods were developed solely by our lab. We greatly respect and were inspired by the foundational work of Jay Shendure’s lab and others in the field.

To address your concerns, we have thoroughly revised the manuscript to ensure all relevant citations are included, particularly those related to SGE methods developed by others. **The revised manuscript now credits these established methods appropriately** and aligns our study within the broader context of existing research.

While we recognize that our workflow builds upon these foundational methods, it also includes several key adaptations specific to our study of p53. As clear genotype-phenotype correlations are crucial, many SGE studies were done in the haploid cell line Hap1. However, Hap1 cells carry a partially active *TP53* point mutation (PMID: 29089570). Moreover, haploid cells die through activation of a p53-dependent cytotoxic response and p53-deficiency stabilizes the haploid state (PMID: 28808015). Because of these connections between haploidy and p53 activity, Hap1 and other haploid cells are not suitable for SGE of *TP53*. Therefore, we chose to conduct our screen in diploid cells, inactivating one of the two *TP53* alleles prior to SGE. Additionally, to avoid the technical selection pressure for loss-of-function variants caused by CRISPR-Cas9-induced p53 activation (PMID: 32424350, 29892067, 29892062), we inserted a floxed stop cassette to reversibly switch off p53 expression during editing, which was later removed with Cre recombinase.

These adaptations distinguish our approach from previous SGE studies and were crucial for obtaining high-quality functional data for *TP53* variants. We believe that these modifications may also inspire future SGE designs by others, particularly for genes involved in the DNA damage response or DNA repair, which could benefit from performing the gene editing step in the absence of expression of the targeted gene.

A second major complaint is that the words pathogenic and likely pathogenic are protected terms. You cannot deem a variant pathogenic or likely pathogenic based solely on the outcome of a functional assay. Functional data must be translated into evidence for clinical variant classification and combined with other sources of evidence to classify variants as pathogenic, likely path, VUS, likely benign or benign (Richards ACMG/AMP guidance and Brnich ClinGen SVI update). Please see ClinGen's guidance for interpreting variants in TP53 (ClinGen CSPEC, <https://cspec.genome.network/cspect/ui/svi/doc/GN009>). This manuscript must be heavily edited to avoid labeling variants pathogenic unless already classified as such. Please see additional guidance specifically for publishing clinically relevant mutational scans (Gelman 2020).

Thank you for your important feedback regarding the use of the terms "pathogenic" and "likely pathogenic." We acknowledge that these terms are protected and must be used with caution. We have revised the language to ensure we do not label variants as pathogenic or likely pathogenic based solely on the outcomes of our functional assays. Following Brnich et al.'s recommendations, we now refer to variants as either functionally normal or abnormal. Functionally abnormal variants are sometimes also referred to as loss of function (LOF) or "potentially pathogenic," as you suggested.

Additionally, we have evaluated our study results using variant classifications from the ClinVar database and the *TP53* variant curation expert panel (**new Extended Data Fig. 8, new Supplementary Table 6**). These analysis results and the corresponding adjustments are described in more detail in the response to your specific comments.

Third, one of the regions of TP53 where variants are the hardest to classify is the tetramerization domain. This is not covered in this study.

The tetramerization domain (TD) is encoded by parts of exons 9 and 10, which are separated by more than 3 kb from the LSL cassette used for selection. This significant genetic distance hindered efficient editing in our cellular model. Additionally, according to the TCGA Pan Cancer Atlas, exons 9 and 10 together account for only 5.75% of all mutations and 2.34% of missense mutations in tumors. Similar frequencies are observed in the GENIE database, which more accurately reflects real-world data. Due to these technical challenges and the smaller number of cases, we did not include exons 9 and 10 in our current study.

Fourth, a complete data set is not included in the supplemental tables and the data and code have not been submitted to a stable repository.

The most relevant data sets were already included in old Supplementary Table 3 which listed all variants with identifiers (DNA, RNA, protein level), read counts for all replicates and conditions and final RFS scores. In addition, the large source data file contained tables with the detailed data for all Figures and Extended Data Figures.

We have now submitted all raw sequencing data in Fastq format to the stable EMBL-EBI Biostudies data repository. Furthermore, as requested, we are in the process of uploading the results to MaveDB. During this process we encountered some technical problems uploading our score tables. We are in contact with Alan

Rubin and confident that we will solve this issue soon. In the meanwhile, all scores are also available in **new Supplemental Table 2**. The code used for the analysis was made available on GitHub. Links are provided in the answer to your last comment.

Specific comments:

Mutome is not (and should never be) a word.

We have replaced “mutome” with “mutational landscape”.

The first sentence of the abstract – over 2000 mutations described in patients... Where is this number coming from? If ClinVar, it is 2000 variants described in individuals, if TCGA 2000 mutations described in tumors. Either way, the sentence should be edited.

The number comes from UMD, so we edited to “tumors”.

The final paragraph of the introduction is missing all the citations for saturation genome editing by HDR, base and prime editing, see below. This list is not comprehensive, there could be some I've missed.

Omitted saturation genome editing citations:

Findlay et al. Nature 2014 (method development)

Findlay et al. Nature 2018 (BRCA1)

Meitlis et al. AJHG 2020 (ssODN-based HDR, CARD11)

Radford et al. Nature Comms 2023 (DDX3X)

Sahu et al. BioRxiv 2023 (ssODN-based HDR, BRCA2)

Huang et al. BioRxiv 2023 (BRCA2)

Findlay lab, BioRxiv 2023 (VHL)

Omitted saturation prime editing citations:

Erwood et al. Nature Biotech (NPC and BRCA2 2022)

Omitted saturation base editing citation:

Hanna et al. Cell 2021 (multiple genes)

Etc

We originally focused the manuscript on p53 and referenced all deep mutational scans specific to p53. We omitted studies on other genes as we did not mean to provide a comprehensive review of the huge literature on DMS and SGE. However, we recognize the importance of citing key studies to credit the underlying method development and contextualize our research within the broader field. Therefore, we have now included the relevant references in our introduction.

Figure 1 DEF?

Why is WT clone 2 showing some LOF? Can you find a structural rearrangement by sequencing the locus?

There is always some experimental variation between different clones of the same genotype. To assess the extent of this variation, we compared 12 genotypes with 10 clones each side-by-side in a pooled experiment by scRNAseq (Extended Data Fig. 2). This experiment showed that all N3a-treated WT clones clustered together and separated from LOF clones, strongly suggesting that WT clone 2 was an outlier. Therefore, we did not further investigate the underlying cause. However, we believe it is important to show that there is some degree of clonal variation to emphasize the need for high coverage in DMS experiments.

Saturation genome editing specifically and DMS generally are accepted methods for understanding the effect of genetic variation. Most of figures 2 and 3 do not add much information beyond the full mutational scan.

Figures 2 and 3 focus on the most frequently mutated codon, R175. While R175 is included in the full DBD scan, these focused scans provide a much deeper understanding of the functional consequences of mutations at this critical codon.

Specifically, by analyzing cellular fitness at different time points post-p53 activation, the R175 scans reveal variant-specific depletion kinetics. Additionally, sorting apoptotic cells elucidates differences in depletion kinetics through distinct effector program engagement. The consistency of depletion/enrichment patterns across different p53 activating conditions – including irradiation, chemotherapy, nutrient starvation, and Mdm2 inhibition – justifies Nutlin-3a as the optimal p53 activator for achieving the highest dynamic range in subsequent mutational scans.

Furthermore, similar fitness effects under treatment with a range of chemically distinct Mdm2/Mdm4 inhibitors confirm that the observed effects of Nutlin-3a treatment are on-target. These insights derived from the R175-focused scans add significant value beyond the full mutational scan and will be especially appreciated by p53 researchers.

Typo line 220

We had introduced 'mis' as an abbreviation for 'missense' earlier in the text. To avoid any confusion, we have now spelled it out in full.

The APR-246 or ZMC1 experiments are a good use of the technology.

Thanks.

It is nice that there is validation across 2 cell lines for a small number of variants. However, they are both cancer cell lines and therefore less informative than a second cell line that is closer to normal.

The choice of tumor cell lines for mutagenesis is driven by both technical and biological considerations. Technically, CRISPR editing is far more efficient in established cell lines compared to primary cell types. Biologically, mutational scans in primary, non-transformed cells would be valuable because *TP53* germline mutations in these cells are associated with Li-Fraumeni syndrome. However, somatic mutations in *TP53* are by far more frequent than germline mutations and are not necessarily founder mutations occurring in non-transformed cells. Cancer genome studies revealed that *TP53* mutations are often subclonal and become enriched during tumor progression or therapy, as they provide a selective advantage even in already transformed cells. Notable examples include the 'Vogelstein model' of colorectal tumorigenesis, where *TP53* mutations occur at the adenoma-to-carcinoma transition (PMID: 2188735), the frequent subclonal origin of *TP53* mutations in NSCLC (PMID: 37046096), and therapy-related AML, often driven by the selection of a pre-existing *TP53*-mutant subclone (PMID: 25487151). Thus, many *TP53* mutations arise in already transformed cells, underscoring the clinical relevance of performing *TP53* mutagenesis in transformed cell lines. Based on these technical and biological considerations, we selected the colorectal HCT116 and NSCLC H460 cell lines as models for our study.

Page14 Lines 279-281. This is an overreach. Many DMS experiments regardless of the way the variants are introduced identify mechanism and separation of function variants.

This statement was made with respect to p53. For p53, the engagement of effector functions is heavily regulated by the p53 protein level. Low p53 expression primarily induces cell cycle arrest, while higher expression levels shift the response towards apoptosis (PMID: 8843196). DMS experiments relying on cDNA overexpression can therefore easily overestimate the apoptotic potential of a variant. To accurately assess the apoptotic potential of p53 variants, it is crucial to evaluate them at physiological expression levels, which are under control of non-coding elements absent in cDNAs. The CRISPR-based DMS approach with SGE ensures physiologically controlled expression levels, providing a more accurate representation of a variant's function. p53 is therefore a prime example where DMS by SGE is superior for identifying such separation-of-function phenotypes.

We have rephrased the statement to better focus it on the p53 context.

Lines 295-300 page 15. Describing both Tile-seq and standard saturation genome editing experimental design. Both are published methods and should be cited (see Findlay et al and the Roth lab's publications).

Done. We have stated that our design is similar to previously published methods and referenced the mentioned publications.

Page 16 lines 317-319 – Relative fitness scores are nearly the same as a procedure also developed by Greg Findlay for SGE (Findlay et al 2018).

Done. We have stated that our design is similar to the previously published method and referenced the mentioned publication.

Page 16 lines 330-331 – kind of. This library contains many extraneous variants that aren't terribly clinically useful (multiple nucleotide aa changes, all the indels). And the Giacomelli and Boettcher papers scanned the whole protein.

Multiple nucleotide changes are rare in patients, but necessary to assess the impact of every possible amino acid substitution, which is crucial for comprehensive structure-function analyses.

While indels are less prevalent in tumors than missense mutations, they proved highly informative to demonstrate experimentally that mRNAs with indels are uniformly degraded. This finding is particularly significant in the context of the p53 research community, where it is often overlooked that mRNAs with indels are subject to nonsense-mediated mRNA decay (NMD), resulting in only negligible amounts of truncated proteins. Several publications in the p53 literature have incorrectly attributed pro-tumorigenic GOF effects to these putative truncated proteins (PMID: 27759562).

Therefore, we included a comprehensive panel of indel variants to show their consistent degradation at the mRNA level. Importantly, NMD is a mutation effect that can only be demonstrated through a CRISPR-based scan, unlike the cDNA overexpression method used by Giacomelli, Kotler and Boettcher.

Page 18 lines 375-380 – if I recall correctly, an analysis of mutational propensity was also performed by Giacomelli et al. and they found something similar. Citation?

In fact, mutational propensities were already considered to explain the TP53 mutation spectrum long time ago (Denissenko et al., Science 1996; Denissenko et al., PNAS 1997). We have used the newest and more comprehensive version of

mutational probability data directly from COSMIC and the Alexandrov et al., 2020, publication instead of the older datasets used by Giacomelli et al. in 2018. The publication by Giacomelli et al., showing accurate modelling of the *TP53* mutation spectrum by combining functional and mutational propensity data, was already mentioned in our discussion: “Moreover, the unique *TP53* mutation spectrum of cancer patients can be computationally modeled fairly accurately without requiring the consideration of GOF effects.”

Page 18 lines 395-388 – unless you’re going to calculate a positive and negative predictive value or likelihood ratio for identifying truly pathogenic variants (that have a label in ClinVar), this is an overreach.

We appreciate your comment and, in response, have made several revisions:

- We have removed the last part of the sentence referring to pathogenicity and added two new paragraphs to address the potential of the RFS in classifying variant pathogenicity.
- Guided by your valuable literature recommendations and your publication on using cDNA-based *TP53* MAVE studies for clinical variant interpretation (Fayer et al., 2021), we utilized the ClinVar database with over 3,400 classified *TP53* variants. As you correctly pointed out in your publication, previous cDNA-based *TP53* MAVE studies failed to cleanly separate known pathogenic from known benign variants (Fayer et al., 2021). However, our CRISPR-derived RFS achieved near-perfect separation (**new Extended Data Fig. 8a-c**). This distinction was evident for the large subset of ClinVar variants with ≥ 1 -star review status but also when focusing on missense variants, and even variants classified by the *TP53* variant curation expert panel.
- Receiver operating characteristic (ROC) and precision-recall curves (PRC) further validated the excellent performance of the RFS as a classifier of pathogenicity, with AUC values exceeding 0.998 (**new Extended Data Fig. 8d-e**). We did not include ROC and PR curves for the VCEP variants because of their small number.
- To evaluate the strength of evidence for variant interpretation, we calculated the odds of pathogenicity (OddsPath) according to Brnich et al., 2019, and analogous to your publication (Fayer et al., 2021). We used ClinVar variants classified as benign/likely benign or pathogenic/likely pathogenic as truth sets (**new Supplementary Table 6**). Depending on the stringency of the selected truth sets, we obtained OddsPath values up to 50.2 and 0.0076, respectively. These OddsPath values correspond with the strong strength of evidence codes PS3 and BS3 by the ACMG/AMP guidelines and surpass the strength of evidence provided by previous cDNA-based *TP53* MAVE studies, even combining four of these studies (Fayer et al., 2021).

This comprehensive analysis robustly supports the value of our MAVE study as a functional data layer of evidence for variant interpretation. We are looking forward to seeing how our dataset is accepted by the community. Thank you for highlighting this critical aspect. We appreciate your guidance through your review, which has significantly strengthened our study.

Figure 6. Could use a more in-depth comparison to Kato. Kato has been the gold standard for assessing variants in TP53 for clinical variant interpretation.

We have added a detailed comparison of our RFS values to the transcriptional activities reported by Kato, based on the data that were already included in the Supplementary Tables. We observed a highly significant negative correlation,

indicating that positive RFS values from our screen are associated with low transcriptional reporter activity in yeast (**new Extended Data Fig. 10a, b**). The correlation coefficients were in the same moderate correlation range as observed in our comparison with the Giacomelli and Kotler datasets.

Also, this is a bit unfair to Giacomelli, they did their screens in 2 different conditions. You're only comparing to one of them. Fayer et al. AJHG 2021 found that combining the 4 datasets from Giacomelli and Boettcher gave them more predictive power to correctly predict variants of known effect (ClinVar variants).

We did not mean to discredit any of the previous DMS studies, all of which have significantly advanced the field and improved clinical variant interpretation.

In response to Reviewer #1, we have included a comparison of the functional activity heatmaps obtained from our and all previously published *TP53* datasets to illustrate the comprehensiveness of ours. While our dataset may not be the most comprehensive compared to the Giacomelli dataset, it is at least one of the most comprehensive for the DBD region, which our statement explicitly referred to. **We have rephrased our statement to avoid any unintended offense.**

From your experience with the older datasets, you know that the main issue is data quality. As highlighted by your publication (Fayer et al., 2021), none of the existing datasets cleanly separates nonsense from synonymous nor pathogenic from benign variants. While combining various datasets might mitigate some quality issues, our new analysis demonstrates that our dataset alone matches, if not exceeds, the predictive power of the others, even when combined.

Importantly, while combining datasets may resolve some quality issues, it cannot overcome the inherent problem of cDNA-based expression screens being blind to splice effects. Our DMS is the only one that correctly identified the LOF associated with the g.7675202A>T variant, which was previously (erroneously) described as a 'L137Q' missense variant but is actually a splice variant.

Given your own experience with the older datasets, we believe you will appreciate the increase in data quality provided by our study.

Page 20, Line 411, overreach. "Likely pathogenic" is a protected term as clinical variant class, maybe "potentially pathogenic"?

We have revised the term to "potentially pathogenic" as suggested, to ensure appropriate use of clinical variant classification terminology.

Page 21, lines 448-451, overreach, this is not the first mutational scan to correctly observe splice defects, see missing saturation genome editing citations above

The statement clearly referred to "conventional mutome scans based on the overexpression of spliced cDNAs", which are obviously blind to splice effects. And yes, it is the advantage of SGE that it can correctly identify such splice effects. We have included the references to SGE studies conducted for other genes in the introduction.

Page 22, line 472, What you describe is validation of the damage or splice defect not pathogenicity. Determining pathogenicity requires a completely different procedure (see Clingen CSPEC for TP53)

We have changed "pathogenicity" to "LOF".

Page 23, lines 494-496, nope. You cannot say this. Determining pathogenicity requires a completely different procedure (see Clingen CSPEC for TP53)

We have changed to “discriminates functionally normal from abnormal variants”.

Page 24, line 497 “intraexonic” is confusing term. Aren’t they just exonic?

Yes, we corrected this.

Page 25, lines 507 -510 Again, what you describe is validation of the damage or splice defect not pathogenicity. Determining pathogenicity requires a completely different procedure (see Clingen CSPEC for TP53)

We have changed “pathogenicity” to “LOF”.

Page 26 you can’t just call any variant pathogenic based on the outcome of the screen. g.7675202A>T, could potentially be pathogenic. Is it seen in tumors? Individuals with Li-Fraumeni syndrome?

We have rephrased the paragraph. Variant g.7675202A>T has indeed been seen in multiple tumors (14 entries in the UMD TP53 mutation database; 8 in the IARC/NCI TP53 mutation database; 9 in TCGA; 18 in GENIE); however, no entry in ClinVar.

Line 556 - more frequent in tumors not patients

We have corrected to “tumors”.

Line 557 – nope. You cannot say this. Determining pathogenicity requires a completely different procedure (see Clingen CSPEC for TP53)

We have changed to “potentially pathogenic”.

Lines 659 – 662 nope. You cannot say this. Determining pathogenicity requires a completely different procedure (see Clingen CSPEC for TP53)

We have changed to “functionally deleterious”.

Methods:

Lines 826 – 832. What was the replicate structure for these experiments? Were they complete/biological replicates e.g. transfected into 3 separate populations of cells or partial/technical replicates, a single transfection with 3 separate extraction/PCR reactions? This is important to describe accurately.

Genomic editing was performed a single time, with cells transfected once using each exon-library and subsequently selected with blasticidin and puromycin. The entire functional assay, including Cre transfection to activate p53 expression and the selection with either Nutlin-3a (N3a) or DMSO, was conducted in true biological triplicates for each exon library. These triplicates were performed sequentially on different days rather than in parallel. This information has been added to the methods section.

Line 843-844: What is meant by editing-specific primers? Would these only pick up edited alleles? How much of the final cell libraries are unedited?

The nested PCR strategy is illustrated in Extended Data Fig. 1a:

In the first PCR, we used the orange primer set, which includes a forward primer in intron 4 (upstream of the LSL cassette) and a reverse primer in intron 9 (downstream of the HDR donor). This setup ensures that the PCR product is obtained only from the endogenous gene locus, not from random integration of donor DNA. The PCR only amplifies successfully after Cre-mediated removal of the LSL cassette due to its large size. Notably, the first PCR amplifies both the edited and inactive Δ -allele and was performed with over 50 PCR reactions to ensure sufficiently high coverage.

In the second (nested) PCR, all products from the first step were pooled, and an aliquot was subjected to PCR using one of the blue/green primer pairs, depending on the targeted exon. These primer pairs flank the targeted exon and specifically amplify the edited allele, as the primer binding site is mutated in unedited cells and repaired by recombination with the HDR donor.

We validated the specificity of the nested PCR with single-cell clones at various stages of editing, as demonstrated here for exon 8:

- Δ /LSL-par: Parental screening cells with one Δ -allele and a second parental (unedited) allele containing the LSL cassette. The parental allele has alterations at the primer binding sites in introns 5 and 7, as shown in Extended Data Fig. 1a.
- Δ /par: Cells derived from Δ /LSL-par cells through Cre-mediated removal of the LSL cassette.
- Δ /LSL-edit: Δ /LSL-par cells after successful HDR-editing (restoration of the secondary primer binding site) but before Cre-mediated removal of the LSL cassette.
- Δ /LSL-edit: Cells resulting from Δ /LSL-edit cells after Cre-mediated removal of the LSL cassette.
- H₂O/NTC: PCR contamination controls.

Since the nested PCR enriches for edited cells, the NGS analysis of the nested PCR product does not provide information about the percentage of unedited cells in

the final cell libraries. However, we estimated the overall editing efficiency by single-cell cloning of puromycin-resistant (and presumably edited) cells following editing with single donor constructs. As illustrated in Fig. 1b, approximately 75% of clones were PCR-positive for donor integration, and 56% contained the desired mutation when analyzed by sequencing.

We have added a sentence explaining the editing specificity of the PCR to the methods section.

Alignment by exact matching analysis was done in the Sanger's DDX3X paper (Radford et al.).

We have added the reference to the methods section.

863-872 Score not sufficiently explained in methods. I think I know what was done, but it should be explained in more detail so people could replicate the score.

We have expanded the description of how our Relative Fitness Score (RFS) was calculated and included the formula in the methods section.

“To obtain a score that is comparable across different libraries and screens, the ES was further normalized into a relative fitness score (RFS) by the following formula:

$$RFS_{Ex}(ES) = \left(\frac{ES - \tilde{x}_{ex}^{non}}{\tilde{x}_{ex}^{non} - \tilde{x}_{ex}^{syn}} \right) * 2 + 1$$

with \tilde{x}_{ex}^{non} denoting the median of the scores for all nonsense mutations in a specific exon ex and \tilde{x}_{ex}^{syn} denoting the median of all synonymous mutations in this exon. RFS scores were calculated for each replicate, then, as our total score, we obtained the median (RFS_{median}) over all three replicates.”

Additionally, we have added a new analysis of our experiment using the published Enrich2 package as an orthogonal analysis approach. We normalized the Enrich2 scores by nonsense and synonymous mutations as described above to obtain the Enrich2-derived RFS scores ($rfs_enrich2$). These showed a near-perfect correlation with our original RFS values (rfs_median). Notable deviations were primarily limited to a few WT-like (RFS-negative) variants with high standard errors ($rfs_enrich2_SE$):

We also used the standard error to calculate adjusted p-values for each score using a one-sided z-test under the null hypothesis that the variant's score is equal or lower from the median of the synonymous (WT-like) variants.

RFS-transformed Enrich2 scores with error estimates and adjusted p-values have been added to Supplementary Table 2.

I can't find the complete data set in the supplementary tables. Table S2 has the variant identifiers and variant sequence, table S7 has missense variants and crispr_rfs scores. But I cannot find a table with both of those together for the complete data set of variants. Can a single table be generated with all the variant identifiers and scores along with scores from individual replicates, variants counts?

The table you were looking for is the old Supplementary Table 3.

The old Supplementary Table 2 contained all 9,225 variants included in the library, along with our internal mutID and gene-, cDNA-, and protein-level identifiers (according to standard nomenclature), as well as the exact donor sequences used to generate each variant. While this detailed information might be interesting for specialists, it is not essential for most readers.

The old Supplementary Table 3, however, was the more important table as it contained the functional assay results. It included all 9,225 variants with the same four identifiers (mutID, gene, cDNA, protein) in the same order as in Supplementary Table 2 and reported read counts from all replicates and treatments, abundance values, and our final score (rfs_median).

Following your recommendation, we have merged the two tables into a single **new Supplementary Table 2**. Please note, all supplementary tables reporting information on the DBD experiment contain the same mutID as stable identifiers, facilitating easy merging, further processing, and re-analysis of the data.

The contents of the supplementary tables are insufficiently described.

We have added a separate sheet to each supplementary table, providing a detailed description for each column.

Is the code available on Github or another stable repository?

Code is available on Github: https://github.com/IMTMarburg/TP53_SGE

Data variants and scores must be uploaded to MaveDB and sequence reads uploaded to SRA before publication.

We have uploaded all sequences as Fastq files to the stable repository EMBL-EBI Biostudies. The accession numbers are as follows:

- E-MTAB-12734: bulk RNAseq of single mutants (Fig. 1d-f):
<https://www.ebi.ac.uk/biostudies/arrayexpress/studies/E-MTAB-12734?key=17f70ec6-66d5-4c10-98a1-081de5a3dc50>
- E-MTAB-13904: scRNAseq (Extended Data Fig. 2)
<https://www.ebi.ac.uk/biostudies/arrayexpress/studies/E-MTAB-13904?key=43b973a6-5f76-4b21-a51f-97d4b8d245bf>
- E-MTAB-14322: R175 SGE experiments (Fig. 2+3)
<https://www.ebi.ac.uk/biostudies/arrayexpress/studies/E-MTAB-14322?key=b3edeeb8-91e9-40e6-95f4-e79925f2c961>
- E-MTAB-12857: TP53 exon5-8 SGE experiments (Fig. 4-6)
<https://www.ebi.ac.uk/biostudies/arrayexpress/studies/E-MTAB-12857?key=68735779-14b1-4f8e-906d-a59c8ecea159>

- E-MTAB-12861: TP53 exon5-8 SGE cDNA sequencing (Fig. 6-7)
<https://www.ebi.ac.uk/biostudies/arrayexpress/studies/E-MTAB-12861?key=365e1957-6e91-49ae-bb16-ac7a678fb29a>

We are also in the process of submitting the dataset to MaveDB. Unfortunately, we encountered some technical problems uploading our score tables and had difficulties reaching a contact. We are now in contact with Alan Rubin and confident that we will resolve the issue soon. However, all read counts and scores are also provided in the new Supplementary Table 2.

Lea Starita, University of Washington

We are grateful to your extremely helpful review! Many thanks!